# *Teach Better or Show Smarter?* On Instructions and Exemplars in Automatic Prompt Optimization

**Xingchen Wan, Ruoxi Sun, Hootan Nakhost, Sercan Ö. Arık**
Google Cloud AI Research
{xingchenw, ruoxis, hootan, soarik}@google.com

## Abstract

Large language models have demonstrated remarkable capabilities but their performance is heavily reliant on effective prompt engineering. Automatic prompt optimization (APO) methods are designed to automate this and can be broadly categorized into those targeting *instructions* (instruction optimization, IO) vs. those targeting *exemplars* (exemplar optimization, EO). Despite their shared objective, these have evolved rather independently, with IO receiving more research attention recently. This paper seeks to bridge this gap by comprehensively comparing the performance of representative IO and EO techniques both isolation and combination on a diverse set of challenging tasks. Our findings reveal that intelligently reusing model-generated input-output pairs obtained from evaluating prompts on the validation set as exemplars, consistently improves performance on top of IO methods but is currently under-investigated. We also find that despite the recent focus on IO, how we select exemplars can outweigh how we optimize instructions, with EO strategies as simple as random search outperforming state-of-the-art IO methods with seed instructions without any optimization. Moreover, we observe a synergy between EO and IO, with optimal combinations surpassing the individual contributions. We conclude that studying exemplar optimization both as a standalone method and its optimal combination with instruction optimization remain a crucial aspect of APO and deserve greater consideration in future research, even in the era of highly capable instruction-following models.

## 1 Introduction

Significant advancements in large language models (LLMs) have revolutionized various natural language processing tasks [9, 4, 1, 15]. One notable aspect of LLMs, however, is their sensitivity to the input "prompts," which has given rise to the burgeoning field of prompt engineering [26, 38]. On black-box LLMs where we can neither modify or access internal parameters, prompt engineering involves crafting input prompts that effectively guide LLMs to generate desired outputs. Starting from manual processes requiring human expertise, the complexity and volume of prompts have necessitated the development of automatic prompt optimization (APO) methods aiming to streamline and automate prompt generation, thereby alleviating the burden of manual intervention. Broadly, since prompts consist of instructions and exemplars, we may roughly categorize APO into *instruction optimization* (IO) and *exemplar optimization* (EO) approaches. IO focuses

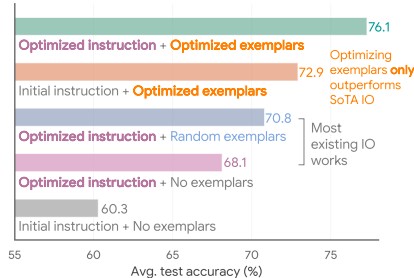

Figure 1: Average performance over >20 tasks on PaLM 2 – We compare and combine APO targeting *exemplars* and *instructions*, and find that how we optimize exemplars (orange) can eclipse how we optimize instructions despite current research favoring the latter (blue and purple), whereas optimizing both is the best (cyan) within similar budget.

38th Conference on Neural Information Processing Systems (NeurIPS 2024).

on refining the textual instructions provided to LLMs that contain task-specific information (i.e., to *teach*), whereas EO emphasizes the selection of relevant examples to guide model behavior (i.e., to *show*). Partially driven by the improved instruction-following ability of LLMs, the research attention has increasingly shifted towards IO, especially using LLMs themselves as optimizers [66, 35, 48].

While EO and IO approaches address the similar overarching problem, they have evolved somewhat independently, with a few exceptions [14, 47]. Indeed, as we elaborate in §2, EO approaches are often based on simple, handcrafted templates without explicit instruction optimization [21, 45], while IO methods seldom optimize exemplars and often rely on random validation set samples [35], require additional fixed exemplars on top of the validation set [66, 16], or consider the "zero-shot" setup with no exemplars at all [48]. Whereas the lack of IO in EO methods is somewhat understandable as many EO approaches predate instruction finetuning [49] and, subsequently, instruction-following models that are sensitive to instructions, the inverse is much less so: concretely, almost *all* existing IO approaches *already* require a labeled dataset as the validation set, and are therefore, by definition, not "zero-shot". With the inputs, labels, and, if applicable, model-generated intermediate outputs (e.g., reasoning steps) on the subset of the validation set that the model has answered correctly, we already have a set of exemplars as a free side-product whenever we perform IO, independent from and on top to any additionally provided, human-annotated exemplars. A common argument for *not* focusing on EO, such as mentioned in Pryzant et al. [35], is the goal to focus on one objective at a time[1]. However, given the common practical goal of and the interplay between EO and IO [30], we argue they should not be treated separately – it is instead critical to understand their relative importance and combined impact, and, where necessary, optimize them jointly for the best performance-cost balance.

This is, to our knowledge, where there is a gap in the literature that we aim to bridge. To do so, on a diverse suite of challenging BIG-Bench and MMLU tasks, we compare the performance gain brought by various representative, state-of-the-art (SoTA) IO and EO methods on the fairground with PaLM 2, Gemini (1.0/1.5) and GPT models to foster better scientific understanding of different APO techniques. While IO comfortably improves the baseline prompts before any instruction or exemplar optimization, this, at best, portrays an incomplete picture. Under the same setup, with simple yet effective EO methods on the model-generated exemplars on the validation set, we show:

- Intelligently incorporating exemplars generated by the target model itself on the validation set significantly and consistently improves performance on top of recently proposed IO methods;
- The performance gains realized by choosing appropriate exemplars via methods as simple as random search can eclipse the improvements brought by SoTA instruction optimization. As a concrete example, as shown in Fig. 1, with a simple optimization routine on exemplars, seed instructions *before any optimization* outperform optimized instructions obtained with complex IO but with random exemplars most commonly used.
- There exists a synergy between EO and IO, and optimally mixing-and-matching IO and EO is greater than the sum of its parts under a comparable computational budget.
- SoTA IO might be itself implicitly generating exemplars, and these exemplars, while somewhat unintentional, contribute more to the performance than the rest of the instruction that IO methods are meant to optimize.
- While arguably receiving less research attention recently, exemplar optimization remains a crucial design consideration in APO. Even in an era of highly capable instruction-following LLMs, the significance of exemplar optimization should not be relegated to an afterthought, and better exemplar optimization both as a standalone tool and as a combinable component with IO is crucial for APO.

## 2   Preliminaries

*Prompts* are natural language inputs to LLMs. Denoting an input task query as $x$, a few-shot prompt $P(x)$ may be represented as $P(x) = [I, e_1, ..., e_k, x]$ where $I$ denotes an *instruction* and $\{e_1, ..., e_k\}$ denote $k$ *exemplars* (or interchangeably, *demonstrations*), each of which is a concatenation of other queries and their outputs (including both the final answer and any possible intermediate outputs) which resemble the current query $x$ or may otherwise guide the LLM to better handle the current task, and we show a common prompt template organizing these components in Fig. 2 – note that not all components are required: e.g., zero-shot prompts feature no exemplars.

---

[1] *"The proposed algorithm is about optimizing the language of prompts, as opposed to selecting the best examples for few-shot learning."*

*Automatic prompt optimization* (APO) aims to automatically design $P(x)$ via optimization. We broadly consider *black-box* API-only LLMs[2]. The proposed framework assumes a *validation dataset* $\mathcal{D}_{\text{val}} := \{(x_i, y_i)\}_{i=1}^{n_{\text{val}}}$, where $x_i$ and $y_i$ represent validation inputs and targets, a *performance metric (e.g., accuracy)* $g(\cdot, \cdot)$, and aims to find the optimal prompt $P^*(x)$ to be used at test time, which is empirically the maximizer on $\mathcal{D}_{\text{val}}$:

$$P^*(x) = \arg\max_{P(\cdot) \sim \mathcal{P}} \mathbb{E}_{(x,y) \sim \mathcal{D}_{\text{val}}} \Big[ g\Big( f_{\text{LLM}}(P(x)), y \Big) \Big], \quad (1)$$

where $f_{\text{LLM}}(\cdot)$ denotes a textual LLM output given input and $\mathcal{P}$ denotes the *search space*, whose definition allows a broad categorization of APO methods into *instruction optimization* methods targeting instructions in Fig. 2, *exemplar optimization* methods targeting exemplars in Fig. 2 and approaches that tackle both.

**Exemplar optimization (EO).** Efforts to optimize exemplars started soon after the discovery of in-context learning (ICL) [6] via retrieval-based approaches to identify the closest labeled examples [28, 61, 51, 64], influences and sensitivity [31, 8], and learning-based approaches [58, 55]. Toolkits like DSPy [21] adopt EO as the main APO paradigm. Works [45, 46, 60, 22, 41, 59] have also extended EO to model-generated exemplars in LLMs and multimodal models. Lastly, by framing EO from an active learning angle, Margatina et al. [29] provide a comprehensive understanding and comparative analyses. These works, however, principally analyze different EO strategies only, nor do they analyze from the angle of APO. Many of these works also primarily focus on and draw findings from earlier and simpler tasks that are arguably less challenging to SoTA LLMs.

Solve the following math problems by following the steps:
1. When multiplying or dividing two negative numbers, the result will be positive.
2. When multiplying or dividing a positive and a negative number, the result will be negative.
3. When adding or subtracting a negative number, it is the same as adding or subtracting its positive counterpart.

```
((6 - 0 * 5 + -3) * (6 - -7 + -2 - -7)) =
(6 - 0 + -3) * (6 - -7 + -2 - -7)
3 * (13 + -2 + 7)
3 * 18
<answer>54</answer>
==

((8 * 9 * 0 - -1) - (-9 - -7 + -4 - 8)) =
1. 8 * 9 * 0 - -1 = 0 + 1 = 1
2. -9 - -7 + -4 - 8 = -9 + 7 - 4 - 8 = -14
3. 1 - (-14) = 1 + 14 = 15
<answer>15</answer>
```

Figure 2: An example prompt: instruction $I$ describes the task; exemplars $(e_1, ..., e_k, k = 1$ in the figure) provide demonstrations and enable ICL; both are prepended to the query $x$ before receiving the LLM responses.

**Instruction optimization (IO).** On black-box LLMs, the origin of IO may be traced to *discrete prompt search* [12, 40, 53, 62, 34] which prepend optimized tokens, which can be viewed as a form of "instructions", to inputs. However, these approaches do not necessarily yield interpretable prompts and most of them require output signals (such as logits) beyond strictly black-box access. Thus, recent advances have shifted towards utilizing an LLM itself to generate natural language instructions for iterative optimization on $\mathcal{D}_{\text{val}}$ in Eq. 1. The seminal works is APE [66], which employs the LLM to iteratively cull top-performing instructions on $\mathcal{D}_{\text{val}}$ and paraphrase them until convergence. Similar evolutionary frameworks are widely used in follow-up works [16, 19, 63] and alternative formulations like Bayesian optimization (BO) [7] and neural bandits [25] were also used. Another line of works [35, 48, 56, 42] employ *reflection*, directing an LLM to articulate reasons for errors to iteratively improve instructions. Other approaches like OPRO and its variants [54, 27], treat the LLM as a black-box optimizer, tasking it with generating new instructions based on the trajectory of previously evaluated instructions and their performances without explicit meta-instructions.

**Combining EO and IO.** As discussed in §1, there is a relative dearth of work combining EO and IO despite their shared objective. Specifically, even when the labeled dataset $\mathcal{D}_{\text{val}}$ is a prerequisite of virtually all IO methods, it is primarily used to estimate the expectation in Eq. 1 only rather than to construct exemplars in a principled way: For instance, ProTeGi [35] randomly samples exemplars from $\mathcal{D}_{\text{val}}$, while OPRO [54] uses them only for instruction induction [18]. Other works [16, 66, 48] either use no exemplars or fix exemplars and only optimize the instructions – for challenging reasoning tasks, these methods require human-annotated chain-of-thought (CoT) exemplars [50] *in addition to* $\mathcal{D}_{\text{val}}$, which arguably runs counter to the goal of automatically designing prompts *without* human intervention in APO. A few exceptions exist: PromptBreeder [14] employs "context shuffling" to co-evolve exemplars and instructions, while Mixture-of-Prompts (MoP) [47] aligns exemplars with multiple prompting "experts" for joint optimization. However, these works still focus their optimization effort on instructions: PromptBreeder emphasizes complex mutation operators for IO while providing only basic EO frameworks, whereas MoP chiefly focuses on IO with the bulk of its contribution being assigning optimized instructions to different exemplar groups, rather

---

[2]i.e., only textual outputs are available; parameters, gradients, and intermediate outputs like logits are neither modifiable or accessible – as of June 2024, many SoTA models like Gemini [15, 36] and the most advanced variants of GPT-4 [1] fall into this category.

than optimizing the exemplars themselves. Other works [13, 57] also include both exemplars and instructions in the search space, but they require information beyond strictly black-box outputs to some extent. Lastly, several works have analyzed the interplay between ICL and instructions [30] or prompt templates [39], but they mainly characterize the performance variation as an *issue* deserving attention. We, however, consider the APO setup specifically, and argue that such an interdependence presents an *opportunity* through holistically considering instructions and exemplars. Concurrent to our work, Agarwal et al. [2] and Opsahl-Ong et al. [32] also study the joint optimization of instructions and exemplars, and in many cases reached conclusions corroborating our findings, demonstrating the community's growing awareness on the importance of the subject of focal interest to this paper.

## 3 Understanding Instruction Optimization and Exemplar Optimization

While studying IO and EO independently has academic value, the practical goal ultimately for both is to optimize the performance of LLMs. Hence, IO and EO, as two dominant genres of APO methods, present practitioners with the challenge of selecting or combining them to maximize cost-performance benefits. We aim to meet this by evaluating EO and IO in the context of APO by answering the following: **1)** What is the relative importance and performance impact of EO and IO, both in isolation and when combined together? **2**) How do we make the optimal use of the limited data and computational budget under the current APO framework?

### 3.1 Experimental Setup

We perform thorough experiments employing various EO and IO methods individually and in combination. We use the PaLM 2 `text-bison-002` [4] and Gemini 1.0 Pro/1.5 Flash [15, 36] as the target models, but we will also validate key findings on GPT-3.5. Modern IO methods often employ another, usually more potent *optimizer model* for to generate and/or critique instructions; we use PaLM 2 `text-unicorn-001` (for `text-bison` target model), Gemini 1.0 Ultra (for Gemini 1.0 Pro target model) or Gemini 1.5 Pro (for Gemini 1.5 Flash target model). We evaluate on tasks selected from BIG-Bench Hard (BBH) [43], a collection of diverse tasks considered to be challenging to LLMs – the suite itself and datasets of similar task types are frequently used in many recent APO works [48, 54, 19, 14, *inter alia*]: the tasks include numerical reasoning, commonsense problem-solving, logical deduction, linguistic manipulation, machine translation, and tabular reasoning, among others. For all tasks, we use 20% of data as validation set and the remaining 80% for testing, the latter of which is held-out and unavailable to the LLM at search time (see App. A for implementation details). We also test some of our key findings on the MMLU benchmark [17], a set of 57 tasks frequently used to gauge the general problem-solving abilities of LLMs – we use the official `val` and `test` splits for validation and testing, respectively. We consider the following IO strategies:

1. **No IO**: we use the seed instruction $I_0$ "*Let's think step by step*." [23] without any optimization.
2. **APE** [66] is the seminal work for LLM-as-an-instruction-optimizer and uses an evolutionary algorithm design: at each iteration, we evaluate a population of instructions on the validation set and the optimizer is asked to generate a new population by paraphrasing the top-performing instructions. This process iterates until convergence.
3. **ProTeGi** [35] collects samples that the target LLM answers incorrectly on $\mathcal{D}_{\mathrm{val}}$ under the current instruction and directs the optimizer LLM to reflect and critique it. The optimizer model is then asked to update the instruction by summarizing and abstracting the feedback. Additionally, at each iteration, ProTeGi also paraphrases instructions similar to APE (referred to as "Monte Carlo sampling") and uses beam search to identify the most promising instructions.
4. **PromptAgent** [48] is similar to ProTeGi but it features a more advanced *planning agent* using Monte Carlo tree search [10].
5. **OPRO** [54] implicitly optimizes instructions. Starting from the seed instruction, at each iteration, OPRO provides the optimizer model a concatenation of previously evaluated instructions and their validation scores. Instead of explicitly requesting paraphrasing or reflection, OPRO treats the optimizer LLM as a black-box optimizer and simply asks the optimizer model to "come up with a better instruction" given these information.

The above methods are selected as each of them represents the state of the art of a genre of approaches as outlined in §2 and collectively represents IO techniques as a whole. We initialize each method at the seed instruction and ensure they consume the same amount of compute measured by the

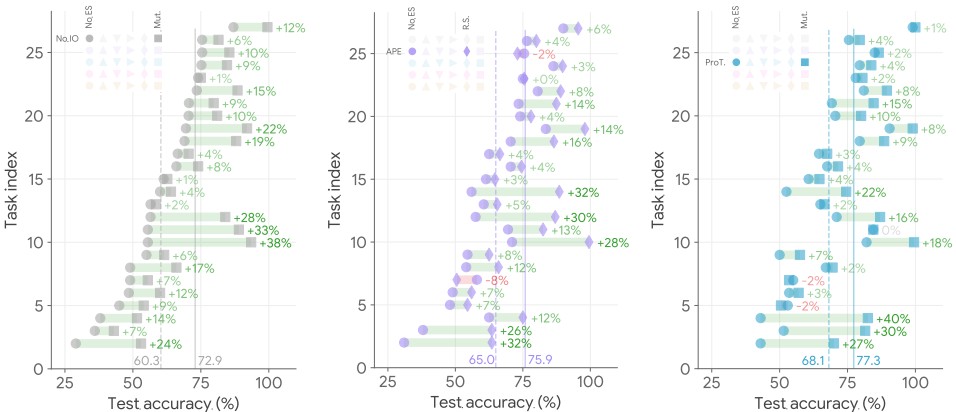

Figure 3: *Appropriate EO improves over any or no IO*: Task-specific BBH performance *with no instruction optimization* (**left**) and *with SoTA IO*: APE (**middle**) and ProTeGi (**right**) before and after applying exemplars found via Mutation (§3.1) on PaLM 2. Dashed and solid lines denote the average performance before and after exemplars, respectively. *Task index* is determined by the ascending order of test accuracy under seed instruction. Refer to additional visualization in App. B.3.

number of prompt evaluations on $\mathcal{D}_{\mathrm{val}}$ $m$ (we cap $m = 32$ except for "No IO" which requires no iteration). We also compare against PromptBreeder [14] in a later section, as it features a much more expansive search space and requires significantly more than 32 iterations before convergence. After obtaining the optimized instruction $I^*$ (or $I_0$ if no IO is performed), we perform EO. At this point, we emphasize that our setup should *not* be confused with the "few-shot" setup considered by some prior works [66, 16] which require additional human-annotated exemplars with reasoning traces to elicit CoT behavior. We perform EO only from the exemplars *self-generated* by the target model (also referred to as "bootstrapped few-shot" in DSPy [21] and "reinforced ICL" in concurrent works like Agarwal et al. [3]) and *do not assume exemplars are given at the start of APO* (i.e., we do not assume the presence of initial $\{e_1, ..., e_k\}$ in $P(x)$). We consider the following EO strategies:

1. **No EO**: no exemplars are used; this is typically referred to as "zero-shot" in the APO literature.
2. **Random**: we randomly sample $k$ input-output pairs from $\mathcal{D}_c(I^*) \subseteq \mathcal{D}_{\mathrm{val}}$, *the subset to the validation set that the target LLM predicted correctly under $I^*$ and the output in this case includes any intermediate output the LLM generates before the final answer.*
3. **Nearest**: We use the same $\mathcal{D}_c(I^*)$ as above, but instead of sampling randomly, we *retrieve* top-$k$ input-output pairs whose inputs are most similar to the current test input based on text embedding cosine similarity. We use the Gecko embedding [24].
4. **Diversity**: We use $\mathcal{D}_c(I^*)$ but select the $k$ input-output pairs closest to the centroids via $k$-means clustering, similar to the approach in Zhang et al. [60] to promote diversity in exemplars.
5. **All exemplars** (*Gemini 1.5 target models only*): With the advent of long-context models like Gemini 1.5, we may also fit the entire set of $\mathcal{D}_c$ into the context and perform no selection at all.

The above *heuristic*-based EO strategies do not use $\mathcal{D}_{\mathrm{val}}$, whereas *optimization*-based EO can utilize it similarly to IO. Instead of generating *instructions*, we select the *exemplar combinations* with the highest validation accuracy for testing [33, 20, 21]. Unlike IO, which creates *new* instructions via an optimizer model, EO selects from *pre-generated* outputs and does not require an optimizer model. Formally, we focus on optimizing exemplars conditional on $I^*$ from IO (or $I_0$ if no IO is involved)[3]:

$$E^* = \{e_j^*\}_{j=1}^k = \arg\max_{e_1,...,e_k \in \mathcal{E}} \mathbb{E}_{(x_i,y_i) \sim \mathcal{D}_{\mathrm{val}}}\Big[g\big(f_{\mathrm{LLM}}(I^*, \{e_j\}_{j=1}^k, x_i), y_i\big)\Big]$$
$$\text{s.t. } I^* = \arg\max_{I \in \mathcal{I}} \mathbb{E}_{(x_i,y_i) \sim \mathcal{D}_{\mathrm{val}}}\Big[g\big(f_{\mathrm{LLM}}(I, x_i), y_i\big)\Big]. \tag{2}$$

We include the following optimization-based EO methods that differ in search strategy:

6. **Random search**: Following the EO procedure in DSPy [21], we randomly sample $m$ combinations of $k$ exemplars: $\{E_1, ..., E_m\}$ where each $E_\ell = \{e_j^\ell\}_{j=1}^k \forall \ell \in \{1, ..., m\}$. We evaluate each combination on the validation set and use the best for testing.
7. **Mutation**: We also implement a mutation-based baseline, initiating with a population of $Q$ combinations for $T = m/Q$ generations, where $Q = 8$. Each generation starts with a randomly

---

[3]We performed EO *after* IO to optimize the exemplars generated by the best instruction; we also tested the *inverted* order (i.e., EO *before* IO) and *interleaved optimization* (detailed in App. B.8) and found the results to be largely robust to these design choices.

Table 1: Average BBH accuracy of all 30 EO-IO combinations with **PaLM 2** (`text-bison-002`) target model and **PaLM 2** (`text-unicorn-001`) optimizer model. The last row/column show the max improvement over the *No IO* and/or *No EO* baseline of the respective row/column. The background shades indicate cost in terms of # prompt evaluations on $\mathcal{D}_{\text{val}}$ by the *target model*: gray cells requires no evaluation on $\mathcal{D}_{\text{val}}$ ($m = 0$) ; blue cells perform $m = 32$ evaluations to iteratively optimize instructions *or* exemplars; orange cells iteratively optimize exemplars $m$ times on top of optimized instructions.

| | | Exemplar optimization (**EO**) | | | | | | Max Δ over *No EO* |
|---|---|---|---|---|---|---|---|---|
| | | *No EO* | Random | Nearest | Diversity | R.S. | Mutation | |
| Instruction Optimization (**IO**) | *No IO* | 60.30 | 66.91 | 66.09 | 66.74 | 71.16 | 72.92 | *+12.63* |
| | APE | 64.96 | 69.11 | 69.01 | 70.81 | 75.88 | 76.25 | *+11.28* |
| | ProTeGi | 68.13 | 70.81 | 70.01 | 69.25 | 75.90 | 77.29 | *+9.16* |
| | PromptAgent | 65.66 | 67.65 | 67.82 | 67.35 | 72.51 | 72.77 | *+7.11* |
| | OPRO | 63.04 | 68.50 | 68.33 | 67.57 | 73.02 | 73.06 | *+10.01* |
| | Max Δ over *No IO* | *+7.83* | *+3.89* | *+3.92* | *+4.07* | *+4.74* | *+4.37* | – |

Table 2: Average BBH accuracy of seed instruction (*No IO*) and ProTeGi (best IO strategy from Table 1) with different EO strategies using **Gemini 1.0 Pro** target model and **Gemini 1.0 Ultra** optimizer model. Refer to Table 1 for further explanations.

| | *No EO* | Random | Nearest | Diversity | R.S. | Mutation | Δ EO |
|---|---|---|---|---|---|---|---|
| *No IO* | 63.14 | 71.12 | 69.19 | 67.82 | 75.77 | 75.77 | *+12.63* |
| ProTeGi | 65.91 | 72.72 | 72.13 | 72.64 | 78.27 | 79.01 | *+13.10* |
| Δ IO | *+2.77* | *+1.60* | *+2.94* | *+4.83* | *+2.50* | *+2.52* | – |

Table 3: Average MMLU accuracy of *No IO* and ProTeGi with different EO strategies with `text-bison` target model and `text-unicorn` optimizer model. See App B.2 for Gemini results.

| | *No EO* | Random | R.S. | ΔEO |
|---|---|---|---|---|
| *No IO* | 65.77 | 72.06 | 72.75 | *+6.98* |
| ProTeGi | 69.73 | 70.82 | 72.31 | *+2.58* |
| Δ IO | *+3.96* | *-1.24* | *-0.44* | – |

Table 4: Average BBH accuracy of seed instruction (*No IO*), APE and ProTeGi (top 2 IO strategies from Table 1) with different EO strategies using **Gemini 1.5 Flash** target model and **Gemini 1.5 Pro** optimizer model. Refer to Table 1 for further explanations.

| | *No EO* | Random | Nearest | Diversity | All | R.S. | Mutation | ΔEO |
|---|---|---|---|---|---|---|---|---|
| *No IO* | 75.07 | 80.02 | 81.71 | 81.52 | 80.43 | 83.25 | 82.42 | *+8.18* |
| APE | 77.52 | 81.20 | 83.71 | 81.55 | 81.20 | 85.04 | 84.76 | *+7.54* |
| ProTeGi | 80.39 | 82.40 | 82.61 | 82.29 | 83.52 | 84.47 | 84.49 | *+4.10* |
| Δ IO | *+5.32* | *+2.20* | *+2.00* | *+0.77* | *+3.09* | *+1.79* | *+2.34* | – |

initialized first generation, similar to random search. For subsequent generations, we populate with $Q$ mutations of the best-performing combination from the previous generation, $E_{\leq t}^{*}$. Each mutation involves swapping one exemplar for another input-output pair from $\mathcal{D}_c$.

For all EO methods except for "*All exemplars*", we use $m = 32, k = 3$ for all main experiments, but we also test with $k = \{1, 3, 5, 10, 20\}$ in App. B.4. Given the IO and EO strategies, we experiment on each of the IO-EO combinations.

## 3.2 Results and Analyses

On BBH, we aggregate the results in Table 1 for PaLM 2 models. We also experiment on Gemini models with *no IO* and with ProTeGi (best overall IO technique from the PaLM 2 results) in Table 2 (Gemini 1.0) and Table 4 (Gemini 1.5). We also validate key findings in representative datasets on GPT-3.5 (Table 13, App B). On MMLU, we present results in Table 3 (PaLM 2) and App. B.1 (Gemini). Below, we highlight and discuss the key insights.

> **Insight 1**: We should almost *always* perform EO whenever we perform APO.

One of the immediate findings from the tables is that comparing the first column against others, *any* EO consistently improves test performance, with any or no instruction optimization. In Fig. 3, we further show that the EO not only benefits at an aggregated level but also leads to significant and almost unanimous improvements across diverse tasks.

While exemplars improving performance may not seem surprising, it is worth noting that as mentioned, in this case they are side-products of evaluating instructions on $\mathcal{D}_{\text{val}}$ with no additional data costs. Thus, we argue that for practical purposes under the framework of Eq. 1, barring unusual constraints

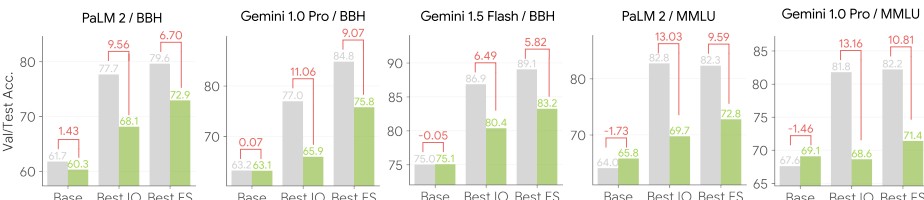

Figure 4: *Optimized exemplars generalize better than optimized instructions*. Comparison of validation accuracy and test accuracy over different model-task combinations. The generalization gap, which is the difference between validation and test accuracy, is marked on each figure. The better generalization of EO is exemplified by the smaller generalization gaps in all cases studied.

like extreme restrictions in context length, which This might restrict applicability of IO methods too as many SoTA IO methods also generate long prompts, and/or extreme long-context tasks where it is not possible to fit exemplars in the context window, *there is little incentive to consider the "zero-shot" setup without exemplars and little incentive not to perform EO*, given that current APO setup requires $\mathcal{D}_{\mathrm{val}}$ anyway, regardless of whether we use them as exemplars. Thus, it is by definition, not "zero-shot" and is not directly comparable to true zero-shot methods requiring no labeled data. Furthermore, there is also the risk that "zero-shot" results neither *reflect* nor accurately *predict* the full potential of the LLM, as what performs well under zero-shot does not necessarily performs well when a better EO strategy (e.g., PromptAgent in Table 1 and ProTeGi in Table 3) is used. Lastly, since obtaining labeled data can be costly, intelligently reusing them as exemplars also represents a more judicious use of scarce resources compared to only using them to evaluate a metric for optimization.

> **Insight 2**: How we select exemplars may outweigh how we optimize instructions, and selecting exemplars via *iterative optimization* consistently outperforms alternative strategies.

**Exemplar optimization outweighs instruction optimization.** Despite the recent focus the community places on IO, we find that how we select exemplars outweighs how we optimize instructions in the model-task combinations we investigate. With reference to Tables 1 − 4 (and task-specific breakdown in Fig. 5), we find that if we optimize instructions *or* exemplars (i.e., the blue cells) under a roughly compute-matched budget, *prompts without instruction optimization but with optimized exemplars* (e.g., the "No IO + Mutation" combination) *outperform prompts with SoTA instruction optimization but without optimized exemplars* (e.g., the "ProTeGi + Random" combinations) in an overwhelming majority of cases. In fact, on a separate set of experiments performed on the PaLM 2 models, we find this to be true *even after halving the evaluation budget of EO* (see App. B.6), and optimization on exemplars as naïve as random search can outperform IO methods that are significantly more complicated and expensive. Further substantiating this argument are that:

**1)** In *isolation*, EO boosts performance more effectively than IO: for example, with reference to Table 2, compared to the seed prompt, using the best EO strategy (Mutation, *first row*) alone increases the average performance by >11%, compared to approximately 8% using the best IO strategy (ProTeGi, *first column*);
**2)** When *combined*, benefits of EO and IO stack up but are largely attributable to EO: under "Mutation" (*last column*), the best EO strategy, the performance gap between the best and worst IO strategies shrinks to less than 4%, suggesting that instructions might be less critical if the LLM is provided with good exemplars after all.

We observe similar conclusions for different models and task combinations. In fact, on MMLU (Table 3) featuring much smaller validation splits, we observe that judicious exemplar optimization completely eliminates the performance gap caused by IO under zero-shot, with *No IO* even surpassing SoTA IO. Interestingly, as we show in Fig. 4 where we further consider the difference between validation accuracy, which is the empirical objective function in Eq. 1, and the test accuracy, which is the reported metric that represents the generalization power of the optimized prompt, *optimized exemplars generalize better than optimized instructions* under all model-task combinations considered. On MMLU tasks (two rightmost plots in Fig. 4), IO even improves validation performance comparable to or better than EO, but the validation improvement does not generalize to the test set. These imply that the superior test performance of EO cannot be solely attributed to a more effective search space $\mathcal{P}$ or optimization strategy in Eq. 1, and the performance gap might not be completely closed by advancing optimization only.

**Optimization-based EO outperforms heuristics.** Between the different EO strategies, we find that optimization-based EO vastly outperform the alternatives: e.g. in all tables, ProTeGi with optimized

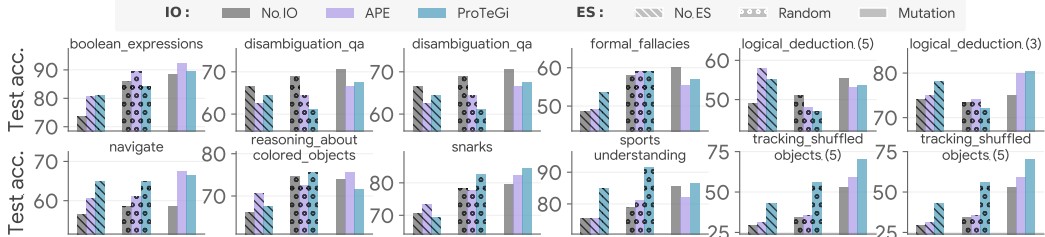

Figure 5: Task-specific BBH performance of selected IO-EO combinations with PaLM 2 (first 12 tasks; refer to App. B.3 for all other tasks/models). Note that **1)** Proper EO almost uniformly improves performance and **2)** With appropriate exemplars, seed instructions **with no optimization** (third bar from the right) can often perform on par or better than SoTA IO but with standard random exemplars or no exemplars commonly used in the literature (first six bars in each figure).

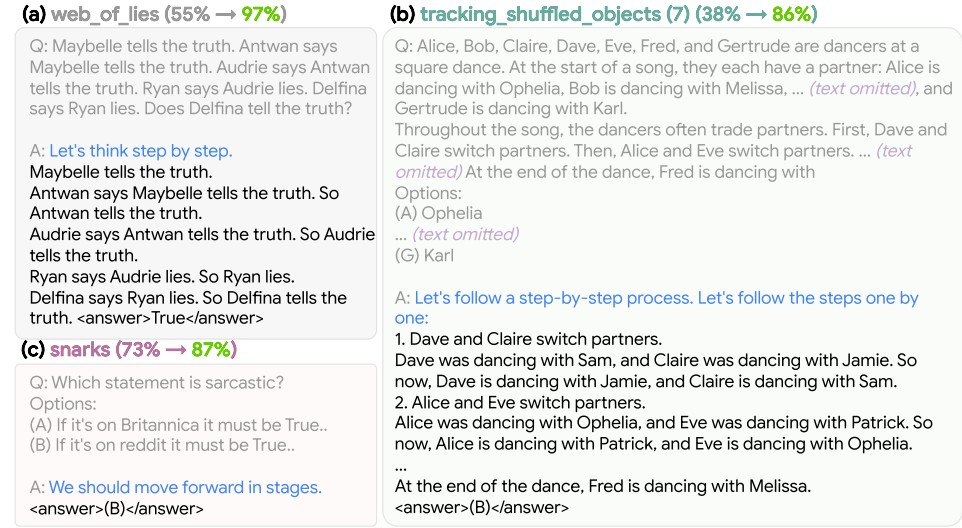

Figure 6: "Winning" exemplars that led to exceedingly high performance: **(a, b)** LLM improves by almost 50% from imitating and chaining the patterns in the optimal exemplars. **(c)** When CoT hurts performance, optimal exemplars encourage LLMs to override instructions and answer directly. Refer to App. B.9 for examples of LLM responses when these exemplars are applied.

exemplars outperforms random exemplars, which is the default design in Pryzant et al. [35], by more than 6% in both Table 1 and 2 and more than 2% in Table 4. Interestingly, as shown by the "All" column in Table 4 for Gemini 1.5 and App. B.4 for Gemini 1.0, naïvely scaling the number of exemplars may not be the most effective – the fact using the entire $\mathcal{D}_c$ underperforms 3 optimized exemplars, which are a subset of $\mathcal{D}_c$, highlights the relevance of EO even for modern LLMs capable of handling long context windows. On the other hand, heuristic-based selection like *Diversity* and *Nearest* do not consistently outperform simple random baseline, echoing previous findings [33].

**Imitation of task-dependent winning patterns outweighs elaborate descriptions.** We further present representative prompts in Fig 6 and how LLM responds to them in App. B.9: Generally, we find that even detailed and task-specific instructions do not regulate the LLM's behavior as effectively as exemplars, which enable *imitation*. For example, for challenging tasks like `tracking_shuffled_objects` and `web_of_lies`, optimization-based EO discover "winning" templates that, when followed, improve performance massively. Even when SoTA IO methods may often state the answer steps equivalently in words, we find LLMs to simply respond better to exemplars from which they can copy behaviors. On the other hand, for tasks where CoT-style answering are known to be unhelpful (e.g., `snarks` [43]), the optimized exemplars are invariably those giving direct answers; when prepended to the test queries, LLMs tend to prioritize exemplar imitation over instruction following to answer directly despite triggers like "We should move forward in stages". These highly task-dependent "winning" patterns that vary from elaborate step-to-step reasoning to direct answering may also explain why heuristic-based EO fares worse to data-driven approaches, since there might not be a single heuristic universally useful for all tasks.

**Concluding remarks.** We argue that the findings are highly significant for the future of APO. First, they point to a need of re-balancing: without disparaging the value of IO, we argue that EO is at least

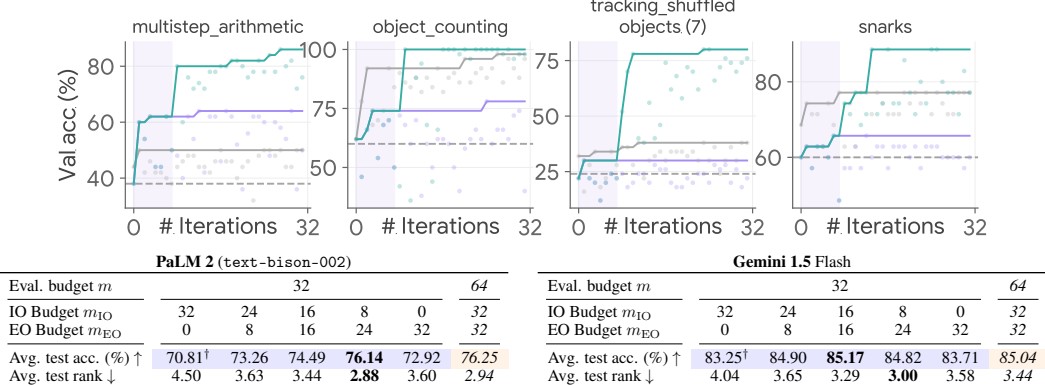

| | **PaLM 2** (text-bison-002) | | | | | | | **Gemini 1.5** Flash | | | | | |
|---|---|---|---|---|---|---|---|---|---|---|---|---|---|
| Eval. budget $m$ | | 32 | | | | *64* | Eval. budget $m$ | | 32 | | | | *64* |
| IO Budget $m_{IO}$ | 32 | 24 | 16 | 8 | 0 | *32* | IO Budget $m_{IO}$ | 32 | 24 | 16 | 8 | 0 | *32* |
| EO Budget $m_{EO}$ | 0 | 8 | 16 | 24 | 32 | *32* | EO Budget $m_{EO}$ | 0 | 8 | 16 | 24 | 32 | *32* |
| Avg. test acc. (%) ↑ | 70.81[†] | 73.26 | 74.49 | **76.14** | 72.92 | *76.25* | Avg. test acc. (%) ↑ | 83.25[†] | 84.90 | **85.17** | 84.82 | 83.71 | *85.04* |
| Avg. test rank ↓ | 4.50 | 3.63 | 3.44 | **2.88** | 3.60 | *2.94* | Avg. test rank ↓ | 4.04 | 3.65 | 3.29 | **3.00** | 3.58 | *3.44* |

Figure 7: *Mixing-and-matching EO and IO outperforms either alone under a similar budget.* **Top figure**: Validation accuracy vs. # evaluations on $\mathcal{D}_{val}$ with PaLM 2 in selected tasks if we optimize instructions only (via APE), exemplars only (Mutation), or both (first 8 evals for IO (purple shade) + remaining 24 for EO). Gray dashed lines denote the performance of $I_0$. **Bottom table**: Test accuracy averaged across all tasks for different IO/EO budget allocations for PaLM 2 and Gemini 1.5. [†]Used best APE results without EO that incur additional evaluations. Refer to App. B.7 for all per-task results and additional results on Gemini 1.0 Pro and other instruction optimizers.

equally crucial and should not be relegated to an afterthought. Second, we note that the EO strategies studied are in no way exhaustive. In fact, in contrast to the sophisticated search and instruction generation approaches adopted by IO methods, they can even be considered elementary. Yet, they deliver comparable or more significant improvements. Thus, we anticipate advanced methods that more effectively optimize exemplars would yield even greater enhancements: some techniques in IO may be adapted to EO with little-to-no modifications. For example, many recent advances in IO adopt an evolutionary search framework – while our "Mutation" baseline can be seen as an elementary version of it in EO, it should be also straightforward to use more advanced search strategies and operators or use techniques like LLM-generated paraphrasing on top of selected exemplars. Other search techniques, such as sample-efficient combinatorial BO [5, 44, 11, 65], can be uniquely suitable for the EO setup, which is itself a combinatorial optimization problem[4]. Furthermore, while we used a fixed set of exemplars (i.e., *task*-wise selection), it might also be fruitful to further explore in the direction of *instance*-wise selection [51, 37]. Lastly, as discussed, the presence of (often large) generalization gap also suggests the importance to consider generalization alongside optimization, which seems to be the chief focus thus far; it might be promising to investigate analogies of well-tested in classical machine learning like regularization and cross-validation in APO.

> **Insight 3**: Optimizing both instructions and exemplars is greater than the sum of its parts, *even under a comparable computational budget.*

For most of the results obtained, we note that iteratively optimizing both instructions *and* exemplars led to the best performance. This naturally leads us to answer the second research question: whereas experiments in Tables 1 and 2 expend additional cost by optimizing exemplars *on top of* the optimized instructions, we show that **1)** such a combinable benefit does not simply root from the additional compute and **2)** optimally mixing-and-matching IO and EO leads to significant performance improvement with negligible additional overhead.

Concretely, we budget a *total* $m = 32$ prompt evaluations on $\mathcal{D}_{val}$ where we use first $m_{IO}=\{0,8,16,24,32\}$ iterations optimizing instructions and the remaining $m_{EO}$ optimizing exemplars. We summarize the results in Fig. 7, where we find that **1)** *any* mix-and-match outperforms IO or EO only (i.e., $m_{IO}=0$ or 32), and **2)** the best allocation bridges the gap or almost bridges the gap compared to the combination that uses twice as many prompt evaluations (last column) – interestingly, in this case the optimal allocation also roughly reflects the relative contribution of IO and EO to the overall performance improvement in Table 1. We show in App. B.7 that the above findings hold for other target models and instruction optimizers, and we also give detailed examples and explanations of the mechanism leading to this synergy. Additionally, in App. B.10, we compare this simple routine against PromptBreeder [14], which is one of the few existing IO methods that supports EO via an

---

[4]Wu et al. [52], a concurrent work, precisely explored a combinatorial BO solution.

optional "context shuffling" routine. It, however, mutates the exemplars purely stochastically rather than optimizing from the validation metric. We show that despite the simplicity, our algorithm converges to comparable or better solutions while incurring a fraction of the PromptBreeder cost, which often requires hundreds of evaluations before convergence. Finally, we note that the presented way to combine optimization-based IO and EO is a proof of concept and room for future improvement can be vast and we experiment several other alternative ways to combine them in App. B.8, but we defer thorough investigations to a future work.

> **Insight 4**: SoTA IO methods may be inadvertently relying on exemplars already.

Beyond inspecting performance metrics only, we also examine the actual instructions and exemplars discovered. While detailed prompts are available in App. C, we highlight a key observation that adds a new dimension to our discussion: SoTA IO strategies may inadvertently utilize exemplars already. Despite IO methods not typically explicitly focusing on exemplars, we find them to frequently generate texts resembling exemplars within instructions through feedback and reflection processes. For instance, PromptBreeder employs "Lamarckian mutation" to reconstruct instructions from input-output pairs, while ProTeGi prompts the optimizer model to analyze target model errors. These operators, though varied, all involve taking actual validation exemplars as inputs to optimizer models. As exemplified by Fig. 8, while the original intention may have been to abstract *task-level* instructions, the model occasionally incorporates these exemplars verbatim in the instructions. Whereas these "quasi-exemplars" may seem unintentional, we observe that they are surprisingly common in high-performing instructions and often contribute more to the performance than the actual instructions themselves.

We argue that the findings here provide further evidence corroborating our insights obtained so far and our suggestions advocating explicit EO. Indeed, in contrast to explicit optimization of the exemplars, the aforementioned mechanism of exemplar discovery via IO is entirely opportunistic and, depending on interpretation, an unintentional artifact. For example, the quasi-exemplars in Fig. 8 almost certainly originate from the optimizer model in ProTeGi taking a convenient shortcut by incorporating a critique into the instruction verbatim (note the presence of traces like "Label: a Prediction: b" which suggests a previous mistake by the target model), which should *not* happen if the optimizer model perfectly executes the intended task of *abstracting* these critiques. Thus, we argue that instead of relying on the opportunistic exemplar generation via IO, explicitly optimizing for exemplars can be more preferable, as shown throughout this study.

Full prompt: 84.5%
Quasi-exemplars only: 80.5% (-4.5%)
Removing quasi-exemplars: 71.0% (-13.5%)

Let's address this systematically. To determine the correct order of adjectives in a sentence, we need to follow a specific order. The general order of adjectives is as follows:

1. Opinion adjectives
2. Size adjectives
…
8. Purpose adjectives
In case of any confusion, you can utilize the ROYAL mnemonic as a memory aid:
R - Relative size (big, small); O - Opinion (beautiful, ugly); Y - Age (young, old); A - Shape (square, round); L - Color (red, blue)
By applying this rule, we can ascertain the correct order of adjectives in a sentence.

Now, let's examine the examples provided.
## Example 1
Text: "Which sentence has the correct adjective order:
Options:
(A) small archaic square leather sock
(B) archaic small leather square sock"
Label: a Prediction: b
The correct answer is (A). The adjectives in this sentence are "small," "archaic," "square," and "leather." Referring to the ROYAL mnemonic, the correct order of adjectives is"size, opinion, shape, material." Therefore, the appropriate order in this sentence is "small, archaic, square, leather."... *(3 more examples omitted)*

Figure 8: *The best instructions might actually be exemplars*: Best instruction discovered by ProTeGi on `hyperbaton` where spontaneously discovered "quasi-exemplars" are highlighted. We also edit the instructions to either remove or retain the higlighted parts and find these quasi-exemplars, rather than the rest of the instruction, drive the performance. See App. B.11 for more examples.

## 4 Conclusion

We present comprehensive evaluations on the SoTA IO and EO methods, both individually and combined. We demonstrate that EO can be a potentially more crucial element in APO, revealing a beneficial synergy through joint optimization with IO. We also find that the high performance of SoTA IO methods can themselves be driven by implicit yet spontaneous exemplar discovery. Overall, we advocate for further research into EO, both as an independent approach and in conjunction with IO, even for highly-capable instruction-following modern models. One limitation is that although the tasks we consider are fairly diverse and findings on it are already of value given the widespread interest just on these tasks *only*, they are not exhaustive, omitting tasks like open-ended longer-form generation and metrics like safety & harmfulness which are important for responsible use of LLMs. As is the case for any inductive study deriving insights from experiments, there is also a possibility that the findings may not fully generalize to other tasks and/or models. Expanding to include these aspects would be important for future work.

## Acknowledgements

We thank all colleagues from Google Cloud AI Research for their feedback. We would also like to thank the anonymous NeurIPS reviewers and area chairs, whose valuable comments have helped to improve our paper.

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

# Appendix

## A Implementation Details

**Input prompt templates.** In this section, we outline the input prompt templates we used for all experiments for reproducibility. For seed instruction (i.e., *No IO*), APE and OPRO, we adopt the following template for all datasets.

```
1 Q: {{ QUERY_TEXT }}
2 {{ ANSWER_INSTRUCTION }}
3 A: {{ INSTRUCTION }} {{ llm() }}
```

In the template above, `QUERY_TEXT` denotes the input text; `INSTRUCTION` denotes the instruction to be added, which is the principal optimizable component of IO methods; `llm()` denotes the location where LLM is prompted to generate an output. `ANSWER_INSTRUCTION` is a special, task-specific sentence to ensure the LLM generates the final answer in a format that can be easily parsed. Specifically, for all multiple-choice questions-style tasks, it has the following content:

```
1 Show your final answer option bracketed between <answer> and <\answer>.
```

For all other tasks, the content is:

```
1 Show your final answer {{ TASK_SPECIFIC_CONSTRAINT }} bracketed between <answer> and <\
    answer>.
```

where the content of `TASK_SPECIFIC_CONSTRAINT` depends on the task:

- boolean_expressions: (True or False only)
- formal_fallacies: (valid or invalid only)
- navigate, sports_understanding, causal_judgement, web_of_lies: (yes or no only)
- word_sorting: (sorted words separated by spaces only)
- *all other tasks*: None (empty string).

At test time, the final answer is extracted with the capturing pattern `<answer>...<\answer>`. When exemplars are added, we use the following template:

```
1 Q: {{ DEMO_1_QUERY_TEXT }}
2 {{ ANSWER_INSTRUCTION }}
3 A: {{ INSTRUCTION }} {{ DEMO_1_OUTPUT }}
4 ==
5
6 Q: {{ DEMO_2_QUERY_TEXT }}
7 {{ ANSWER_INSTRUCTION }}
8 A: {{ INSTRUCTION }} {{ DEMO_2_OUTPUT }}
9 ==
10 ...
11
12 Q: {{ QUERY_TEXT }}
13 {{ ANSWER_INSTRUCTION }}
14 A: {{ INSTRUCTION }} {{ llm() }}
```

where `DEMO_{i}_OUTPUT` contains the entire response from the LLM to the corresponding input (not the final answer only). For ProTeGi and PromptAgent, we follow the templates used in the respective original papers with the format that puts the instruction and answer instruction in front of the test query:

```
1 {{ INSTRUCTION }} {{ ANSWER_INSTRUCTION }}
2
3 {{ QUERY_TEXT }}
4 {{ llm() }}
```

Accordingly, we modify the template with exemplars to:

```
1 {{ INSTRUCTION }}
2 {{ DEMO_1_QUERY_TEXT }}  {{ ANSWER_INSTRUCTION }}
3 {{ DEMO_1_OUTPUT }}
4 ==
5
6 {{ DEMO_2_QUERY_TEXT }}  {{ ANSWER_INSTRUCTION }}
7 {{ DEMO_2_OUTPUT }}
8 ==
9 ...
10
11 {{ QUERY_TEXT }}  {{ ANSWER_INSTRUCTION }}
12 {{ llm() }}
```

Noting that instruction is stated once at the beginning only rather than repeated at each exemplar, in consistency to the original styles adopted by these papers.

**Implementation details of IO methods.** In this section, we describe the implementation details of the IO and EO methods adopted. For all methods, we use greedy decoding (temperature = 0) for the PaLM 2 (`text-bison-002`) or Gemini target models. Whenever an optimizer model is used, we use temperature = 1.0, `top_k` = 40 and `top_p` = 0.8. For both PaLM 2 and Gemini models, we use the Google Cloud Vertex AI API available at https://cloud.google.com/vertex-ai.

- **APE**: We adapt the official implementation available at https://github.com/keirp/automatic_prompt_engineer. Instead of using "instruction induction" [18] from exemplars which is the primary initialization method introduced in the original paper in §3.1, *Forward Mode Generation* or *Reverse Mode Generation*, we opt for the third option, *Customized Prompts* where we initialize APE at the seed prompt "Let's think step by step" because **1)** this ensures fairness in comparison with other methods and **2)** we find initializing at the seed prompt actually leads to much better performance because it is well-known to induce step-by-step, CoT-style reasoning from the LLM. On the other hand, while the LLM induced initial instructions may describe the task better, the model often tends to utter the final answer without intermediate steps which we observe lead to much worse performance: on the BBH tasks selected for experimentation in this work, instruction induction using the meta-prompt provided by the APE paper only led to an average test accuracy of 56.7% on PaLM 2 (`text-bison-002`), which is even worse than using the seed prompt with no additional optimization. For APE, we use a population size of 8 and allow for 4 generations.
- **OPRO**: we adapt the official implementation available at https://github.com/google-deepmind/opro. At each optimization step, we follow the authors by asking the optimizer model to generate 8 candidate prompts and we budget for 4 steps.
- **ProTeGi**: We use the official implementation available at https://github.com/microsoft/LMOps. We set the initial prompt to the seed prompt, and we always use the entire validation set to generate the "gradients" (i.e., we use no mini-batching). We set the number of newly proposed instructions per optimization step to 8, where half of them come from "gradients" (i.e., new instructions generated by using the optimizer model to critique past mistakes made by the target model) and the other half come from "Monte Carlo samples" which are paraphrased/rewritten variants of the past prompts by the optimizer model. We again allow for 4 generations of mutations.
- **PromptAgent**: We use the official implementation at https://github.com/XinyuanWangCS/PromptAgent. We set the initial prompt to the seed prompt and use the default Monte Carlo Tree Search algorithm and set the number of maximum iterations to 32 to be consistent with the other methods described above.

**Computational Resources.** All experiments conducted in this work are accessible via public APIs where the underlying LLMs are hosted from the server side. There is no computational resource requirement on the client machine except that one can access Google Colab and run Python 3.10.

**Datasets.** As discussed briefly in §3.1, we rely on existing assets to perform experiments. Both the BIG-Bench Hard (BBH) dataset and the MMLU dataset we used are licensed under the MIT License (BBH: https://github.com/suzgunmirac/BIG-Bench-Hard/blob/main/LICENSE; MMLU: https://github.com/hendrycks/test/blob/master/LICENSE).

# B Additional Experimental Results

## B.1 Detailed Results on BBH

In Tables 5 to 10, we show the per-task performance breakdown using the PaLM 2 (`text-bison-002`) target model whose aggregated results are presented in Table 1. In Table 11 and Table 12, we show the results using the Gemini 1.0 Pro and Gemini 1.5 Flash target models whose aggregated results are presented in Table 2 and Table 4, respectively. The aggregated and task-specific breakdown results on the GPT-3.5 (`gpt-3.5-turbo-0125`) model are shown in Tables 13 and 14, respectively.

Table 5: Per-task test accuracy (%) of the **PaLM-2** (`text-bison-002`) target model without exemplar optimization (**No EO**).

| EO method | **No EO** | | | | |
|---|---|---|---|---|---|
| IO method | *No IO* | OPRO | APE | ProTeGi | PromptAgent |
| boolean_expressions | 73.50 | 80.00 | 80.50 | 81.00 | 77.00 |
| causal_judgement | 61.33 | 58.67 | 61.33 | 60.67 | 65.33 |
| date_understanding | 70.50 | 67.00 | 74.00 | 70.50 | 66.00 |
| disambiguation_qa | 66.50 | 68.50 | 62.50 | 64.50 | 69.50 |
| formal_fallacies | 48.50 | 49.50 | 49.00 | 53.50 | 50.50 |
| geometric_shapes | 60.00 | 59.00 | 56.00 | 52.50 | 46.00 |
| hyperbaton | 55.50 | 79.00 | 69.50 | 84.50 | 86.50 |
| logical_deduction_five_objects | 49.00 | 50.50 | 58.00 | 55.00 | 52.50 |
| logical_deduction_seven_objects | 45.00 | 48.50 | 48.00 | 53.00 | 54.50 |
| logical_deduction_three_objects | 74.00 | 70.00 | 75.00 | 78.00 | 76.50 |
| movie_recommendation | 56.50 | 67.00 | 57.50 | 71.00 | 82.00 |
| multistep_arithmetic_two | 38.00 | 61.50 | 62.50 | 43.00 | 50.00 |
| navigate | 56.50 | 56.00 | 60.50 | 65.00 | 63.00 |
| object_counting | 69.50 | 81.50 | 83.50 | 90.50 | 81.50 |
| penguins_in_a_table | 75.21 | 80.34 | 86.32 | 79.49 | 78.63 |
| reasoning_about_colored_objects | 66.00 | 67.50 | 70.50 | 67.50 | 65.50 |
| ruin_names | 69.00 | 73.50 | 70.50 | 79.50 | 74.00 |
| salient_translation_error_detection | 55.00 | 53.50 | 54.50 | 50.00 | 53.00 |
| snarks | 70.63 | 70.63 | 73.43 | 69.23 | 81.12 |
| sports_understanding | 75.50 | 76.00 | 75.50 | 85.00 | 73.50 |
| temporal_sequences | 87.00 | 86.50 | 90.00 | 99.00 | 90.00 |
| tracking_shuffled_objects_five_objects | 29.00 | 40.50 | 31.00 | 43.00 | 38.50 |
| tracking_shuffled_objects_seven_objects | 36.00 | 37.00 | 38.00 | 51.50 | 33.50 |
| tracking_shuffled_objects_three_objects | 49.00 | 30.50 | 54.00 | 67.00 | 56.50 |
| web_of_lies | 55.50 | 53.50 | 71.00 | 82.00 | 70.00 |
| word_sorting | 75.50 | 73.00 | 76.50 | 75.50 | 72.00 |
| *Average test accuracy (%)* ↑ | 60.30 | 63.04 | 64.96 | **68.13** | 65.66 |

Table 6: Per-task test accuracy (%) of the **PaLM-2** (`text-bison-002`) target model with random exemplar optimization (**Random**).

| EO method | **Random** | | | | |
|---|---|---|---|---|---|
| IO method | *No IO* | OPRO | APE | ProTeGi | PromptAgent |
| boolean_expressions | 86.00 | 88.00 | 89.50 | 84.00 | 86.50 |
| causal_judgement | 67.33 | 64.67 | 62.00 | 63.33 | 62.67 |
| date_understanding | 77.00 | 74.00 | 76.00 | 79.00 | 79.00 |
| disambiguation_qa | 69.00 | 68.50 | 64.50 | 61.00 | 72.50 |
| formal_fallacies | 58.00 | 56.50 | 59.00 | 59.00 | 58.00 |
| geometric_shapes | 57.00 | 57.00 | 60.50 | 50.00 | 46.50 |
| hyperbaton | 81.50 | 71.00 | 72.00 | 77.00 | 82.50 |
| logical_deduction_five_objects | 51.00 | 50.00 | 48.00 | 47.00 | 45.00 |
| logical_deduction_seven_objects | 49.50 | 54.00 | 35.00 | 42.00 | 52.00 |
| logical_deduction_three_objects | 73.50 | 70.50 | 74.00 | 72.00 | 73.00 |
| movie_recommendation | 76.50 | 77.50 | 80.50 | 84.00 | 94.00 |
| multistep_arithmetic_two | 49.50 | 76.50 | 71.00 | 53.50 | 53.00 |
| navigate | 58.50 | 61.00 | 61.00 | 65.00 | 54.00 |
| object_counting | 78.00 | 85.50 | 92.00 | 97.50 | 97.50 |
| penguins_in_a_table | 76.07 | 82.91 | 77.78 | 84.62 | 78.63 |
| reasoning_about_colored_objects | 74.50 | 75.50 | 72.50 | 75.50 | 72.00 |
| ruin_names | 81.50 | 83.00 | 78.50 | 89.00 | 86.00 |
| salient_translation_error_detection | 61.50 | 60.50 | 60.50 | 56.50 | 58.00 |
| snarks | 78.32 | 79.02 | 77.62 | 82.52 | 82.52 |
| sports_understanding | 79.00 | 82.00 | 81.00 | 91.50 | 88.50 |
| temporal_sequences | 94.00 | 95.50 | 91.00 | 94.00 | 92.00 |
| tracking_shuffled_objects_five_objects | 34.00 | 45.50 | 35.50 | 56.00 | 23.50 |
| tracking_shuffled_objects_seven_objects | 33.00 | 41.00 | 55.50 | 38.50 | 43.50 |
| tracking_shuffled_objects_three_objects | 52.50 | 33.00 | 58.50 | 63.00 | 56.00 |
| web_of_lies | 65.50 | 70.50 | 86.00 | 97.50 | 47.50 |
| word_sorting | 77.50 | 78.00 | 77.50 | 78.00 | 74.50 |
| *Average test accuracy (%)* ↑ | 66.91 | 68.50 | 69.11 | **70.81** | 67.65 |

Table 7: Per-task test accuracy (%) of the **PaLM-2** (`text-bison-002`) target model with nearest exemplar optimization (**Nearest**).

| EO method | Nearest | | | | |
| IO method | *No IO* | OPRO | APE | ProTeGi | PromptAgent |
|---|---|---|---|---|---|
| boolean_expressions | 86.50 | 86.50 | 87.50 | 76.50 | 77.50 |
| causal_judgement | 68.00 | 62.67 | 67.33 | 66.67 | 65.33 |
| date_understanding | 81.50 | 77.50 | 78.50 | 77.50 | 78.00 |
| disambiguation_qa | 62.50 | 71.50 | 60.00 | 59.00 | 68.50 |
| formal_fallacies | 58.00 | 60.50 | 58.00 | 56.50 | 54.50 |
| geometric_shapes | 66.00 | 62.00 | 63.00 | 54.50 | 57.50 |
| hyperbaton | 76.50 | 68.50 | 71.50 | 80.00 | 82.00 |
| logical_deduction_five_objects | 49.00 | 46.50 | 44.00 | 38.00 | 46.00 |
| logical_deduction_seven_objects | 41.50 | 48.50 | 36.50 | 43.00 | 44.00 |
| logical_deduction_three_objects | 68.50 | 72.50 | 71.50 | 72.50 | 76.00 |
| movie_recommendation | 80.00 | 81.00 | 81.50 | 86.50 | 90.00 |
| multistep_arithmetic_two | 51.50 | 71.00 | 70.50 | 63.00 | 51.00 |
| navigate | 51.00 | 56.50 | 51.00 | 51.00 | 48.00 |
| object_counting | 80.50 | 84.50 | 92.50 | 92.00 | 96.00 |
| penguins_in_a_table | 80.34 | 84.62 | 84.62 | 82.05 | 76.07 |
| reasoning_about_colored_objects | 73.50 | 75.50 | 77.00 | 69.00 | 71.50 |
| ruin_names | 77.00 | 84.50 | 80.00 | 86.50 | 78.00 |
| salient_translation_error_detection | 58.50 | 58.00 | 58.50 | 58.00 | 61.00 |
| snarks | 76.92 | 83.22 | 80.42 | 82.52 | 83.92 |
| sports_understanding | 82.50 | 83.00 | 82.00 | 89.50 | 83.00 |
| temporal_sequences | 93.50 | 94.50 | 85.50 | 97.50 | 94.00 |
| tracking_shuffled_objects_five_objects | 24.50 | 51.00 | 29.00 | 49.50 | 31.50 |
| tracking_shuffled_objects_seven_objects | 34.00 | 41.00 | 52.00 | 63.00 | 42.00 |
| tracking_shuffled_objects_three_objects | 56.50 | 28.00 | 59.00 | 64.00 | 55.00 |
| web_of_lies | 60.00 | 62.00 | 91.50 | 82.50 | 75.00 |
| word_sorting | 80.00 | 81.50 | 81.50 | 79.50 | 78.00 |
| *Average test accuracy (%)* ↑ | 66.09 | 68.33 | 69.01 | **70.01** | 67.82 |

Table 8: Per-task test accuracy (%) of the **PaLM-2** (`text-bison-002`) target model with diversity exemplar optimization (**Diversity**).

| EO method | Diversity | | | | |
| IO method | *No IO* | OPRO | APE | ProTeGi | PromptAgent |
|---|---|---|---|---|---|
| boolean_expressions | 82.00 | 93.00 | 88.50 | 86.50 | 84.00 |
| causal_judgement | 65.33 | 63.33 | 67.33 | 66.00 | 66.67 |
| date_understanding | 75.50 | 66.50 | 75.50 | 76.50 | 75.50 |
| disambiguation_qa | 68.50 | 69.50 | 65.50 | 60.50 | 69.00 |
| formal_fallacies | 56.00 | 56.50 | 55.50 | 55.50 | 54.50 |
| geometric_shapes | 53.00 | 59.50 | 57.00 | 56.50 | 41.00 |
| hyperbaton | 80.50 | 66.00 | 70.50 | 80.50 | 87.00 |
| logical_deduction_five_objects | 48.50 | 59.00 | 46.00 | 34.50 | 51.50 |
| logical_deduction_seven_objects | 57.50 | 58.50 | 38.00 | 46.00 | 50.00 |
| logical_deduction_three_objects | 74.00 | 71.50 | 75.00 | 72.00 | 73.50 |
| movie_recommendation | 72.00 | 76.00 | 79.00 | 84.00 | 88.50 |
| multistep_arithmetic_two | 55.00 | 72.00 | 72.00 | 64.50 | 54.00 |
| navigate | 50.00 | 63.50 | 64.00 | 60.50 | 49.50 |
| object_counting | 84.00 | 86.00 | 93.00 | 86.50 | 97.00 |
| penguins_in_a_table | 72.65 | 66.67 | 85.47 | 76.07 | 76.92 |
| reasoning_about_colored_objects | 68.50 | 72.50 | 80.50 | 69.50 | 74.50 |
| ruin_names | 81.50 | 82.50 | 81.50 | 82.50 | 83.50 |
| salient_translation_error_detection | 55.50 | 62.00 | 56.00 | 55.00 | 61.50 |
| snarks | 79.72 | 83.22 | 83.22 | 86.01 | 83.92 |
| sports_understanding | 83.50 | 79.00 | 83.00 | 90.00 | 73.50 |
| temporal_sequences | 92.50 | 89.50 | 96.50 | 99.00 | 92.50 |
| tracking_shuffled_objects_five_objects | 27.00 | 59.50 | 42.50 | 37.50 | 37.00 |
| tracking_shuffled_objects_seven_objects | 44.50 | 33.50 | 47.00 | 53.00 | 42.50 |
| tracking_shuffled_objects_three_objects | 68.00 | 33.50 | 62.50 | 60.50 | 57.00 |
| web_of_lies | 64.50 | 56.50 | 97.50 | 84.50 | 48.00 |
| word_sorting | 75.50 | 77.50 | 78.50 | 77.00 | 78.50 |
| *Average test accuracy (%)* ↑ | 66.74 | 67.57 | **70.81** | 69.25 | 67.35 |

Table 9: Per-task test accuracy (%) of the **PaLM-2** (`text-bison-002`) target model with random search exemplar optimization (**Random search**) with search budget $m = 32$.

| EO method | **Random Search** ($m = 32$) | | | | |
| IO method | *No IO* | OPRO | APE | ProTeGi | PromptAgent |
|---|---|---|---|---|---|
| boolean_expressions | 85.50 | 91.50 | 89.00 | 83.50 | 84.50 |
| causal_judgement | 68.67 | 68.67 | 64.67 | 63.33 | 64.67 |
| date_understanding | 74.00 | 79.00 | 78.00 | 80.00 | 81.00 |
| disambiguation_qa | 70.00 | 65.50 | 66.50 | 68.00 | 72.00 |
| formal_fallacies | 56.00 | 54.00 | 56.00 | 63.00 | 55.00 |
| geometric_shapes | 62.50 | 59.00 | 88.50 | 67.50 | 65.50 |
| hyperbaton | 80.00 | 83.00 | 82.50 | 85.00 | 90.50 |
| logical_deduction_five_objects | 59.50 | 62.50 | 50.50 | 47.50 | 51.00 |
| logical_deduction_seven_objects | 45.00 | 59.00 | 54.50 | 47.00 | 54.00 |
| logical_deduction_three_objects | 84.50 | 81.00 | 75.50 | 76.50 | 76.00 |
| movie_recommendation | 80.00 | 87.50 | 87.00 | 85.50 | 93.50 |
| multistep_arithmetic_two | 53.00 | 73.50 | 75.00 | 75.00 | 54.00 |
| navigate | 65.50 | 67.00 | 65.50 | 68.50 | 66.00 |
| object_counting | 94.50 | 99.00 | 98.00 | 98.50 | 98.00 |
| penguins_in_a_table | 82.91 | 87.18 | 89.74 | 82.91 | 83.76 |
| reasoning_about_colored_objects | 73.00 | 80.50 | 74.50 | 76.00 | 76.50 |
| ruin_names | 85.00 | 87.00 | 86.50 | 85.00 | 87.00 |
| salient_translation_error_detection | 56.00 | 54.50 | 62.50 | 56.50 | 58.00 |
| snarks | 82.52 | 83.22 | 87.41 | 84.62 | 87.41 |
| sports_understanding | 78.50 | 82.00 | 73.00 | 88.50 | 82.50 |
| temporal_sequences | 93.50 | 92.00 | 95.50 | 98.50 | 96.50 |
| tracking_shuffled_objects_five_objects | 49.50 | 49.00 | 63.50 | 70.00 | 43.50 |
| tracking_shuffled_objects_seven_objects | 40.00 | 48.50 | 63.50 | 70.50 | 39.50 |
| tracking_shuffled_objects_three_objects | 56.50 | 28.50 | 66.00 | 74.00 | 55.00 |
| web_of_lies | 97.50 | 97.00 | 99.50 | 98.00 | 95.00 |
| word_sorting | 76.50 | 79.00 | 80.00 | 80.00 | 75.00 |
| *Average test accuracy (%)* ↑ | 71.16 | 73.02 | 75.88 | **75.90** | 72.51 |

Table 10: Per-task test accuracy (%) of the **PaLM-2** (`text-bison-002`) target model with mutation exemplar optimization (**Mutation**) with search budget $m = 32$.

| EO method | **Mutation** ($m = 32$) | | | | |
| IO method | *No IO* | OPRO | APE | ProTeGi | PromptAgent |
|---|---|---|---|---|---|
| boolean_expressions | 88.50 | 95.00 | 92.50 | 89.50 | 88.50 |
| causal_judgement | 62.67 | 65.33 | 65.33 | 64.67 | 68.67 |
| date_understanding | 81.00 | 76.00 | 79.00 | 80.00 | 78.50 |
| disambiguation_qa | 70.50 | 69.00 | 66.50 | 67.50 | 71.50 |
| formal_fallacies | 60.00 | 50.50 | 55.50 | 57.00 | 57.50 |
| geometric_shapes | 64.00 | 64.00 | 86.00 | 74.50 | 54.50 |
| hyperbaton | 89.00 | 76.00 | 77.50 | 84.50 | 81.50 |
| logical_deduction_five_objects | 55.50 | 55.00 | 53.00 | 53.50 | 62.50 |
| logical_deduction_seven_objects | 54.00 | 62.50 | 51.50 | 50.50 | 51.00 |
| logical_deduction_three_objects | 75.00 | 84.00 | 80.00 | 80.50 | 78.50 |
| movie_recommendation | 84.00 | 86.00 | 89.00 | 87.00 | 95.00 |
| multistep_arithmetic_two | 51.50 | 75.00 | 75.00 | 82.50 | 55.50 |
| navigate | 58.50 | 67.50 | 67.50 | 66.50 | 68.50 |
| object_counting | 92.00 | 92.50 | 97.50 | 99.00 | 99.00 |
| penguins_in_a_table | 84.62 | 75.21 | 78.63 | 83.76 | 76.92 |
| reasoning_about_colored_objects | 74.00 | 74.00 | 75.50 | 71.50 | 76.00 |
| ruin_names | 88.00 | 91.00 | 87.50 | 88.50 | 83.00 |
| salient_translation_error_detection | 61.50 | 61.50 | 61.50 | 57.50 | 59.00 |
| snarks | 79.72 | 80.42 | 82.52 | 84.62 | 85.31 |
| sports_understanding | 85.50 | 83.50 | 82.00 | 86.50 | 81.00 |
| temporal_sequences | 99.50 | 98.00 | 95.00 | 100.00 | 95.00 |
| tracking_shuffled_objects_five_objects | 53.00 | 64.50 | 59.00 | 70.00 | 50.50 |
| tracking_shuffled_objects_seven_objects | 43.00 | 53.50 | 86.00 | 81.50 | 41.50 |
| tracking_shuffled_objects_three_objects | 66.00 | 26.00 | 63.50 | 69.50 | 58.00 |
| web_of_lies | 93.50 | 96.00 | 98.50 | 99.50 | 98.00 |
| word_sorting | 81.50 | 77.50 | 77.00 | 79.50 | 77.00 |
| *Average test accuracy (%)* ↑ | 72.92 | 73.06 | 76.25 | **77.29** | 72.77 |

Table 11: Per-task test accuracy (%) of the **Gemini 1.0 Pro** target model.

| EO method | No EO | | Random | | Nearest | | Diversity | | Random Search | | Mutation | |
| IO method | No IO | ProTeGi | No IO | ProTeGi | No IO | ProTeGi | No IO | ProTeGi | No IO | ProTeGi | No IO | ProTeGi |
|---|---|---|---|---|---|---|---|---|---|---|---|---|
| boolean_expressions | 81.50 | 82.50 | 88.00 | 88.50 | 90.50 | 87.50 | 90.50 | 87.00 | 91.00 | 91.50 | 87.50 | 92.50 |
| causal_judgement | 54.67 | 58.00 | 64.00 | 68.67 | 61.33 | 66.00 | 59.33 | 63.33 | 63.33 | 66.00 | 63.33 | 65.33 |
| date_understanding | 70.50 | 64.50 | 71.00 | 73.50 | 79.00 | 75.50 | 73.00 | 70.00 | 78.50 | 78.50 | 83.00 | 78.00 |
| disambiguation_qa | 53.50 | 66.50 | 58.00 | 68.00 | 63.00 | 56.00 | 67.50 | 69.00 | 69.00 | 70.50 | 67.00 | 73.50 |
| formal_fallacies | 56.50 | 54.00 | 60.50 | 59.00 | 52.00 | 58.00 | 57.00 | 61.50 | 59.50 | 57.00 | 57.50 | 54.00 |
| geometric_shapes | 36.50 | 37.50 | 39.00 | 56.50 | 50.50 | 77.00 | 37.50 | 47.50 | 71.00 | 81.50 | 77.00 | 87.00 |
| hyperbaton | 75.00 | 72.00 | 79.00 | 63.00 | 76.00 | 73.00 | 83.00 | 78.00 | 87.00 | 91.00 | 85.00 | 87.50 |
| logical_deduction_five_objects | 52.00 | 53.00 | 44.50 | 48.50 | 48.50 | 51.50 | 59.50 | 52.50 | 51.00 | 49.00 | 64.00 | 57.00 |
| logical_deduction_seven_objects | 44.50 | 48.00 | 49.50 | 50.00 | 42.50 | 49.00 | 44.50 | 46.00 | 50.00 | 51.50 | 53.50 | 49.50 |
| logical_deduction_three_objects | 75.00 | 76.00 | 87.50 | 73.50 | 70.00 | 73.00 | 48.00 | 81.00 | 76.50 | 92.50 | 84.50 | 94.50 |
| movie_recommendation | 49.50 | 69.50 | 60.50 | 70.50 | 50.50 | 63.00 | 53.00 | 71.50 | 62.50 | 82.00 | 52.00 | 80.00 |
| multistep_arithmetic_two | 61.00 | 62.50 | 72.00 | 82.50 | 65.50 | 65.00 | 72.50 | 85.00 | 84.00 | 75.00 | 73.50 | 84.00 |
| navigate | 70.50 | 61.00 | 78.00 | 77.00 | 76.50 | 67.50 | 76.00 | 69.50 | 85.50 | 78.00 | 82.00 | 80.00 |
| object_counting | 74.50 | 79.00 | 89.00 | 88.50 | 83.00 | 90.50 | 93.50 | 92.00 | 95.50 | 93.50 | 92.00 | 95.50 |
| penguins_in_a_table | 87.18 | 86.32 | 84.62 | 83.76 | 83.76 | 82.91 | 81.20 | 79.49 | 84.62 | 87.18 | 82.05 | 79.49 |
| reasoning_about_colored_objects | 70.00 | 69.50 | 80.50 | 74.50 | 71.00 | 78.00 | 74.00 | 68.50 | 72.50 | 80.00 | 80.50 | 78.50 |
| ruin_names | 62.00 | 63.00 | 79.50 | 74.00 | 72.50 | 76.50 | 68.00 | 68.50 | 77.00 | 77.00 | 76.50 | 78.00 |
| salient_translation_error_detection | 50.50 | 46.00 | 53.00 | 59.00 | 52.00 | 60.50 | 60.50 | 62.00 | 58.00 | 58.50 | 58.00 | 58.50 |
| snarks | 81.82 | 76.92 | 82.52 | 79.72 | 83.92 | 76.92 | 82.52 | 80.42 | 86.01 | 83.22 | 86.01 | 83.92 |
| sports_understanding | 74.50 | 79.00 | 82.00 | 85.50 | 82.50 | 87.00 | 86.00 | 92.50 | 93.50 | 86.00 | 92.50 | 96.00 |
| temporal_sequences | 67.00 | 87.00 | 81.50 | 88.00 | 74.50 | 85.50 | 82.00 | 95.50 | 84.00 | 98.00 | 76.50 | 98.00 |
| tracking_shuffled_objects_five_objects | 54.00 | 57.50 | 75.50 | 78.00 | 65.00 | 71.50 | 55.50 | 66.00 | 71.50 | 80.00 | 75.50 | 80.00 |
| tracking_shuffled_objects_seven_objects | 50.00 | 60.00 | 44.00 | 49.00 | 64.50 | 62.50 | 43.00 | 55.50 | 71.00 | 71.50 | 70.00 | 77.00 |
| tracking_shuffled_objects_three_objects | 70.00 | 70.50 | 83.00 | 86.50 | 81.00 | 81.00 | 66.50 | 80.50 | 86.50 | 85.50 | 82.50 | 83.50 |
| web_of_lies | 56.50 | 74.50 | 94.50 | 97.50 | 88.00 | 90.00 | 77.50 | 97.00 | 97.50 | 99.50 | 97.00 | 98.50 |
| word_sorting | 63.00 | 59.50 | 68.00 | 68.50 | 71.50 | 70.50 | 68.00 | 69.00 | 67.50 | 71.00 | 71.00 | 64.50 |
| *Average test accuracy (%)* ↑ | 63.14 | 65.91 | 71.12 | 72.72 | 69.19 | 72.13 | 67.82 | 72.64 | 75.77 | 78.27 | 75.77 | 79.01 |

Table 12: Per-task test accuracy (%) of the **Gemini 1.5 Flash** target model.

| EO method | No EO | | | Random | | | Nearest | | | Diversity | | | All exemplars | | | Random Search | | | Mutation | | |
| IO method | No IO | APE | ProTeGi | No IO | APE | ProTeGi | No IO | APE | ProTeGi | No IO | APE | ProTeGi | No IO | APE | ProTeGi | No IO | APE | ProTeGi | No IO | APE | ProTeGi |
|---|---|---|---|---|---|---|---|---|---|---|---|---|---|---|---|---|---|---|---|---|---|
| boolean_expressions | 88.00 | 90.50 | 96.00 | 97.50 | 95.50 | 94.50 | 96.50 | 94.50 | 92.50 | 97.00 | 96.50 | 94.00 | 96.00 | 98.50 | 96.50 | 99.50 | 96.00 | 95.00 | 97.50 | 98.00 | 97.00 |
| causal_judgement | 62.00 | 68.00 | 61.33 | 65.33 | 66.67 | 64.67 | 61.33 | 64.67 | 64.00 | 64.00 | 68.00 | 64.00 | 62.00 | 64.67 | 67.33 | 67.33 | 67.33 | 66.00 | 62.00 | 68.00 | 62.67 |
| date_understanding | 76.50 | 80.00 | 79.50 | 85.00 | 85.00 | 85.00 | 87.50 | 87.00 | 87.50 | 86.00 | 88.00 | 82.50 | 84.50 | 91.00 | 88.50 | 84.50 | 84.00 | 87.00 | 85.00 | 86.50 | 87.00 |
| disambiguation_qa | 33.50 | 54.00 | 58.50 | 55.00 | 60.50 | 67.00 | 54.00 | 63.00 | 63.50 | 52.00 | 69.00 | 68.00 | 66.00 | 69.50 | 70.50 | 65.00 | 61.50 | 61.00 | 68.50 | 65.00 | 66.50 |
| formal_fallacies | 67.00 | 62.50 | 63.50 | 67.00 | 63.00 | 64.50 | 69.50 | 69.50 | 67.00 | 65.50 | 66.00 | 64.50 | 66.50 | 63.00 | 64.50 | 65.00 | 62.50 | 63.00 | 65.00 | 58.00 | 63.00 |
| geometric_shapes | 52.50 | 54.50 | 61.50 | 67.50 | 61.50 | 74.00 | 76.00 | 80.50 | 77.00 | 70.00 | 58.00 | 56.50 | 83.50 | 57.00 | 73.00 | 82.00 | 90.50 | 93.00 | 66.50 | 91.00 | 90.00 |
| hyperbaton | 84.50 | 87.00 | 88.00 | 86.00 | 80.50 | 82.50 | 89.00 | 88.50 | 86.50 | 91.00 | 87.50 | 89.00 | 92.00 | 88.00 | 91.50 | 91.00 | 92.50 | 94.50 | 92.50 | 90.00 | 93.50 |
| logical_deduction_five_objects | 60.50 | 73.00 | 71.50 | 71.50 | 72.00 | 76.00 | 77.00 | 78.50 | 72.50 | 75.00 | 67.50 | 73.50 | 70.00 | 70.00 | 70.00 | 76.50 | 78.50 | 75.00 | 81.50 | 72.50 | 80.00 |
| logical_deduction_seven_objects | 56.50 | 50.00 | 66.00 | 65.00 | 62.00 | 63.50 | 71.00 | 58.50 | 60.50 | 72.50 | 59.00 | 67.00 | 66.00 | 61.50 | 61.50 | 70.00 | 80.50 | 71.00 | 64.00 | 73.50 | 60.50 |
| logical_deduction_three_objects | 91.00 | 91.00 | 94.50 | 92.50 | 90.00 | 94.50 | 91.00 | 88.50 | 92.00 | 92.50 | 89.50 | 96.00 | 88.50 | 91.50 | 93.50 | 89.50 | 92.50 | 98.00 | 92.50 | 94.00 | 94.00 |
| movie_recommendation | 56.50 | 62.50 | 68.50 | 49.00 | 62.00 | 71.00 | 50.50 | 67.00 | 72.00 | 52.00 | 68.50 | 70.50 | 63.00 | 81.00 | 85.00 | 45.50 | 75.00 | 74.50 | 53.00 | 74.50 | 74.50 |
| multistep_arithmetic_two | 91.50 | 91.50 | 85.00 | 93.00 | 91.50 | 91.50 | 95.00 | 96.00 | 93.50 | 93.50 | 96.00 | 95.00 | 91.50 | 94.00 | 90.50 | 91.50 | 92.50 | 99.00 | 95.50 | 97.50 | 95.50 |
| navigate | 66.50 | 66.00 | 62.00 | 71.50 | 71.50 | 73.00 | 74.50 | 74.00 | 75.00 | 71.50 | 69.00 | 72.50 | 67.50 | 71.00 | 71.50 | 69.00 | 74.00 | 68.00 | 74.50 | 68.50 | 71.50 |
| object_counting | 87.50 | 86.50 | 88.00 | 89.00 | 90.00 | 90.50 | 92.00 | 94.00 | 91.00 | 91.50 | 89.50 | 89.50 | 95.50 | 96.50 | 92.50 | 95.50 | 93.50 | 90.50 | 92.50 | 95.50 | 91.50 |
| penguins_in_a_table | 93.16 | 95.73 | 92.31 | 93.16 | 96.58 | 98.29 | 96.58 | 99.15 | 95.73 | 92.31 | 97.44 | 97.44 | 93.16 | 97.44 | 96.58 | 94.87 | 95.73 | 94.02 | 95.73 | 94.02 | 96.58 |
| reasoning_about_colored_objects | 81.00 | 84.00 | 93.00 | 88.00 | 91.50 | 90.50 | 92.00 | 91.00 | 92.50 | 89.50 | 91.00 | 89.50 | 88.50 | 91.50 | 95.00 | 94.50 | 91.50 | 92.50 | 93.50 | 95.50 | 92.00 |
| ruin_names | 74.00 | 81.00 | 89.00 | 81.00 | 88.50 | 89.50 | 80.00 | 90.00 | 86.50 | 81.00 | 88.50 | 90.00 | 82.50 | 90.00 | 90.00 | 84.00 | 90.50 | 90.50 | 86.00 | 91.50 | 90.00 |
| salient_translation_error_detection | 58.00 | 61.50 | 65.00 | 60.00 | 64.00 | 59.50 | 65.50 | 66.00 | 63.00 | 67.50 | 68.00 | 58.00 | 64.50 | 65.50 | 64.50 | 63.50 | 60.50 | 60.50 | 59.50 | 62.00 | 61.00 |
| snarks | 77.62 | 71.33 | 82.52 | 83.92 | 82.52 | 85.31 | 83.92 | 83.22 | 86.01 | 86.71 | 74.83 | 83.22 | 87.41 | 79.02 | 84.62 | 87.41 | 83.92 | 81.82 | 88.81 | 79.72 | 86.01 |
| sports_understanding | 68.50 | 73.50 | 79.00 | 74.50 | 85.00 | 72.50 | 77.00 | 85.00 | 74.50 | 79.00 | 66.50 | 78.00 | 88.00 | 91.50 | 86.00 | 76.50 | 84.00 | 82.00 | 77.00 | 86.00 | 81.50 |
| temporal_sequences | 98.00 | 97.50 | 96.00 | 100.00 | 99.00 | 97.50 | 100.00 | 99.50 | 99.50 | 100.00 | 100.00 | 98.50 | 100.00 | 99.00 | 93.50 | 99.50 | 99.50 | 98.50 | 99.00 | 99.50 | 100.00 |
| tracking_shuffled_objects_five_objects | 93.00 | 90.50 | 93.50 | 95.50 | 96.50 | 96.00 | 94.00 | 97.50 | 95.00 | 97.50 | 97.50 | 97.50 | 93.00 | 93.00 | 92.50 | 98.00 | 97.50 | 96.00 | 96.00 | 99.00 | 99.00 |
| tracking_shuffled_objects_seven_objects | 91.50 | 92.50 | 96.00 | 93.50 | 95.50 | 98.50 | 96.00 | 98.00 | 99.00 | 96.50 | 98.50 | 98.50 | 87.50 | 88.50 | 92.00 | 100.00 | 99.50 | 98.50 | 99.00 | 99.50 | 99.00 |
| tracking_shuffled_objects_three_objects | 97.00 | 96.50 | 97.50 | 96.00 | 97.00 | 98.00 | 98.00 | 96.50 | 96.50 | 96.00 | 97.00 | 99.00 | 98.00 | 98.00 | 96.00 | 100.00 | 93.50 | 98.00 | 98.00 | 98.00 | 98.00 |
| web_of_lies | 93.00 | 94.50 | 100.00 | 99.00 | 99.50 | 100.00 | 100.00 | 98.50 | 100.00 | 100.00 | 100.00 | 100.00 | 52.00 | 54.00 | 98.50 | 100.00 | 100.00 | 100.00 | 100.00 | 100.00 | 100.00 |
| word_sorting | 53.00 | 60.00 | 62.50 | 61.00 | 64.00 | 64.50 | 59.50 | 68.00 | 65.00 | 53.50 | 68.50 | 67.50 | 64.00 | 66.50 | 66.00 | 74.50 | 66.50 | 68.50 | 60.00 | 66.50 | 66.50 |
| *Average test accuracy (%)* ↑ | 75.07 | 77.52 | 80.39 | 80.02 | 81.20 | 82.40 | 81.71 | 83.71 | 82.61 | 81.52 | 81.55 | 82.29 | 80.43 | 81.20 | 83.52 | 83.25 | **85.04** | 84.47 | 82.42 | _84.76_ | 84.49 |

Table 13: Average BBH accuracy (over 11 subtasks) of seed instruction (*No IO*), APE and ProTeGi with different EO strategies using **GPT-3.5** (gpt-3.5-turbo-0125) target model. Refer to Table 1 for further explanations.

| | *No EO* | Random | R.S. | Δ EO |
|---|---|---|---|---|
| *No IO* | 59.0 | 68.6 | 76.8 | *+17.8* |
| APE | 63.0 | 68.9 | 78.4 | *+15.4* |
| ProTeGi | 68.9 | 72.2 | 80.2 | *+11.3* |
| Δ IO | *+9.9* | *+3.6* | *+3.4* | – |

Table 14: Per-task test accuracy (%) of the **GPT-3.5** target model.

| EO method | No EO | | | Random | | | Random Search | | |
| IO method | No IO | APE | ProTeGi | No IO | APE | ProTeGi | No IO | APE | ProTeGi |
|---|---|---|---|---|---|---|---|---|---|
| boolean_expressions | 78.50 | 79.50 | 83.50 | 90.50 | 91.50 | 86.00 | 88.00 | 93.50 | 85.00 |
| causal_judgement | 52.00 | 54.67 | 54.67 | 59.33 | 55.33 | 62.67 | 66.00 | 63.33 | 62.67 |
| date_understanding | 71.50 | 73.50 | 75.00 | 79.50 | 78.50 | 78.50 | 75.50 | 75.00 | 80.00 |
| geometric_shapes | 31.00 | 33.00 | 52.50 | 33.50 | 35.00 | 42.50 | 62.50 | 65.00 | 72.50 |
| logical_deduction_seven_objects | 26.50 | 31.00 | 34.00 | 34.00 | 38.00 | 37.00 | 43.50 | 44.00 | 47.50 |
| logical_deduction_three_objects | 52.00 | 49.00 | 70.00 | 74.00 | 79.00 | 68.50 | 73.00 | 83.50 | 73.50 |
| multistep_arithmetic_two | 75.50 | 77.50 | 65.50 | 71.50 | 76.00 | 80.50 | 87.00 | 87.50 | 90.50 |
| navigate | 45.50 | 58.00 | 70.00 | 65.00 | 56.00 | 86.00 | 76.00 | 69.50 | 90.50 |
| object_counting | 76.50 | 78.50 | 79.50 | 76.00 | 74.50 | 83.00 | 96.00 | 94.00 | 94.50 |
| penguins_in_a_table | 76.92 | 78.63 | 87.18 | 84.62 | 88.03 | 88.89 | 87.18 | 88.03 | 89.74 |
| temporal_sequences | 63.00 | 79.50 | 86.00 | 86.50 | 86.00 | 81.00 | 90.00 | 99.50 | 95.50 |
| *Average test accuracy (%)* ↑ | 58.99 | 62.98 | 68.90 | 68.59 | 68.90 | 72.23 | 76.79 | 78.44 | 80.17 |

## B.2 Detailed Per-task Results on MMLU

In Table 15, we show the per-task performance breakdown using the PaLM 2 (text-bison-002) target model whose aggregated results are presented in Table 3. In Table 16, we show the MMLU results using the Gemini 1.0 Pro target model.

Table 15: Per-task test accuracy (%) of the **PaLM 2** target model on MMLU.

| EO method / IO method | No EO | | Random | | Random Search | |
|---|---|---|---|---|---|---|
| | No IO | ProTeGi | No IO | ProTeGi | No IO | ProTeGi |
| abstract_algebra | 41.00 | 44.00 | 43.00 | 42.00 | 40.00 | 44.00 |
| anatomy | 51.85 | 65.93 | 70.37 | 66.67 | 71.11 | 68.89 |
| astronomy | 74.34 | 75.00 | 81.58 | 80.92 | 83.55 | 81.58 |
| business_ethics | 62.00 | 65.00 | 68.00 | 72.00 | 70.00 | 67.00 |
| clinical_knowledge | 67.17 | 76.98 | 77.74 | 69.81 | 81.13 | 72.08 |
| college_biology | 71.53 | 76.39 | 83.33 | 80.56 | 84.03 | 82.64 |
| college_chemistry | 46.00 | 45.00 | 44.00 | 47.00 | 45.00 | 55.00 |
| college_computer_science | 57.00 | 59.00 | 63.00 | 56.00 | 61.00 | 63.00 |
| college_mathematics | 37.00 | 43.00 | 39.00 | 43.00 | 35.00 | 40.00 |
| college_medicine | 65.32 | 67.05 | 70.52 | 72.25 | 72.83 | 75.72 |
| college_physics | 48.04 | 59.80 | 57.84 | 57.84 | 58.82 | 60.78 |
| computer_security | 69.00 | 74.00 | 81.00 | 78.00 | 81.00 | 85.00 |
| conceptual_physics | 60.43 | 70.64 | 81.28 | 64.26 | 80.00 | 75.32 |
| econometrics | 52.63 | 48.25 | 53.51 | 57.89 | 54.39 | 53.51 |
| electrical_engineering | 66.90 | 66.90 | 75.86 | 68.28 | 71.72 | 73.10 |
| elementary_mathematics | 73.54 | 82.28 | 79.10 | 80.95 | 82.80 | 82.01 |
| formal_logic | 48.41 | 53.17 | 50.79 | 45.24 | 51.59 | 53.97 |
| global_facts | 58.00 | 46.00 | 61.00 | 47.00 | 66.00 | 48.00 |
| high_school_biology | 75.48 | 82.58 | 85.81 | 85.48 | 85.48 | 81.94 |
| high_school_chemistry | 55.17 | 65.02 | 65.52 | 58.62 | 64.04 | 59.61 |
| high_school_computer_science | 84.00 | 78.00 | 75.00 | 83.00 | 83.00 | 86.00 |
| high_school_european_history | 75.76 | 75.76 | 77.58 | 73.94 | 78.18 | 81.21 |
| high_school_geography | 71.21 | 83.33 | 88.38 | 88.38 | 90.40 | 86.87 |
| high_school_government_and_politics | 82.90 | 86.53 | 90.16 | 95.34 | 90.16 | 93.26 |
| high_school_macroeconomics | 67.44 | 73.33 | 74.36 | 58.21 | 72.82 | 73.08 |
| high_school_mathematics | 55.19 | 52.22 | 57.04 | 55.19 | 57.78 | 57.04 |
| high_school_microeconomics | 69.33 | 73.95 | 77.73 | 76.47 | 78.57 | 79.41 |
| high_school_physics | 53.64 | 49.67 | 44.37 | 50.33 | 44.37 | 54.97 |
| high_school_psychology | 79.27 | 88.81 | 90.28 | 86.97 | 87.89 | 90.46 |
| high_school_statistics | 63.43 | 62.50 | 68.06 | 65.74 | 68.52 | 68.98 |
| high_school_us_history | 75.98 | 79.41 | 78.43 | 87.75 | 79.41 | 86.27 |
| high_school_world_history | 77.22 | 77.64 | 80.59 | 80.59 | 81.43 | 73.84 |
| human_aging | 68.16 | 71.75 | 75.78 | 55.16 | 75.34 | 75.34 |
| human_sexuality | 72.52 | 82.44 | 82.44 | 81.68 | 81.68 | 80.15 |
| international_law | 76.03 | 74.38 | 76.86 | 73.55 | 80.17 | 81.82 |
| jurisprudence | 80.56 | 83.33 | 85.19 | 84.26 | 85.19 | 83.33 |
| logical_fallacies | 72.39 | 74.23 | 82.21 | 75.46 | 81.60 | 77.30 |
| machine_learning | 57.14 | 58.93 | 54.46 | 62.50 | 51.79 | 49.11 |
| management | 75.73 | 82.52 | 80.58 | 77.67 | 83.50 | 76.70 |
| marketing | 77.35 | 88.46 | 74.36 | 90.17 | 91.45 | 87.18 |
| medical_genetics | 62.00 | 79.00 | 76.00 | 86.00 | 81.00 | 87.00 |
| miscellaneous | 72.54 | 87.99 | 89.14 | 89.27 | 90.80 | 88.76 |
| moral_disputes | 67.63 | 73.70 | 74.86 | 79.19 | 75.72 | 77.17 |
| moral_scenarios | 51.62 | 51.51 | 61.45 | 57.88 | 71.06 | 65.47 |
| nutrition | 69.28 | 75.49 | 77.45 | 76.47 | 76.80 | 77.45 |
| philosophy | 65.59 | 75.88 | 78.78 | 75.24 | 79.74 | 76.53 |
| prehistory | 70.99 | 75.62 | 83.33 | 79.94 | 78.70 | 78.09 |
| professional_accounting | 57.45 | 60.28 | 60.28 | 56.74 | 60.28 | 58.87 |
| professional_law | 50.59 | 52.28 | 54.50 | 54.50 | 54.63 | 53.46 |
| professional_medicine | 72.79 | 72.79 | 76.84 | 77.21 | 78.68 | 77.94 |
| professional_psychology | 67.97 | 70.59 | 77.12 | 76.63 | 77.94 | 77.12 |
| public_relations | 61.82 | 63.64 | 56.36 | 60.00 | 57.27 | 60.91 |
| security_studies | 76.73 | 73.88 | 81.63 | 81.63 | 72.65 | 71.84 |
| sociology | 76.62 | 81.09 | 86.57 | 86.57 | 81.59 | 87.56 |
| us_foreign_policy | 81.00 | 81.00 | 85.00 | 84.00 | 84.00 | 85.00 |
| virology | 49.40 | 50.60 | 53.61 | 51.81 | 53.01 | 47.59 |
| world_religions | 78.95 | 85.96 | 88.30 | 87.72 | 90.06 | 85.38 |
| *Average test accuracy (%)* ↑ | 65.77 | 69.73 | 72.06 | 70.82 | **72.75** | 72.31 |

Table 16: Per-task test accuracy (%) of the **Gemini 1.0 Pro** target model on MMLU.

| EO method / IO method | No EO | | Random | | Random Search | |
|---|---|---|---|---|---|---|
| | No IO | ProTeGi | No IO | ProTeGi | No IO | ProTeGi |
| abstract_algebra | 42.00 | 42.00 | 42.00 | 51.00 | 46.00 | 43.00 |
| anatomy | 71.85 | 66.67 | 65.19 | 64.44 | 68.89 | 62.22 |
| astronomy | 78.29 | 74.34 | 82.89 | 77.63 | 79.61 | 80.26 |
| business_ethics | 65.00 | 65.00 | 65.00 | 71.00 | 67.00 | 71.00 |
| clinical_knowledge | 74.34 | 73.58 | 75.47 | 75.85 | 70.57 | 77.36 |
| college_biology | 80.56 | 81.25 | 86.81 | 86.11 | 83.33 | 86.81 |
| college_chemistry | 56.00 | 57.00 | 53.00 | 53.00 | 54.00 | 47.00 |
| college_computer_science | 56.00 | 54.00 | 56.00 | 60.00 | 52.00 | 59.00 |
| college_mathematics | 44.00 | 39.00 | 37.00 | 39.00 | 41.00 | 41.00 |
| college_medicine | 67.63 | 66.47 | 66.47 | 70.52 | 71.68 | 64.16 |
| college_physics | 63.73 | 71.57 | 54.90 | 66.67 | 58.82 | 62.75 |
| computer_security | 77.00 | 69.00 | 75.00 | 73.00 | 70.00 | 74.00 |
| conceptual_physics | 72.77 | 68.94 | 71.06 | 66.81 | 71.49 | 74.89 |
| econometrics | 48.25 | 46.49 | 44.74 | 54.39 | 50.00 | 58.77 |
| electrical_engineering | 61.38 | 63.45 | 67.59 | 64.97 | 66.90 | 67.59 |
| elementary_mathematics | 81.22 | 82.54 | 85.71 | 82.54 | 84.39 | 82.28 |
| formal_logic | 37.30 | 38.89 | 46.83 | 45.24 | 46.83 | 48.41 |
| global_facts | 48.00 | 57.00 | 56.00 | 57.00 | 56.00 | 54.00 |
| high_school_biology | 85.16 | 84.19 | 86.13 | 83.55 | 86.13 | 84.52 |
| high_school_chemistry | 62.07 | 60.10 | 66.01 | 60.10 | 69.46 | 58.13 |
| high_school_computer_science | 79.00 | 77.00 | 76.00 | 80.00 | 87.00 | 82.00 |
| high_school_european_history | 76.36 | 77.58 | 71.52 | 73.94 | 81.82 | 78.79 |
| high_school_geography | 81.31 | 79.80 | 85.35 | 88.89 | 84.85 | 87.88 |
| high_school_government_and_politics | 89.64 | 88.08 | 92.23 | 94.82 | 90.16 | 92.23 |
| high_school_macroeconomics | 74.87 | 75.90 | 72.82 | 63.08 | 80.00 | 74.62 |
| high_school_mathematics | 53.70 | 53.33 | 61.48 | 56.67 | 55.56 | 55.19 |
| high_school_microeconomics | 77.31 | 78.15 | 78.99 | 81.51 | 78.99 | 82.77 |
| high_school_physics | 58.94 | 58.94 | 54.97 | 45.03 | 62.91 | 54.30 |
| high_school_psychology | 86.79 | 83.49 | 84.77 | 87.89 | 87.34 | 86.79 |
| high_school_statistics | 66.20 | 62.96 | 64.81 | 64.35 | 68.52 | 67.59 |
| high_school_us_history | 81.37 | 83.33 | 82.84 | 86.27 | 82.84 | 86.76 |
| high_school_world_history | 83.12 | 77.64 | 86.08 | 83.54 | 83.54 | 82.28 |
| human_aging | 75.34 | 76.23 | 74.89 | 75.78 | 73.09 | 71.75 |
| human_sexuality | 70.99 | 79.39 | 77.86 | 81.68 | 75.57 | 79.39 |
| international_law | 77.69 | 74.38 | 80.99 | 81.82 | 76.86 | 78.51 |
| jurisprudence | 77.78 | 72.22 | 81.48 | 78.70 | 82.41 | 77.78 |
| logical_fallacies | 75.46 | 77.30 | 73.62 | 78.53 | 75.46 | 77.91 |
| machine_learning | 55.36 | 50.89 | 54.46 | 57.14 | 59.82 | 50.00 |
| management | 79.61 | 76.70 | 81.55 | 82.52 | 78.64 | 74.76 |
| marketing | 87.61 | 84.62 | 87.18 | 88.89 | 85.04 | 89.32 |
| medical_genetics | 75.00 | 68.00 | 73.00 | 79.00 | 81.00 | 76.00 |
| miscellaneous | 84.29 | 85.82 | 85.95 | 87.23 | 84.16 | 85.44 |
| moral_disputes | 67.05 | 65.61 | 70.81 | 77.46 | 73.70 | 74.86 |
| moral_scenarios | 44.36 | 50.95 | 53.85 | 65.36 | 57.21 | 64.13 |
| nutrition | 66.34 | 70.92 | 74.84 | 70.59 | 73.20 | 74.51 |
| philosophy | 67.85 | 69.13 | 71.38 | 76.53 | 72.03 | 76.21 |
| prehistory | 73.77 | 72.22 | 79.32 | 79.94 | 79.63 | 83.95 |
| professional_accounting | 60.64 | 54.26 | 60.99 | 55.32 | 58.16 | 54.26 |
| professional_law | 51.37 | 53.32 | 50.65 | 54.69 | 50.20 | 52.93 |
| professional_medicine | 76.10 | 73.16 | 76.84 | 72.43 | 76.10 | 71.69 |
| professional_psychology | 70.92 | 71.90 | 73.20 | 70.42 | 72.06 | 73.20 |
| public_relations | 60.91 | 60.00 | 62.73 | 65.45 | 62.73 | 59.09 |
| security_studies | 70.20 | 68.98 | 71.84 | 79.59 | 70.61 | 72.65 |
| sociology | 83.08 | 81.09 | 84.58 | 87.06 | 89.05 | 86.57 |
| us_foreign_policy | 76.00 | 84.00 | 80.00 | 85.00 | 85.00 | 87.00 |
| virology | 45.78 | 54.82 | 50.60 | 51.81 | 52.41 | 54.22 |
| world_religions | 81.87 | 78.36 | 79.53 | 83.04 | 87.13 | 83.63 |
| *Average test accuracy (%)* ↑ | 69.06 | 68.63 | 70.31 | **71.56** | 71.38 | 71.19 |

## B.3 Additional Visualization of Comparison Across IO and EO

Complementary to Fig. 3 in the main text, in PaLM 2 model, we show the per-task change of test accuracy before and after applying EO for APE and OPRO in Fig. 9 (comparison between *optimized exemplars* vs. *no exemplars*) and Fig. 10 (comparison between *optimized exemplars* vs *random exemplars*). We also include a visualization on the effect of IO (comparing optimized instructions and initial instructions) in Fig. 11. We also visualize the Gemini 1.0 Pro results in Fig. 12.

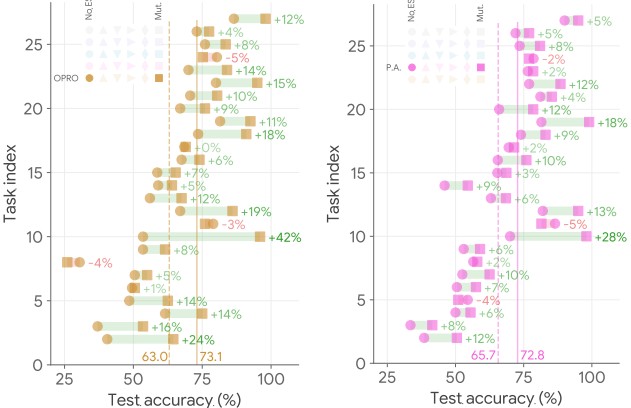

Figure 9: *Optimized exemplars vs. no exemplars in PaLM 2 model*: Comparison of performance of OPRO and PromptAgent (left to right) before (*no exemplars)* and after applying exemplars found via *Mutation*. Dashed and solid lines denote the average performance before and after exemplars, respectively. The results using APE, ProTeGi and no IO are shown in Fig. 3.

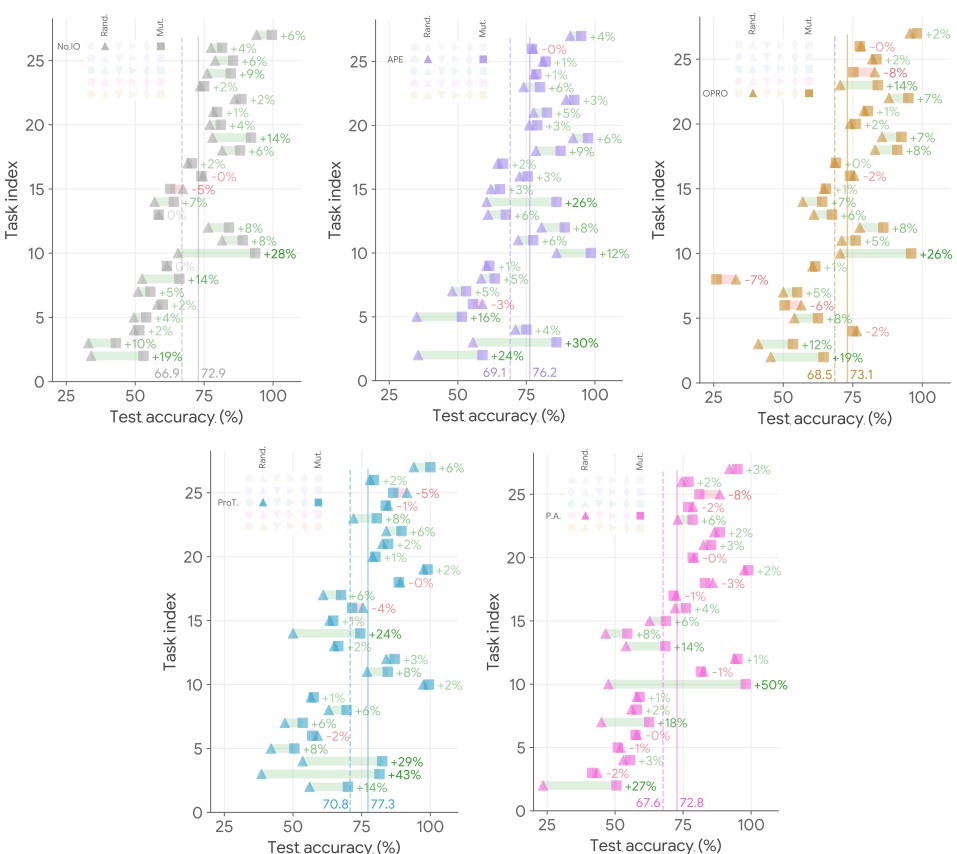

Figure 10: *Influence of using **optimized exemplars** compared to **random exemplars** in PaLM 2 model*: Comparison of performance of No IO, APE, OPRO, ProTeGi and PromptAgent (left to right and top to bottom) using *random exemplars* and *optimized exemplars* found via *Mutation*. Dashed and solid lines denote the average performance before and after exemplars, respectively. .

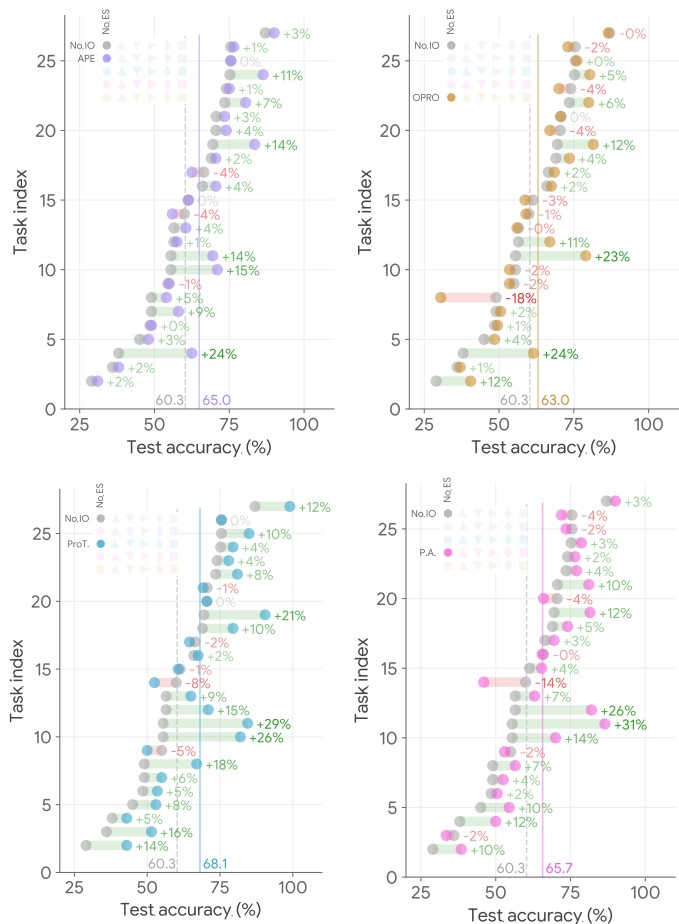

Figure 11: *Influence of using **optimized instructions** compared to **initial instructions** in PaLM 2 model*: Comparison of performance before and after using optimized instructions found via APE, OPRO (**Top row**), ProTeGi and PromptAgent (**Bottom row**). All results are without exemplars (i.e., "zero-shot" setup). Dashed and solid lines denote the average performance before and after examplers, respectively.

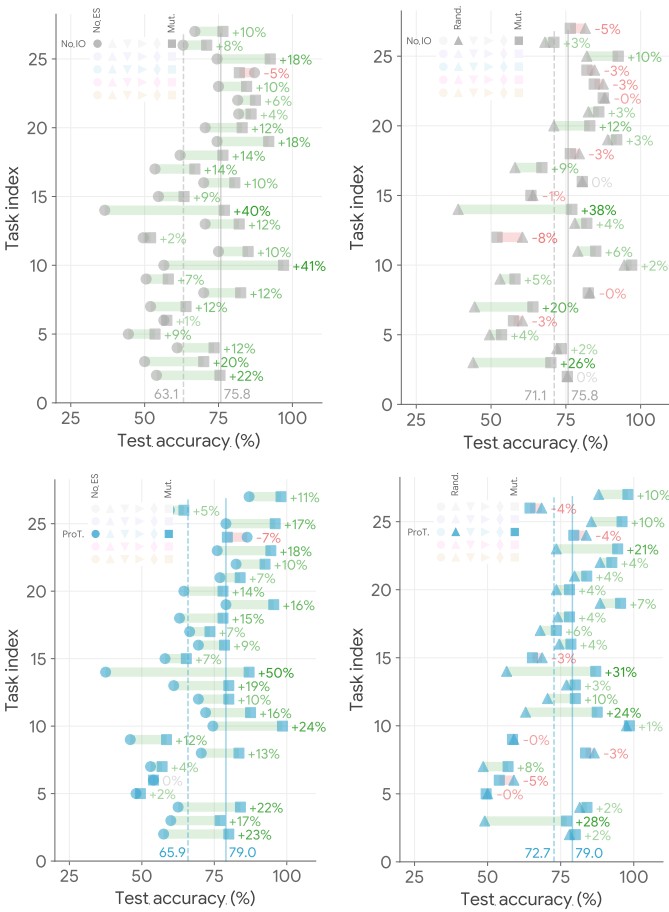

Figure 12: *Influence of using **optimized exemplars** compared to **no exemplars** and **random exemplars** in **Gemini 1.0 Pro** model*: Comparison of performance of No IO (top row) and ProTeGi (bottom row) using *no/random exemplars* vs *optimized exemplars* found via *Mutation*. Dashed and solid lines denote the average performance before and after exemplars, respectively.

Complementary to Fig. 5 in the main text, we visualize the additional BBH tasks on PaLM 2 in Fig. 13(a), Gemini 1.0 Pro in Fig. 13(b) and Gemini 1.5 Flash in Fig. 13. We also test key findings on a representative subset of 11 tasks in GPT-3.5 (Fig. 14. We visualize the MMLU results in Fig. 15.

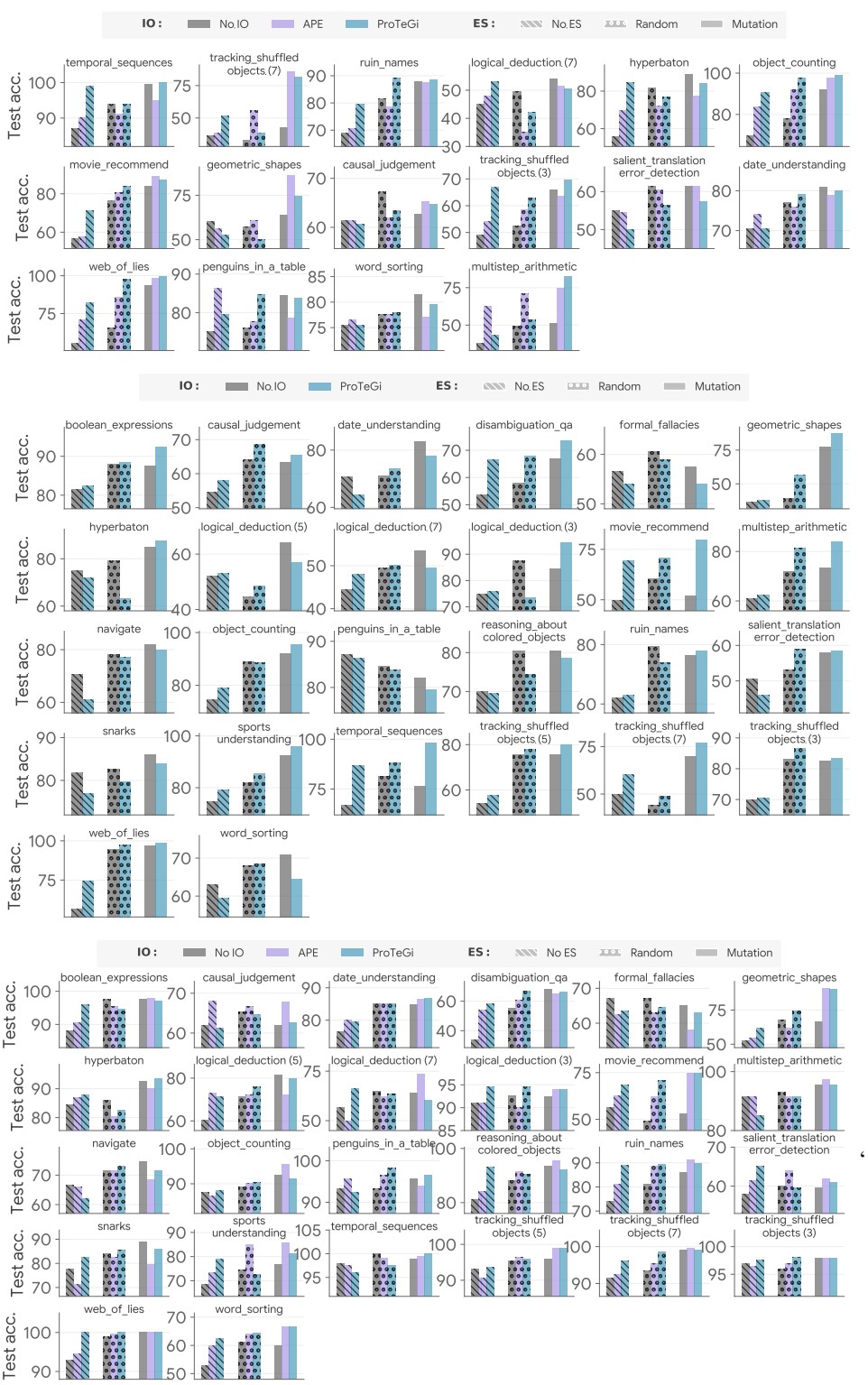

Figure 13: Visualization of per-task performance comparison on (first panel) **PaLM 2** (except the first 12 tasks, which are shown in Fig. 5), (middle panel) **Gemini 1.0 Pro** and (bottom panel) **Gemini 1.5 Flash**.

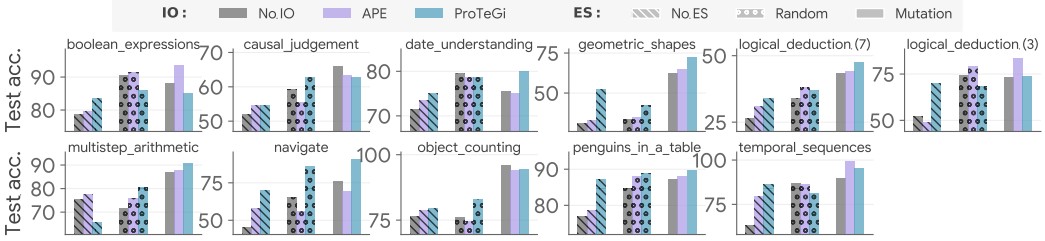

Figure 14: Visualization of per-task performance comparison on **GPT-3.5**.

Figure 15: Visualization of MMLU per-task performance comparison on PaLM 2.

## B.4 Varying Number of Shots

With the advent of LLMs that support longer context windows, an alternative approach to performing exemplar optimization is to scale the number of exemplars. Taking into the context of our setup, instead of selecting $k$ exemplars from $\mathcal{D}_c$ where $k$ is a small value, we can scale $k$ to a larger value or even use the entire $\mathcal{D}_c$ as exemplars. In this section, we perform experiments on the relative merits of the two approaches.

We show the aggregated results in Table 17 and task-specific results for both models in Fig. 16 and Fig. 17. While we find increasing the number of exemplars generally leads to improved test performance up to $k = 10$, further increase leads to performance deterioration. The increase in performance, however, does not diminish the importance of exemplar optimization as we show that using optimized exemplars outperform many more random exemplars, both at an aggregated level and on a task-specific level (the instances where optimized exemplars outperform random exemplars of any $k$, which are marked by magenta lines in Figs. 16 & 17, are more than 50% in both target models) – we argue that this suggests that exemplar optimization remains relevant even in modern LLMs with long contexts. In fact, we believe that novel exemplar search strategy in such many-shot setup can be an important next step, given that the number of possible combinations and permutations of exemplars explodes exponentially for a higher $k$, necessitating more advanced search strategies that would navigate this optimization landscape more effectively and efficiently. We defer a thorough investigation of this direction to a future work.

Table 17: *3 optimized exemplars outperform 20 random exemplars*: average performance across BBH tasks comparing different number of random exemplars per sample vs. optimized exemplars as presented in the paper.

| IO
EO
$k$ | No IO
Random | | | | | | No IO
Mutation | ProTeGi
Mutation |
|---|---|---|---|---|---|---|---|---|
| | 0 | 1 | 3 | 5 | 10 | 20 | 3 | 3 |
| PaLM 2 | 60.30 | 62.43 | 66.91 | 66.81 | 67.82 | 66.91 | 72.92 | 77.29 |
| Gemini 1.0 Pro | 63.14 | 68.65 | 71.12 | 71.41 | 71.94 | 69.17 | 75.77 | 79.01 |

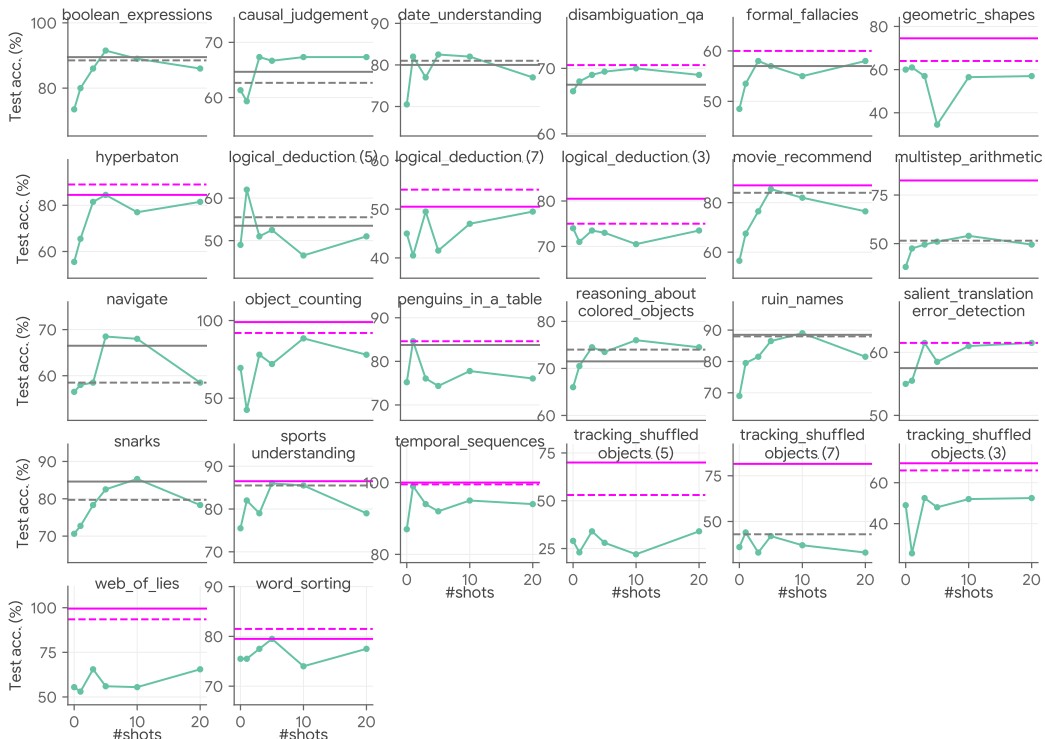

Figure 16: Per-task results on BBH tasks where we vary $k = \{0, 1, 3, 5, 10, 20\}$ for PaLM 2. The green curves denote the performance of "No IO + random exemplars" combination under different $k$ values; the dashed and solid lines correspond to the "No IO + Mutation" and "ProTeGi + Mutation" combinations under $k = 3$, respectively, and these lines are colored magenta when they outperform random exemplars of *any* $k$. $k = 0$ and $k = 3$ values are taken from Table 1 and 2 for PaLM 2 and Gemini models, respectively.

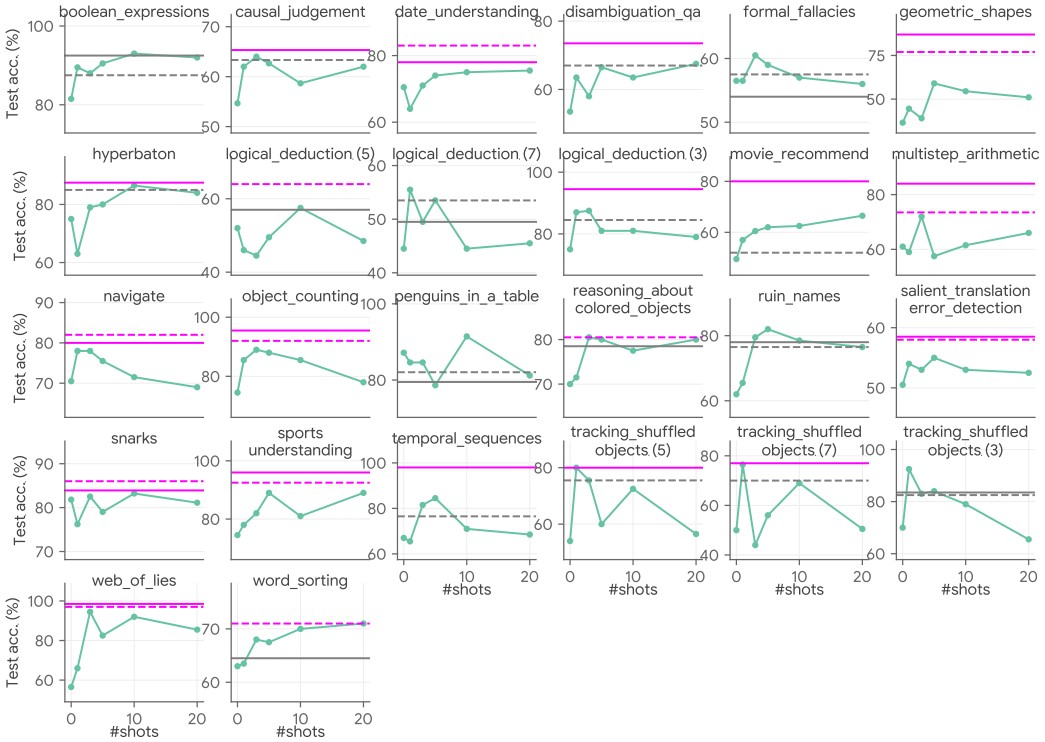

Figure 17: Per-task results on BBH tasks where we vary $k = \{0, 1, 3, 5, 10, 20\}$ for Gemini 1.0 Pro. Refer to Fig. 16 for additional explanations.

## B.5 Varying Validation Dataset Sizes

In the main experiments, we used 20%-80% validation-test split for all BBH tasks. In this section, we investigate the sensitivity of the various methods to changing validation data sizes, especially under very data-constrained setups such as very small validation data sizes. To do so, we vary the validation split to be $\{2, 4, 10, 20\}$% and we investigate on the combination of ProTeGi and Mutation, the best-performing IO and EO methods, respectively. We summarize the average test accuracy against the size of validation data split (as a fraction of the total data available) in Fig. 18.

We find that optimization-based EO is remarkably robust towards the size of the validation size, with the smallest performance drop even with an extremely small validation set (i.e., 2% or 4%, which correspond to 5 or 10 validation samples). On the other hand,

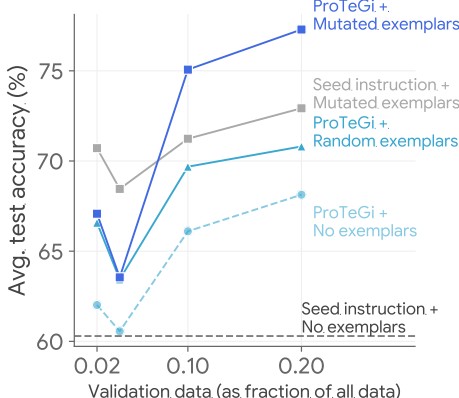

Figure 18: Average test accuracy with different validation split sizes with different IO-EO Combinations.

ProTeGi dropped to a performance no better than un-optimized seed instruction under such an extremely data-constrained setup, and even the performance of ProTeGi + Mutation, which performs the best overall in the main text dropped significantly with a small validation data size.

## B.6 EO with a Reduced Budget

In this section, we show the results of EO with search budget approximately halved (from $m = 32$ to $m = 16$) in Table 18 on top of the seed instructions without instruction optimization. We find that even after halving the budget (i.e., the EO strategies are now half as expensive), *seed instruction and*

*optimized exemplars still outperform optimized instruction and random exemplars* both in terms of average test accuracy and average rank.

Table 18: Per-task test accuracy (%) of the **PaLM-2** (`text-bison-002`) target model using EO of halved budget ($m = 16$). The last column (marked by †) is the result obtained by running ProTeGi (the best overall IO method from Table 1 with $m = 32$ with random exemplar optimization, as used originally in Pryzant et al. [35].

| Budget
EO method
IO method | $m = 16$
Random Search
*No IO* | Mutation
*No IO* | $m = 32^{\dagger}$
Random
ProTeGi |
|---|---|---|---|
| boolean_expressions | 89.00 | 85.50 | 84.00 |
| causal_judgement | 68.67 | 66.67 | 63.33 |
| date_understanding | 77.00 | 82.50 | 79.00 |
| disambiguation_qa | 71.50 | 69.50 | 61.00 |
| formal_fallacies | 56.50 | 57.00 | 59.00 |
| geometric_shapes | 59.50 | 64.00 | 50.00 |
| hyperbaton | 81.00 | 90.50 | 77.00 |
| logical_deduction_five_objects | 49.00 | 54.50 | 47.00 |
| logical_deduction_seven_objects | 50.50 | 51.00 | 42.00 |
| logical_deduction_three_objects | 82.00 | 73.50 | 72.00 |
| movie_recommendation | 84.00 | 83.50 | 84.00 |
| multistep_arithmetic_two | 50.00 | 48.50 | 53.50 |
| navigate | 65.00 | 70.50 | 65.00 |
| object_counting | 88.00 | 96.00 | 97.50 |
| penguins_in_a_table | 82.91 | 75.21 | 84.62 |
| reasoning_about_colored_objects | 79.00 | 71.50 | 75.50 |
| ruin_names | 87.50 | 85.50 | 89.00 |
| salient_translation_error_detection | 58.00 | 62.00 | 56.50 |
| snarks | 81.82 | 82.52 | 82.52 |
| sports_understanding | 83.00 | 80.00 | 91.50 |
| temporal_sequences | 100.00 | 91.00 | 94.00 |
| tracking_shuffled_objects_five_objects | 35.00 | 32.00 | 56.00 |
| tracking_shuffled_objects_seven_objects | 42.50 | 43.00 | 38.50 |
| tracking_shuffled_objects_three_objects | 51.50 | 70.50 | 63.00 |
| web_of_lies | 99.00 | 96.00 | 97.50 |
| word_sorting | 83.00 | 76.00 | 78.00 |
| *Average test accuracy (%)* ↑ | 71.34 | **71.48** | 70.81 |
| *Average rank* ↓ | **1.88** | 2.02 | 2.10 |

## B.7  Additional Results by Mixing-and-Matching IO and EO

**Other choices of instruction optimizers and/or target models.**  We show results from mixing-and-matching IO and EO in a similar manner to Fig. 7 but use ProTeGi (instead of APE in Fig. 7) on both PaLM 2 (`text-bison-002`) and Gemini 1.0 Pro target models in Table 19, where we see that two key findings (that *any* mix-and-match outperforms IO and EO only and that optimal allocation completely bridges the gap compared to a more expensive routine) we made still hold, except on Gemini model, the optimal allocation occurs at 16/16 instead of 8/24. This confirms that the phenomenon we see in Fig. 7 in the main text is not specific to the choice of the target models and/or instruction optimizers but is instead generally applicable.

Table 19: *Mixing-and-matching IO and EO leads to performance benefits under different IO strategies and/or target models.* Results on BBH tasks using two-stage IO-EO described in *Insight 3*, §3.2 but we instead use ProTeGi as the instruction optimizer on both PaLM 2 / Gemini 1.0 Pro target models. The color of the cells denotes the computational costs in terms of target model traversals on $\mathcal{D}_{\text{val}}$ consistent to Table 1.

| Method | ProTeGi + Mutation | | | | | | | | | |
|---|---|---|---|---|---|---|---|---|---|---|
| Model | PaLM 2 | | | | | Gemini 1.0 Pro | | | | |
| Budget for IO $m_{\text{IO}}$ | 32 | 16 | 8 | 0 | *32* | 32 | 16 | 8 | 0 | *32* |
| Budget for EO $m_{\text{EO}}$ | 0 | 16 | 24 | 32 | *32* | 0 | 16 | 24 | 32 | *32* |
| Avg. test accuracy (%) ↑ | 70.81 | 73.98 | **76.41** | 72.92 | *77.29* | 72.72 | **78.64** | 77.36 | 75.77 | *79.01* |
| Avg. rank ↓ | 3.77 | 3.15 | **2.52** | 3.17 | *2.38* | 3.85 | **2.44** | 2.79 | 3.35 | *2.58* |

**(a) IO only: Test acc: 62.5%**

```
Q: ((-4 + 5 - 4 - -6) * (-3 * -1 * 2 - -4)) =
Show your final answer bracketed between <answer> and
</answer>.
A: Let's carry out the task in a structured manner.
First, let's simplify the expression inside the
parentheses:

(-4 + 5 - 4 - -6) = (-4 + 5 - 4 + 6) = 5

Next, let's simplify the expression inside the second
set of parentheses:

(-3 * -1 * 2 - -4) = (3 * 2 + 4) = 10

Now, we can multiply the two simplified expressions:

5 * 10 = 50

Therefore, the final answer is <answer>50</answer>.
```

**(b) ES only: Test acc: 51.5%**

```
Q: ((5 - -2 * 7 + -2) - (0 * 9 * -6 - -7)) =
Show your final answer bracketed between <answer> and
</answer>.
A: Let's think step by step.  ((5 - -2 * 7 + -2) - (0
* 9 * -6 - -7)) =(5 - -14 + -2) - (0 - -7)(5 + 14 -
2) - (7)(19 - 2) - 717 - 7<answer>10</answer>
==

Q: ((-4 + 5 - 4 - -6) * (-3 * -1 * 2 - -4)) =
Show your final answer bracketed between <answer> and
</answer>.
A: Let's think step by step. ((-4 + 5 - 4 - -6) * (-3
* -1 * 2 - -4))=
(1 - 4 - -6) * (-3 * 2 + 4)
(-3 - -6) * (-6 + 4)
(-3 + 6) * -2
3 * -2 <answer>-6</answer>
```

**(c) IO + ES: Test acc: 75.0%**

```
Q: ((-3 + -4 * 4 + -8) + (1 - -3 + 4 - -9)) =
Show your final answer bracketed between <answer> and </answer>.
A: Let's carry out the task in a structured manner. **Step 1: Simplify the
expressions within the parentheses**

a) Simplify the first expression:
(-3 + -4 * 4 + -8) = (-3 + -16 + -8) = -27

b) Simplify the second expression:
(1 - -3 + 4 - -9) = (1 + 3 + 4 + 9) = 17

**Step 2: Substitute the simplified expressions back into the main equation**
((-3 + -4 * 4 + -8) + (1 - -3 + 4 - -9)) = (-27 + 17)

**Step 3: Perform the final calculation**
-27 + 17 = -10

**Final Answer:**
The final answer is <answer>-10</answer>.
==

Q: ((-4 + 5 - 4 - -6) * (-3 * -1 * 2 - -4)) =
Show your final answer bracketed between <answer> and </answer>.
A:  Let's carry out the task in a structured manner. **Step 1: Simplify the
expressions within the parentheses**

a) Simplify the first expression:
(-4 + 5 - 4 - (-6)) = (-4 + 5 - 4 + 6) = 3

b) Simplify the second expression:
(-3 * -1 * 2 - (-4)) = (3 * 2 + 4) = 10

**Step 2: Substitute the simplified expressions back into the main equation**
((-4 + 5 - 4 - (-6)) * (-3 * -1 * 2 - (-4))) = (3 * 10)

**Step 3: Perform the final calculation**
3 * 10 = 30

**Final Answer:**
The final answer is <answer>30</answer>.
```

Figure 19: An example of how IO and EO may synergize on `multistep_arithmetic_two` dataset using PaLM 2 target model.

**Mechanism of synergy between IO and EO.**    In this section, we give an example of why combined IO and EO outperform its constituting building blocks in Fig. 19: performing IO only (Fig. 19a) generates step-by-step text instructions, but without exemplars, the model is prone to making arithmetic errors. Performing EO only (Fig. 19b), on the other hand, subjects the model to the shortcomings of the un-optimized seed prompt, which, in this case, generates arithmetic operations-only outputs without textual reasoning. Combined IO and EO (Fig. 19c) addresses both issues, leading to vastly improved performance from both textual reasoning and exemplars.

## B.8   Alternative Ways of Combining IO and EO

As discussed in §3.2, we conducted EO after IO in the main text: in our setup, the exemplars are bootstrapped from the model's own predictions, which in turn depend on the instructions given. As such, we performed instruction optimization first on the zero-shot setup and search for the optimal exemplar(s) amongst the predictions induced by the optimal instruction. While we believe this order is the most natural, it is also important to ensure that any findings derived in this paper are not biased by the particular order we chose. Thus, in this section, we also experiment with two alternatives:

1. *Inverted order.* We first optimize the exemplars based on the seed instruction ("Let's think step by step." for most cases), and then freeze the exemplars and optimize the instruction on both Gemini 1.0 Pro and Gemini 1.5 Flash models. We used APE to optimize the instructions and random search to optimize exemplars wiht a total budget of $m = 32$, which is identical to the setup presented in Fig. 7 in the main text.

2. *Interleaved.* Instead of framing the exemplar and instruction optimization as a two-staged approach, a better approach in principle is to optimize them jointly. Here, we propose an *interleaved approach* with the intention of conditioning instructions to the best exemplars discovered so far and vice versa given their inter-dependence. The pseudocode representation of the algorithm in shown in Algorithm 1.

**Algorithm 1** Interleaved instructions and exemplars optimization. Exemplar optimization steps marked in blue.

1: **Inputs**: $\mathcal{D}_{\text{val}}$, target LLM, optimizer LLM, # generations $T$, instruction pop size $|\mathcal{I}|$, exemplar pop size $|\mathcal{E}|$, number of exemplar shots: $k$.
2: Initialize *instructions* $\mathcal{I}_0$ with seed instruction or LLM-sampled instructions; initialize *exemplars* $\mathcal{E}_0 \leftarrow \emptyset$, best instruction & exemplar so far $I^*_{\leq 0}, E^*_{\leq 0} \leftarrow$ None
3: **for all** generation $t \in \{1, ..., T\}$ **do**
4:     *# Optimize instructions conditional on best exemplars*
5:     Evaluate prompt metrics on $\mathcal{D}_{\text{val}}$: $\widetilde{s} \leftarrow \mathbb{E}_{(x,y) \sim \mathcal{D}_{\text{val}}}\left[g\left(f_{\text{LLM}}\left(x, I_t^{(i)}, E^*_{\leq t},\right), y\right)\right] \forall I_t^{(i)} \in \mathcal{I}_t$.
6:     Retain $\hat{\mathcal{I}}_t$, the top instructions in $\mathcal{I}_t$ in terms of $\widetilde{s}$ and use the optimizer LLM to rewrite without changing the semantic meaning $\mathcal{I}_{t+1} \leftarrow \text{resample}(\hat{\mathcal{I}}_t)$.
7:     Update $I^*_{\leq t} \leftarrow \arg\max_{I_t \in \mathcal{I}_t} \widetilde{s}(I_t | E^*_{\leq t})$
8:     *# Optimize exemplars conditional on best instructions*
9:     **if** $\mathcal{E}_t = \emptyset$ **then**
10:         Fill $\mathcal{E}_{t+1}$ with input-output (inc. intermediate outputs) pairs randomly sampled from $\mathcal{D}_c(I^*_{\leq t})$, the subset of $\mathcal{D}_{\text{val}}$ where the target model answered correctly under the *best instruction seen so far*: $\forall E_j \in \mathcal{E}_{t+1}, E_j \leftarrow \{e_j^{(m)}\}_{m=1}^k, e_j \sim \mathcal{D}_c(I^*_{\leq t})$
11:     **else**
12:         Fill $\mathcal{E}_{t+1}$ with $|\mathcal{E}|$ mutated copies of $E^*_{\leq t}$: randomly replace an input-output pair in $E^*_{\leq t}$ with another pair from $\mathcal{D}_c(I^*_{\leq t})$.
13:     **end if**
14:     Evaluate prompt metrics on $\mathcal{D}_{\text{val}}$: $\widetilde{s} \leftarrow \mathbb{E}_{(x,y) \sim \mathcal{D}_{\text{val}}}\left[g\left(f_{\text{LLM}}\left(x, I^*_{\leq t}, E_t^{(j)},\right), y\right)\right] \forall E_t^{(j)} \in \mathcal{E}_t$
15:     Update $E^*_{\leq t} \leftarrow \arg\max_{E_t \in \mathcal{E}_t} \widetilde{s}(E_t | I^*_{\leq t})$
16: **end for**
17: **Return** instruction-exemplar combination with the highest score.

We present the comparison between the original and inverted orders in Table 20, where we found that the findings (in particular, *Insights 2 and 3*) are not affected by the optimization order. For the interleaved approach, we find it to achieve a test accuracy of 74.6% on 26 BBH tasks on PaLM 2, which is broadly on par with the two-staged approach with $m_{\text{IO}} = 16, m_{\text{EO}} = 16$ – we thus primarily focused on the two-staged approach for its simplicity, and defer a detailed exploration on better ways to exploit the interdependence of exemplars and instructions to future work.

Table 20: Effect of ordering between IO and EO on selected BBH tasks of (**left**) Gemini 1.0 Pro and (**right**) Gemini 1.5 Flash. *Original* refers to the IO *before* EO order used in the main text and *Inverted* refers to the IO-*after*-EO optimization order.

| Task $m = 32$ | IO only | EO only | Combined *Original* | Combined *Inverted* | Task $m = 32$ | IO only | EO only | Combined *Original* | Combined *Inverted* |
|---|---|---|---|---|---|---|---|---|---|
| boolean_expressions | 79.50 | 91.00 | 90.50 | 90.50 | boolean_expressions | 90.50 | 99.50 | 96.00 | 99.00 |
| causal_judgement | 54.67 | 59.33 | 66.00 | 66.00 | causal_judgement | 68.00 | 67.33 | 67.33 | 63.33 |
| date_understanding | 73.50 | 78.50 | 79.50 | 74.50 | date_understanding | 80.00 | 84.50 | 84.00 | 83.00 |
| geometric_shapes | 33.00 | 71.00 | 67.50 | 84.50 | geometric_shapes | 54.50 | 82.00 | 90.50 | 81.00 |
| logical_deduction_seven_objects | 31.00 | 50.00 | 47.50 | 51.00 | logical_deduction_seven_objects | 50.00 | 70.00 | 80.50 | 72.00 |
| logical_deduction_three_objects | 49.00 | 76.50 | 78.00 | 79.00 | logical_deduction_three_objects | 91.00 | 89.50 | 92.50 | 93.50 |
| multistep_arithmetic_two | 77.50 | 84.00 | 79.50 | 81.00 | multistep_arithmetic_two | 91.50 | 91.50 | 92.50 | 91.50 |
| navigate | 58.00 | 85.50 | 89.50 | 89.00 | navigate | 66.00 | 69.00 | 74.00 | 85.00 |
| object_counting | 78.50 | 95.50 | 94.50 | 94.00 | object_counting | 86.50 | 95.50 | 93.50 | 93.50 |
| penguins_in_a_table | 78.63 | 84.62 | 93.16 | 84.62 | penguins_in_a_table | 95.73 | 94.87 | 95.73 | 94.87 |
| temporal_sequences | 79.50 | 84.00 | 77.50 | 82.50 | temporal_sequences | 97.50 | 99.50 | 99.50 | 100.00 |
| Avg. test acc. (%) ↑ | 68.98 | 78.18 | 78.47 | **79.69** | Avg. test acc. (%) ↑ | 79.20† | 85.75 | **87.82** | 86.97 |

### B.9 Case Study on How LLMs Respond to Exemplars and Instructions

Complementary to *Insight 2* and Fig. 8, in this section, we give some examples on the LLM response when prompted with the optimized instructions and/or exemplars in representative tasks.

Table 21: Case study on `tracking_shuffled_objects` (7): **Top table**: the discovered prompts by various methods (only one exemplar is shown under "Best EO" for conciseness). **Bottom table**: response of LLM to two representative questions in the test set when conditioned on each of the discovered prompt – note that instructions in this case do not fully specify the model behavior. Even when the LLM is indeed following instructions generate reasoning steps, the effectiveness of reasoning templates vary and LLMs are error-prone and often track unnecessary items. With carefully selected exemplars, however, the LLM can perform behavior imitation via adapting and chaining the pattern "`[A] and [B] [do something]. [A] had [X] and [B] had [Y]. So now, [A] has [Y] and [B] has [X]`" to achieve an extraordinary accuracy. Some texts are truncated with ... for better presentation.

| Method (Test acc.) | Prompt |
|---|---|
| Base (38%) | Let's follow a step-by-step process. |
| Best IO (52%) | Let's carefully analyze the question and dissect our reasoning process. In order to do this, we will need to track multiple pieces of information over time. We will also need to pay attention to the details of each swap that occurs. |
| Best EO (85%) | Q: Alice, Bob, Claire, Dave, Eve, Fred, and Gertrude are dancers at a square dance. At the start of a song, they each have a partner: Alice is dancing with Ophelia, Bob is dancing with Melissa, Claire is dancing with Jamie, Dave is dancing with Sam, Eve is dancing with Patrick, Fred is dancing with Rodrigo, and Gertrude is dancing with Karl. Throughout the song, the dancers often trade partners. First, Dave and Claire switch partners. Then, Alice and Eve switch partners. ... Finally, Dave and Alice switch partners. At the end of the dance, Fred is dancing with Options: (A) Ophelia ... (G) Karl
Show your final answer option bracketed between <answer> and </answer> at the end.
A: Let's follow a step-by-step process. Let's follow the steps one by one:
1. Dave and Claire switch partners.
Dave was dancing with Sam, and Claire was dancing with Jamie. So now, Dave is dancing with Jamie, and Claire is dancing with Sam. ...
7. Dave and Alice switch partners.
Dave was dancing with Karl, and Alice was dancing with Patrick. So now, Dave is dancing with Patrick, and Alice is dancing with Karl.
At the end of the dance, Fred is dancing with Melissa.
<answer>(B)</answer> |

| Question | Responses | | |
|---|---|---|---|
| | Base | Best IO | Best EO |
| Alice, Bob, Claire, Dave, Eve, Fred, and Gertrude are holding a white elephant gift exchange. At the start of the event, they are each holding a present of a different color: Alice has a pink ball, Bob has a red present, Claire has a blue present, Dave has a brown present, Eve has a yellow present, Fred has a white present, and Gertrude has a black ball. As the event progresses, pairs of people swap gifts. First, Alice and Claire swap their gifts. Then, Dave and Bob swap their gifts. ... Finally, Claire and Alice swap their gifts. At the end of the event, Alice has the | 1. Alice and Claire swap their gifts: Alice now has the blue present, and Claire has the pink ball.
2. Dave and Bob swap their gifts: Dave now has the red present, and Bob has the brown present.
...
7. Claire and Alice swap their gifts: Claire now has the yellow present, and Alice has the black ball.
At the end of the event, Alice has the <answer>(G)</answer> black ball. | 1. Alice and Claire swap: Alice (pink ball) -> Claire, Claire (blue present) -> Alice.
2. Dave and Bob swap: Dave (brown present) -> Bob, Bob (red present) -> Dave.
...
7. Claire and Alice swap: Claire (pink ball) -> Alice, Alice (yellow present) -> Claire.
At the end of the swaps, Alice has the yellow present. <answer>(E)</answer>. | 1. Alice and Claire swap their gifts. Alice had a pink ball, and Claire had a blue present. So now, Alice has the blue present, and Claire has the pink ball.
2. Dave and Bob swap their gifts. Dave had a brown present, and Bob had a red present. So now, Dave has the red present, and Bob has the brown present.
...
7. Claire and Alice swap their gifts. Claire had the blue present, and Alice had the yellow present. So now, Claire has the yellow present, and Alice has the blue present.
At the end of the event, Alice has the blue present. <answer>(C)</answer> |
| Q: Alice, Bob, Claire, Dave, Eve, Fred, and Gertrude are on the same team in a soccer match. At the start of the match, they are each assigned to a position: Alice is playing cheerleader, Bob is playing right midfielder, Claire is playing right winger, Dave is playing center midfielder, Eve is playing fullback, Fred is playing striker, and Gertrude is playing goalkeeper. As the game progresses, pairs of players occasionally swap positions. First, Bob and Alice trade positions. Then, Claire and Fred trade positions. ... Finally, Bob and Dave trade positions. At the end of the match, Bob is playing Options: (A) cheerleader ... (G) goalkeeper | At the start of the match, Bob is playing right midfielder.
1. Bob and Alice trade positions. Now, Bob is playing cheerleader and Alice is playing right midfielder.
2. Claire and Fred trade positions. This does not affect Bob's position.
...
7. Bob and Dave trade positions. Now, Bob is playing center midfielder and Dave is playing right midfielder.
At the end of the match, Bob is playing center midfielder. <answer>(D)</answer> | 1. Bob and Alice trade positions: Bob becomes the cheerleader, and Alice becomes the right midfielder.
2. Claire and Fred trade positions: Claire becomes the striker, and Fred becomes the right winger.
...
7. Bob and Dave trade positions: Bob becomes the center midfielder, and Dave becomes the right midfielder.
At the end of the swaps, Bob is playing center midfielder. <answer>(D)</answer> | 1. Bob and Alice trade positions. Bob was playing right midfielder, and Alice was playing cheerleader. So now, Bob is playing cheerleader, and Alice is playing right midfielder.
2. Claire and Fred trade positions. Claire was playing right winger, and Fred was playing striker. So now, Claire is playing striker, and Fred is playing right winger.
...
7. Bob and Dave trade positions. Bob was playing cheerleader, and Dave was playing right midfielder. So now, Bob is playing right midfielder, and Dave is playing cheerleader.
At the end of the match, Bob is playing right midfielder. <answer>(B)</answer> |

Table 22: Case study on `web_of_lies`: **Top table**: discovered prompts. **Bottom table**: response of LLM to two representative questions in the test set when conditioned on each of the discovered prompt. The winning pattern that the LLM copy behavior from is "`[A] says [B] [lies/tells the truth]. So [A] [lies/tells the truth]`" to achieve a near-perfect accuracy.

| Method (Test acc.) | Prompt |
|---|---|
| Base (55%) | Let's think step by step. |
| Best IO (82%) | Let's think step by step. In each step, we will look at the statement made by each person and determine whether they are telling the truth or not. If a person is telling the truth, we will continue to the next step. If a person is lying, we will stop and conclude that the person who said the previous statement is also lying. |
| Best EO (97%) | Question: Maybelle tells the truth. Antwan says Maybelle tells the truth. Audrie says Antwan tells the truth. Ryan says Audrie lies. Delfina says Ryan lies. Does Delfina tell the truth? Show your final answer (Yes or No only) bracketed between <answer> and </answer>. 
 A: Let's think step by step. Let's think step by step. 
 Maybelle tells the truth. 
 Antwan says Maybelle tells the truth. So Antwan tells the truth. 
 Audrie says Antwan tells the truth. So Audrie tells the truth. 
 Ryan says Audrie lies. So Ryan lies. 
 Delfina says Ryan lies. So Delfina tells the truth. 
 <answer>Yes</answer> |

| Question | Responses | | |
|---|---|---|---|
| | Base | Best IO | Best EO |
| Question: Jamey tells the truth. Delfina says Jamey lies. Vina says Delfina tells the truth. Teressa says Vina tells the truth. Conception says Teressa lies. Does Conception tell the truth? | If Conception tells the truth, then Teressa lies. 
 If Teressa lies, then Vina tells the truth. 
 If Vina tells the truth, then Delfina tells the truth. 
 If Delfina tells the truth, then Jamey lies. 
 But Jamey tells the truth. So, Conception cannot tell the truth. 
 The final answer is <answer>No</answer>. | Jamey tells the truth. 
 Delfina says Jamey lies. (If Delfina is telling the truth, then Jamey is lying, which contradicts the given statement. Therefore, Delfina is lying.) 
 Vina says Delfina tells the truth. (If Vina is telling the truth, then Delfina is telling the truth, which contradicts the previous conclusion. Therefore, Vina is lying.) 
 Teressa says Vina tells the truth. (If Teressa is telling the truth, then Vina is telling the truth, which contradicts the previous conclusion. Therefore, Teressa is lying.) 
 Conception says Teressa lies. (If Conception is telling the truth, then Teressa is telling the truth, which contradicts the previous conclusion. Therefore, Conception is lying.) 
 The final answer is <answer>No</answer>. | Jamey tells the truth. 
 Delfina says Jamey lies. So Delfina lies. 
 Vina says Delfina tells the truth. So Vina lies. 
 Teressa says Vina tells the truth. So Teressa lies. 
 Conception says Teressa lies. So Conception tells the truth. 
 <answer>Yes</answer> |
| Question: Kandi tells the truth. Ryan says Kandi lies. Michaela says Ryan tells the truth. Sima says Michaela lies. Andree says Sima tells the truth. Does Andree tell the truth? | If Andree tells the truth, then Sima lies. If Sima lies, then Michaela tells the truth. If Michaela tells the truth, then Ryan lies. If Ryan lies, then Kandi tells the truth. But Kandi tells the truth in the first place. This is a contradiction. 
 Therefore, Andree does not tell the truth. 
 <answer>No</answer> | Step 1: Kandi tells the truth. 
 Step 2: Ryan says Kandi lies. (Contradicts Step 1, so Ryan is lying.) 
 Step 3: Michaela says Ryan tells the truth. (Contradicts Step 2, so Michaela is lying.) 
 Step 4: Sima says Michaela lies. (Contradicts Step 3, so Sima is lying.) 
 Step 5: Andree says Sima tells the truth. (Contradicts Step 4, so Andree is lying.) 
 <answer>No</answer> | Kandi tells the truth. 
 Ryan says Kandi lies. So Ryan lies. 
 Michaela says Ryan tells the truth. So Michaela lies. 
 Sima says Michaela lies. So Sima tells the truth. 
 Andree says Sima tells the truth. So Andree tells the truth. 
 The final answer is <answer>Yes</answer>. |

## B.10 Comparison Against PromptBreeder

In this section, we compare the two-stage EO-IO combined search mentioned in §3.2 against PromptBreeder [14], the SoTA method that has the option of optimizing both instructions and exemplars. In contrast to our method which is a simple combination of APE (for instruction optimization) and mutation (for exemplar optimization), PromptBreeder features a much expansive search space introduced by its different mutation operators on the instruction and the meta-prompts (referred to as *hyper-mutation* in Fernando et al. [14]). As such, the algorithm requires many more iterations before convergence – for our experiment, we use a population size of 40 and allow for 10 generations. Since PromptBreeder uses binary tournament evolution for each generation which will result in half of its population being replaced with new prompts requiring evaluations $\mathcal{D}_{\text{val}}$, there are total $40 + 10 \times (40/2) = 240$ traversals on the validation set $\mathcal{D}_{\text{val}}$, which is approximately an order-of-magnitude larger than the budget used in this paper and the discovered prompts can be

deemed as an approximation of the performance "upper bound" when a SoTA optimizing agent is left to freely explore an expansive search space with ample search budget. As a comparison, we run our two-staged EO and IO algorithm for 100 iterations, with first 25 iterations used for IO and the remaining 75 for EO. Due to the much larger computational costs, we run the comparison on four selected tasks: `movie_recommendation`, `multistep_arithmetic_two`, `object_counting` and `ruin_names`.

Table 23: Comparison of the simple two-stage IO-EO algorithm introduced in §3.2 (with a budget of 100 validation set evaluations) and PromptBreeder (with a budget of 240 validation set evaluations) on 4 representative tasks using PaLM 2 (`text-bison-002`) model.

| Dataset | Best validation acc. (%) | | Test acc. (%) | |
|---|---|---|---|---|
| | Ours | PromptBreeder | Ours | PromptBreeder |
| movie_recommendation | 94.00 | 94.00 | 89.50 | 89.00 |
| multistep_arithmetic_two | 86.00 | 84.00 | 79.00 | 74.50 |
| object_counting | 100.00 | 100.00 | 99.50 | 98.00 |
| ruin_names | 86.00 | 92.00 | 82.00 | 87.00 |

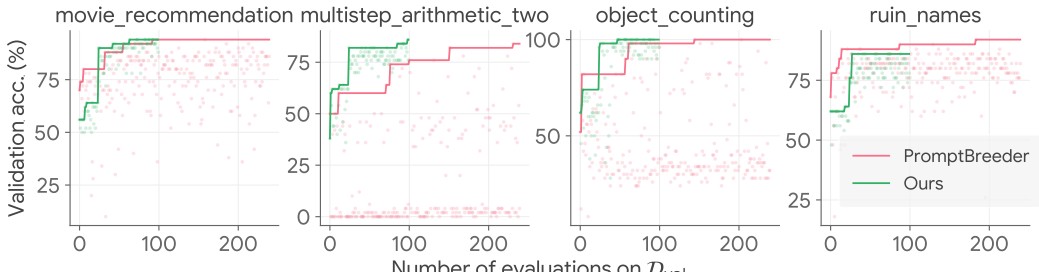

Figure 20: Comparison of validation accuracy vs. number of evaluations on the validation set between our two-stage IO-EO search and PromptBreeder. Scattered points denote the performance of each evaluated instruction and the line denotes the best instruction up to that point. Note that PromptBreeder uses 2.5× more search budget.

We summarize the results in Table 23 and Fig. 20. It is worth noting that our simple two-stage search *performs on par or better than PromptBreeder in 3/4 tasks* in terms of both validation and test metrics while using less than half of the budget and using a much simpler instruction optimization heuristic. Given that the chief difference between our algorithm and PromptBreeder is the explicit iterative optimization for exemplars (noting that the "context shuffling" routine used in PromptBreeder, which permutes the exemplars in the prompt, relies purely on stochasticity), this again suggests that the importance of exemplar optimization may actually outweigh complicated instruction optimization as we highlighted in *Insight 2* in §3.2 – in fact, even the the PromptBreeder authors themselves remarked in App. J that "... *We find that in the few-shot evolution case, the contexts dominate, and often the task-prompts drift into nonsense. They are less critically determining of fitness than the evolved contexts.*".

### B.11  Additional Examples of How SoTA IO Uses Exemplars

Complementary to *Insight 4* in §3.2 of the main text, in this section, we show additional examples illustrating how SoTA IO techniques may spontaneously and opportunistically use quasi-exemplars. For selected examples, we also perform ablation experiments similar to the experiment in Fig. 8 where we either retain or remove the identified quasi-exemplars to study the effect of their presence on the final performance and we show the results in Table 24 – it is further worth noting that in multiple cases, the instruction optimizers discovered instructions that consist of quasi-exemplars only (e.g., `multi_arithmetic_two` and `movie_recommendation` in Table 24), which further corroborates our point made in *Insight 4*.

Table 24: Additional examples where SoTA instruction optimizers generate quasi-exemplars (highlighted in yellow) discussed in *Insight 4* in §3.2 in the main text. Where possible, we perform similar ablation experiment as in Fig. 8 to either retain the quasi-exemplars or remove the quasi-exemplars and re-run evaluation on the test set. It is worth noting that in several cases, the instruction optimizer generates instructions consisting of quasi-exemplars *only* (marked by asterisks in the table below) where it is not possible to perform the ablation experiments.

| Dataset | Prompt | Test accuracy (%) | | |
| --- | --- | --- | --- | --- |
| | | Full prompt | Quasi-exemplars **only** | **Removing** quasi-exemplars |
| movie_recommendation | **Watched Movies:**
- The Shawshank Redemption (1994)
- The Godfather (1972)
- The Dark Knight (2008)
- Pulp Fiction (1994)
**Potential Recommendations:**
1. The Green Mile (1999)
2. One Flew Over the Cuckoo's Nest (1975)
3. The Silence of the Lambs (1991)
4. Fight Club (1999)
**Hint:** The recommended movie is a psychological thriller that explores the dark side of human nature and features a memorable performance from its lead actor. | 89.0 | * | * |
| multi_arithmetic_two | **Solve for x:**
$$2(x + 3) - 5 = 15$$
**Hint:** Distribute the 2 to the expression inside the parentheses, then combine like terms.
**Solution:**
$$2(x + 3) - 5 = 15$$
$$2x + 6 - 5 = 15$$
$$2x + 1 = 15$$
$$2x = 14$$
$$x = 7$$ | 74.5 | * | * |
| ruin_names | **Instruction Mutant:**
When faced with a challenge like this, where you need to find a one-character edit that humorously changes the meaning of an artist, band, or movie name, here's a creative approach to help you solve it:
1. **Think Beyond the Obvious:** Don't just focus on the first letter or the last letter. Look for opportunities to change the meaning by altering any character within the name.
2. **Play with Homonyms:** Explore words that sound similar but have different meanings. For example, "Beatles" could become "Beetles," or "Metallica" could become "Meatballs."
3. **Consider Puns:** Puns can be a great way to add humor. For instance, "The Rolling Stones" could become "The Rolling Scones."
4. **Look for Visual Changes:** Sometimes, a simple change in punctuation or capitalization can create a humorous visual effect. For example, "The Cure" could become "The Cure?"
5. **Incorporate Pop Culture:** Reference current events, popular trends, or well-known phrases to add a topical twist. For instance, "Star Wars" could become "Star Whores."
6. **Don't Be Afraid to Be Silly:** The goal is to make people laugh, so don't be afraid to come up with something wacky and unexpected. Remember, the key is to be creative and have fun with it. | 87.0 | * | * |
| object_counting | Let's think step by step:
1. First, we need to understand the problem. We have a collection of possessions that a person has along with their quantities. We need to determine the number of a certain object/item class.
2. Next, we need to identify the object/item class for which we want to determine the number.
3. Once we have identified the object/item class, we need to find all the possessions that belong to that class.
4. We then need to sum up the quantities of all the possessions that belong to the object/item class.
5. The sum of the quantities will give us the total number of possessions that belong to the object/item class.
Let's take an example to understand this better. Suppose we have the following collection of possessions:
```
{
"car": 1,
"house": 1,
"bicycle": 2,
"dog": 1,
"cat": 2
}
```
We want to determine the number of pets that the person has. In this case, the object/item class is "pets". The possessions that belong to this class are "dog" and "cat". The quantities of these possessions are 1 and 2, respectively. The sum of these quantities is 3. Therefore, the person has 3 pets.
We can use the same approach to determine the number of any other object/item class in the collection. | 84.0 | 72.5 (-11.5) | 80.0 (-4.0) |

## C Best Prompts Discovered

In this section, we present the best optimized prompts discovered by any method on each of the BBH tasks studied.

```
Q: ( not ( False ) or False ) is
Show your final answer (True or False only) bracketed between <answer> and </answer>.
A: Let's use a stepwise approach. ( not ( False ) or False ) is
Show your final answer (True or False only) bracketed between <answer> and </answer>.
A: Let's use a stepwise approach.
1. not ( False ) = True
2. True or False = True
<answer>True</answer>
==

Q: ( True and not True and True ) is
Show your final answer (True or False only) bracketed between <answer> and </answer>.
A: Let's use a stepwise approach. ( True and not True and True ) is
Show your final answer (True or False only) bracketed between <answer> and </answer>.
A: Let's use a stepwise approach.
1. not True is False.
2. True and False is False.
3. False and True is False.
<answer>False</answer>
==

Q: True and False or ( not False ) is
Show your final answer (True or False only) bracketed between <answer> and </answer>.
A: Let's use a stepwise approach. True and False or ( not False ) is
Show your final answer (True or False only) bracketed between <answer> and </answer>.
A: Let's use a stepwise approach.
1. ( not False ) is True.
2. True or True is True.
<answer>True</answer>
==
```

Listing 1: `boolean_expressions`: OPRO + Mutation. Test accuracy = 95.0%

```
Q:: How would a typical person answer each of the following questions about causation?
A lieutenant was talking with a sergeant. The lieutenant gave the order: 'Send your squad
    to the top of Thompson Hill.' The sergeant said: 'But if I send my squad to the top
    of Thompson Hill, we'll be moving the men directly into the enemy's line of fire.
    Some of them will surely be killed!' The lieutenant answered: 'Look, I know that
    they'll be in the line of fire, and I know that some of them will be killed. But I
    don't care at all about what happens to our soldiers. All I care about is taking
    control of Thompson Hill.' The squad was sent to the top of Thompson Hill. As
    expected, the soldiers were moved into the enemy's line of fire, and some of them
    were killed. Did the lieutenant intentionally put the soldiers in the line of fire?
Options:
- Yes
- No
Show your final answer (Yes or No only) bracketed between <answer> and </answer>.
A: Let's think step by step. <answer>Yes</answer>
==

Q: How would a typical person answer each of the following questions about causation?
Jen sees some puppies playing next to her driveway again and wants to kill them. She
    decides to go to the hardware store to buy some rat poison that she thinks will work
    on the puppies. As she pulls out of her garage, the wheel slips in her hand and she
    drives off to the side of the driveway. All the puppies are crushed and killed
    under the car. With the puppies eliminated, Jen doesn't need to go to the hardware
    store. Did Jen intentionally kill the puppies?
Options:
- Yes
- No
Show your final answer (Yes or No only) bracketed between <answer> and </answer>.
A: Let's think step by step. <answer>No</answer>
==

Q: How would a typical person answer each of the following questions about causation?
Billy and Suzy work for the same company. They work in different rooms, and both of them
    sometimes need to access the central computer of the company. Nobody at the company
    is aware that if two people are logged into the central computer at the same time,
    some spam emails containing important customer information are immediately deleted
    from the central computer. In order to make sure that one person is always available
```

```
      to answer incoming phone calls, the company issued the following official policy:
      Suzy is the only one permitted to log into the central computer in the mornings,
      whereas Billy is the only one permitted to log into the central computer in the
      afternoons. Billy is not permitted to log into the central computer in the morning.
      Today at 9 am, Billy and Suzy both log into the central computer at the same time.
      Immediately, some work emails containing important customer information are deleted
      from the central computer. Did Suzy cause the central computer to delete some work
      emails containing important customer information?
Options:
- Yes
- No
Show your final answer (Yes or No only) bracketed between <answer> and </answer>.
A: Let's think step by step. <answer>No</answer>
==
```

Listing 2: `causal_judgement`: No IO + Random Search. Test accuracy = 68.7%

```
Q: Jane thinks today is 6/18/2019, but John thinks today is 6/19/2019. John is correct.
      What is the date 10 days ago in MM/DD/YYYY?
Options:
(A) 06/08/2019
(B) 08/09/2018
(C) 06/09/2019
(D) 05/19/2019
(E) 06/09/2044
(F) 06/15/2019
Show your final answer option bracketed between <answer> and </answer> at the end.
A: Let's think step by step. Today is 6/19/2019. 10 days ago would be 6/19/2019 - 10 =
      6/9/2019.
<answer>(C)</answer>
==

Q: Jane was born on the last day of Feburary in 2001. Today is her 16-year-old birthday.
      What is the date a month ago in MM/DD/YYYY?
Options:
(A) 11/12/2016
(B) 01/21/2017
(C) 01/14/2017
(D) 01/28/2017
(E) 02/03/2017
Show your final answer option bracketed between <answer> and </answer> at the end.
A: Let's think step by step. Jane was born on the last day of February in 2001, which is
      February 28, 2001. Today is her 16th birthday, which means it is February 28, 2017.
      A month ago from today would be January 28, 2017.
<answer>(D)</answer>
==

Q: The concert was scheduled to be on 06/01/1943, but was delayed by one day to today.
      What is the date tomorrow in MM/DD/YYYY?
Options:
(A) 06/03/1943
(B) 03/15/1943
(C) 08/03/1942
(D) 06/02/1943
(E) 07/04/1943
(F) 05/06/1943
Show your final answer option bracketed between <answer> and </answer> at the end.
A: Let's think step by step. The concert was scheduled to be on 06/01/1943, but was
      delayed by one day to today. So today is 06/02/1943. Tomorrow will be 06/03/1943.
<answer>(A)</answer>
==
```

Listing 3: `date_understanding`: No IO + Mutation. Test accuracy = 81.0%

```
Q: In the following sentences, explain the antecedent of the pronoun (which thing the
      pronoun refers to), or state that it is ambiguous.
Sentence: The writer tried to fool the salesperson and told him a fake story.
Options:
(A) Told the writer a fake story
(B) Told the salesperson a fake story
(C) Ambiguous
Show your final answer option bracketed between <answer> and </answer> at the end.
A: Let's think step by step. The pronoun "him" refers to "the salesperson".
<answer>(B)</answer>
==

Q: In the following sentences, explain the antecedent of the pronoun (which thing the
      pronoun refers to), or state that it is ambiguous.
```

```
Sentence: The chief encouraged the writer because she wrote a good book.
Options:
(A) The chief wrote a good book
(B) The writer wrote a good book
(C) Ambiguous
Show your final answer option bracketed between <answer> and </answer> at the end.
A: Let's think step by step. The pronoun "she" refers to the writer.
The antecedent of the pronoun "she" is "the writer".
<answer>(B)</answer>
==

Q: In the following sentences, explain the antecedent of the pronoun (which thing the
    pronoun refers to), or state that it is ambiguous.
Sentence: The reporter and the chef discuss their favorite dishes.
Options:
(A) They discuss the reporter's favorite dishes
(B) They discuss the chef's favorite dishes
(C) Ambiguous
Show your final answer option bracketed between <answer> and </answer> at the end.
A: Let's think step by step. The pronoun "their" refers to both the reporter and the chef
    .
The final answer is: <answer>(C) Ambiguous</answer>
==
```

Listing 4: `disambiguation_qa`: No IO + Mutation. Test accuracy = 71.5%

```
For each given argument, classify it as deductively valid or invalid based on the
    explicitly stated premises. A deductively valid argument is one where the conclusion
     logically follows from the premises. If an argument is deductively invalid, even
    when the premises are true, the conclusion can still be false.

Be cautious, some arguments may have premises that are nonsensical or contradictory. In
    such cases, simply focus on determining whether or not the conclusion is supported
    by the premises, regardless of their content.
Read each argument and provided premises carefully and attentively. If the argument can
    be demonstrated to be invalid based on the premises, respond with "invalid,"
    otherwise, answer "valid."
"Is Fred a cousin of Robert? Is Joe related to Bob? In large families, it is sometimes
    difficult to keep track of all one's relatives. The following argument seeks to
    clarify some such relations: First of all, every great-grandfather of Chad is an
    ancestor of Douglas or a cousin of Henry. Next, every schoolmate of Trevor is
    neither a cousin of Henry nor an ancestor of Douglas. Hence, no great-grandfather of
     Chad is a schoolmate of Trevor."
Is the argument, given the explicitly stated premises, deductively valid or invalid?
Options:
- valid
- invalid
Show your final answer (valid or invalid only) bracketed between <answer> and </answer>.
 <answer>valid</answer>
==

Be cautious, some arguments may have premises that are nonsensical or contradictory. In
    such cases, simply focus on determining whether or not the conclusion is supported
    by the premises, regardless of their content.
Read each argument and provided premises carefully and attentively. If the argument can
    be demonstrated to be invalid based on the premises, respond with "invalid,"
    otherwise, answer "valid."
"Here comes a perfectly valid argument: First of all, every loyal buyer of Tocca soap is
    an occasional purchaser of Bentley Organic soap. Next, being a loyal buyer of Tocca
    soap is sufficient for being a frequent consumer of L'Oreal shampoo. Plus,some
    regular user of Lever soap is not an occasional purchaser of Bentley Organic soap or
     not a frequent consumer of L'Oreal shampoo. We may conclude that not every regular
     user of Lever soap is a loyal buyer of Tocca soap."
Is the argument, given the explicitly stated premises, deductively valid or invalid?
Options:
- valid
- invalid
Show your final answer (valid or invalid only) bracketed between <answer> and </answer>.
 <answer>valid</answer>
==

Be cautious, some arguments may have premises that are nonsensical or contradictory. In
    such cases, simply focus on determining whether or not the conclusion is supported
    by the premises, regardless of their content.
Read each argument and provided premises carefully and attentively. If the argument can
    be demonstrated to be invalid based on the premises, respond with "invalid,"
    otherwise, answer "valid."
```

```
"Here comes a perfectly valid argument: First premise: Whatever is ingredient of
    Concealer is at least one of these: an ingredient of HEART SHAPED BALM, an
    ingredient of Goth Fairy or an ingredient of Love Me Like A Wimp. Second premise:
    Being an ingredient of Clarifying Mask is necessary for being an ingredient of Goth
    Fairy. Third premise: Being an ingredient of HEART SHAPED BALM is sufficient for
    being an ingredient of Clarifying Mask. Fourth premise: Every ingredient of Love Me
    Like A Wimp is an ingredient of Clarifying Mask. So, necessarily, being an
    ingredient of Concealer is sufficient for being an ingredient of Clarifying Mask."
Is the argument, given the explicitly stated premises, deductively valid or invalid?
Options:
- valid
- invalid
Show your final answer (valid or invalid only) bracketed between <answer> and </answer>.
 <answer>valid</answer>
==
```

Listing 5: `formal_fallacies`: ProTeGi + Random Search. Test accuracy = 63.0%

```
Q: This SVG path element <path d="M 25.00,38.00 L 89.00,58.00"/> draws a
Options:
(A) circle
(B) heptagon
(C) hexagon
(D) kite
(E) line
(F) octagon
(G) pentagon
(H) rectangle
(I) sector
(J) triangle
Show your final answer option bracketed between <answer> and </answer> at the end.
A: We should proceed in a sequential manner. A: We should proceed in a sequential manner.
     The path element starts at point (25, 38) and draws a line to point (89, 58). So,
    it is a line.
<answer>(E)</answer>
==

Q: This SVG path element <path d="M 21.22,43.76 L 13.59,7.93 L 49.15,35.84 M 49.15,35.84
    L 21.22,43.76"/> draws a
Options:
(A) circle
(B) heptagon
(C) hexagon
(D) kite
(E) line
(F) octagon
(G) pentagon
(H) rectangle
(I) sector
(J) triangle
Show your final answer option bracketed between <answer> and </answer> at the end.
A: We should proceed in a sequential manner. A: We should proceed in a sequential manner.

The first part of the path element, "M 21.22,43.76 L 13.59,7.93", draws a line from the
    point (21.22, 43.76) to the point (13.59, 7.93).

The second part of the path element, "L 49.15,35.84", draws a line from the point (13.59,
     7.93) to the point (49.15, 35.84).

The third part of the path element, "M 49.15,35.84 L 21.22,43.76", draws a line from the
    point (49.15, 35.84) back to the point (21.22, 43.76).

This path element draws a triangle.
<answer>(J)</answer>
==

Q: This SVG path element <path d="M 53.07,5.04 L 61.70,11.14 L 62.83,22.38 L 66.68,29.10
    L 63.41,40.33 L 49.09,37.83 L 38.28,25.14 L 52.62,23.25 L 53.07,5.04"/> draws a
Options:
(A) circle
(B) heptagon
(C) hexagon
(D) kite
(E) line
(F) octagon
(G) pentagon
(H) rectangle
(I) sector
(J) triangle
```

```
Show your final answer option bracketed between <answer> and </answer> at the end.
A: We should proceed in a sequential manner. The path starts at point (53.07, 5.04) and
    then goes to point (61.70, 11.14). This is a line segment.
Next, it goes to point (62.83, 22.38). This is another line segment.
It continues to point (66.68, 29.10). This is a third line segment.
Then, it goes to point (63.41, 40.33). This is a fourth line segment.
Next, it goes to point (49.09, 37.83). This is a fifth line segment.
It continues to point (38.28, 25.14). This is a sixth line segment.
Then, it goes to point (52.62, 23.25). This is a seventh line segment.
Finally, it returns to the starting point (53.07, 5.04). This is an eighth line segment.

So, the path consists of eight line segments that connect nine points. This forms a
    polygon. To determine the type of polygon, we need to count the number of sides.
    Since there are eight sides, the polygon is an octagon.

<answer>(F)</answer>
==
```

Listing 6: `geometric_shapes`: APE + Random Search. Test accuracy = 88.5%

```
Let's think step by step.

The correct order of adjectives is:
1. Opinion
2. Size
3. Shape
4. Color
5. Origin
6. Material
7. Purpose

Now, let's look at the options:
(A) little black driving ship
(B) driving little black ship

The correct answer is (A) because it follows the correct order of adjectives.

<answer>(A)</answer>

Which sentence has the correct adjective order:
Options:
(A) large archaic red Indonesian sock
(B) large red Indonesian archaic sock
Show your final answer option bracketed between <answer> and </answer> at the end.

 The correct order of adjectives is:
1. Size
2. Shape
3. Color
4. Origin
5. Material
6. Purpose

Now, let's look at the options:
(A) large archaic red Indonesian sock
(B) large red Indonesian archaic sock

The correct answer is (A) because it follows the correct order of adjectives.

<answer>(A)</answer>
==

Which sentence has the correct adjective order:
Options:
(A) German gray lovely ship
(B) lovely gray German ship
Show your final answer option bracketed between <answer> and </answer> at the end.

 The correct order of adjectives is:
1. Opinion
2. Size
3. Shape
4. Color
5. Origin
6. Material
7. Purpose

Now, let's look at the options:
(A) German gray lovely ship
(B) lovely gray German ship
```

```
The correct answer is (B) because it follows the correct order of adjectives.

<answer>(B)</answer>
==
```

Listing 7: hyperbaton: PromptAgent + Random Search. Test accuracy = 90.5%

```
Q: The following paragraphs each describe a set of five objects arranged in a fixed order
    . The statements are logically consistent within each paragraph. In a golf
    tournament, there were five golfers: Mel, Dan, Amy, Joe, and Eve. Amy finished below
     Dan. Mel finished first. Joe finished above Dan. Eve finished last.
Options:
(A) Mel finished second
(B) Dan finished second
(C) Amy finished second
(D) Joe finished second
(E) Eve finished second
Show your final answer option bracketed between <answer> and </answer> at the end.
A: Here's the thought process I used to solve the problem. **Thought process:**

1. Mel finished first.
2. Eve finished last.
3. Amy finished below Dan.
4. Joe finished above Dan.

Based on these statements, we can conclude that:

- Mel finished first.
- Eve finished fifth.
- Dan finished third.
- Amy finished fourth.
- Joe finished second.

**Answer:** <answer>(D)</answer>
==

Q: The following paragraphs each describe a set of five objects arranged in a fixed order
    . The statements are logically consistent within each paragraph. On a shelf, there
    are five books: a green book, an orange book, a gray book, a yellow book, and a
    purple book. The green book is the rightmost. The gray book is to the right of the
    orange book. The purple book is to the left of the yellow book. The purple book is
    to the right of the gray book.
Options:
(A) The green book is the leftmost
(B) The orange book is the leftmost
(C) The gray book is the leftmost
(D) The yellow book is the leftmost
(E) The purple book is the leftmost
Show your final answer option bracketed between <answer> and </answer> at the end.
A: Here's the thought process I used to solve the problem. The green book is the
    rightmost.
The gray book is to the right of the orange book.
The purple book is to the left of the yellow book.
The purple book is to the right of the gray book.

From the first statement, we know that the green book is on the rightmost side. From the
    second statement, we know that the gray book is to the right of the orange book.
    This means that the orange book must be to the left of the gray book. From the third
     statement, we know that the purple book is to the left of the yellow book. From the
     fourth statement, we know that the purple book is to the right of the gray book.
    This means that the gray book must be between the purple book and the orange book.
    Therefore, the order of the books from left to right is: orange, gray, purple,
    yellow, green.

<answer>(B)</answer>
==

Q: The following paragraphs each describe a set of five objects arranged in a fixed order
    . The statements are logically consistent within each paragraph. In a golf
    tournament, there were five golfers: Amy, Mel, Rob, Joe, and Ada. Joe finished
    second. Joe finished below Amy. Mel finished second-to-last. Ada finished last.
Options:
(A) Amy finished first
(B) Mel finished first
(C) Rob finished first
(D) Joe finished first
(E) Ada finished first
Show your final answer option bracketed between <answer> and </answer> at the end.
```

```
A: Here's the thought process I used to solve the problem. **Step 1: Understand the
    information given in each paragraph.**

The first paragraph describes a golf tournament with five golfers: Amy, Mel, Rob, Joe,
    and Ada.

**Step 2: Identify the key information related to the order of finish.**

- Joe finished second.
- Joe finished below Amy.
- Mel finished second-to-last.
- Ada finished last.

**Step 3: Use the given information to deduce the order of finish.**

Based on the information provided, we can conclude the following:

- Ada finished last.
- Mel finished second-to-last.
- Joe finished second.
- Amy finished above Joe.

**Step 4: Determine the golfer who finished first.**

Since Amy finished above Joe, who finished second, and no other information is given
    about the golfers who finished ahead of Joe, we can conclude that Amy finished first
    .

**Step 5: Select the correct answer option.**

The answer option that corresponds to Amy finishing first is (A).

**Final Answer:** <answer>(A)</answer>
==
```

Listing 8: `logical_deduction_five_objects`: OPRO + Random Search. Test accuracy = 62.5%

```
Q: The following paragraphs each describe a set of seven objects arranged in a fixed
    order. The statements are logically consistent within each paragraph. A fruit stand
    sells seven fruits: plums, kiwis, cantaloupes, pears, watermelons, apples, and
    loquats. The watermelons are more expensive than the cantaloupes. The apples are
    less expensive than the cantaloupes. The watermelons are the second-most expensive.
    The loquats are less expensive than the kiwis. The apples are more expensive than
    the loquats. The loquats are the third-cheapest. The plums are the cheapest.
Options:
(A) The plums are the cheapest
(B) The kiwis are the cheapest
(C) The cantaloupes are the cheapest
(D) The pears are the cheapest
(E) The watermelons are the cheapest
(F) The apples are the cheapest
(G) The loquats are the cheapest
Show your final answer option bracketed between <answer> and </answer> at the end.
A: I need to think step-by-step. Step 1: Identify the most expensive fruit.
The watermelons are the second-most expensive. So the most expensive fruit must be either
     the cantaloupes or the kiwis.

Step 2: Identify the cheapest fruit.
The plums are the cheapest.

Step 3: Determine the order of the fruits from most expensive to least expensive.
The most expensive fruit is either the cantaloupes or the kiwis. The watermelons are the
    second-most expensive. The apples are less expensive than the cantaloupes. The
    loquats are less expensive than the kiwis. The apples are more expensive than the
    loquats. The loquats are the third-cheapest.

Step 4: Identify the cheapest fruit.
The plums are the cheapest.

<answer>A</answer>
==

Q: The following paragraphs each describe a set of seven objects arranged in a fixed
    order. The statements are logically consistent within each paragraph. On a shelf,
    there are seven books: a purple book, a brown book, a red book, a blue book, a gray
    book, a black book, and a green book. The brown book is to the left of the green
    book. The gray book is the second from the left. The black book is to the left of
    the gray book. The blue book is to the left of the red book. The blue book is the
    second from the right. The green book is to the left of the purple book.
Options:
```

```
(A) The purple book is the second from the right
(B) The brown book is the second from the right
(C) The red book is the second from the right
(D) The blue book is the second from the right
(E) The gray book is the second from the right
(F) The black book is the second from the right
(G) The green book is the second from the right
Show your final answer option bracketed between <answer> and </answer> at the end.
A: I need to think step-by-step. Step 1: Identify the position of the gray book.
The gray book is the second from the left.

Step 2: Identify the books to the left of the gray book.
The black book is to the left of the gray book.

Step 3: Identify the books to the right of the gray book.
The brown book is to the left of the green book.
The green book is to the left of the purple book.
Therefore, the books to the right of the gray book are the green book, the purple book,
    the red book, and the blue book.

Step 4: Identify the position of the blue book.
The blue book is the second from the right.

Step 5: Determine which book is the second from the right.
The blue book is the second from the right.

<answer>D</answer>
==

Q: The following paragraphs each describe a set of seven objects arranged in a fixed
    order. The statements are logically consistent within each paragraph. In a golf
    tournament, there were seven golfers: Joe, Mel, Mya, Eve, Rob, Ada, and Dan. Rob
    finished first. Eve finished last. Dan finished above Ada. Mel finished above Mya.
    Joe finished second-to-last. Mya finished third.
Options:
(A) Joe finished second-to-last
(B) Mel finished second-to-last
(C) Mya finished second-to-last
(D) Eve finished second-to-last
(E) Rob finished second-to-last
(F) Ada finished second-to-last
(G) Dan finished second-to-last
Show your final answer option bracketed between <answer> and </answer> at the end.
A: I need to think step-by-step. **Step 1:** Identify the finishing positions of the
    golfers mentioned in the paragraph.
- Rob finished first.
- Eve finished last.
- Dan finished above Ada.
- Mel finished above Mya.
- Joe finished second-to-last.
- Mya finished third.

**Step 2:** Determine the remaining finishing positions.
- Since Rob finished first and Mya finished third, the remaining golfers must finish in
    positions 2, 4, 5, 6, and 7.
- Dan finished above Ada, so Dan must be in position 4 or 5 and Ada must be in position 5
    or 6.
- Mel finished above Mya, so Mel must be in position 2 or 3.

**Step 3:** Identify the golfer who finished second-to-last.
- Joe finished second-to-last, so the answer is (A).

**Final Answer:** <answer>(A)</answer>
==
```

Listing 9: `logical_deduction_seven_objects`: OPRO + Mutation. Test accuracy = 62.5%

```
Q: The following paragraphs each describe a set of three objects arranged in a fixed
    order. The statements are logically consistent within each paragraph. On a branch,
    there are three birds: a blue jay, a cardinal, and a quail. The cardinal is to the
    right of the quail. The quail is to the right of the blue jay.
Options:
(A) The blue jay is the rightmost
(B) The cardinal is the rightmost
(C) The quail is the rightmost
Show your final answer option bracketed between <answer> and </answer> at the end.
A: Let's think step by step. The quail is to the right of the blue jay. The cardinal is
    to the right of the quail. So the order is: blue jay, quail, cardinal.
<answer>(B)</answer>
==
```

```
Q: The following paragraphs each describe a set of three objects arranged in a fixed
     order. The statements are logically consistent within each paragraph. In a golf
     tournament, there were three golfers: Ana, Mel, and Ada. Ada finished above Ana. Mel
     finished below Ana.
Options:
(A) Ana finished first
(B) Mel finished first
(C) Ada finished first
Show your final answer option bracketed between <answer> and </answer> at the end.
A: Let's think step by step. Ada finished above Ana.
Mel finished below Ana.
So, the order is Ada, Ana, Mel.
<answer>(C)</answer>
==

Q: The following paragraphs each describe a set of three objects arranged in a fixed
     order. The statements are logically consistent within each paragraph. On a branch,
     there are three birds: a crow, a falcon, and a raven. The crow is to the left of the
     raven. The falcon is to the right of the raven.
Options:
(A) The crow is the rightmost
(B) The falcon is the rightmost
(C) The raven is the rightmost
Show your final answer option bracketed between <answer> and </answer> at the end.
A: Let's think step by step. A: Let's think step by step.
The crow is to the left of the raven.
The falcon is to the right of the raven.
So the order is: crow, raven, falcon.
The falcon is the rightmost.
<answer>(B)</answer>
==
```

Listing 10: `logical_deduction_three_objects`: No IO + Random Search. Test accuracy =
84.5%

```
Let's think step by step.

First, let's identify the common themes or genres of the given movies.

Then, let's look at the options and choose the one that best fits the common themes or
    genres.

If none of the options fit the common themes or genres perfectly, let's choose the option
     that is most similar to the given movies in terms of its popularity and well-
    knownness.

Finally, let's bracket the final answer option between <answer> and </answer>.

Find a movie similar to Schindler's List, Pulp Fiction, Braveheart, The Usual Suspects:
Options:
(A) 12 Angry Men
(B) Mo' Better Blues
(C) Mindhunters
(D) The Shawshank Redemption
Show your final answer option bracketed between <answer> and </answer> at the end.

 The common theme among the given movies is that they are all critically acclaimed and
    highly regarded in the film industry. They also cover various genres, including
    historical drama (Schindler's List), crime (Pulp Fiction), historical epic (
    Braveheart), and crime thriller (The Usual Suspects).

Out of the options provided, the movie that best fits these criteria is:

<answer>(D) The Shawshank Redemption</answer>
==

Find a movie similar to American History X, Requiem for a Dream, Raiders of the Lost Ark,
     The Sixth Sense:
Options:
(A) Miss Meadows
(B) Little Man
(C) Goodfellas
(D) You Can't Take It with You
Show your final answer option bracketed between <answer> and </answer> at the end.

 The common theme among the given movies is that they are all critically acclaimed and
    well-known films.
```

```
While the genres of the movies vary, they are all considered to be classic films that
    have stood the test of time.

Out of the options provided, the movie that is most similar to the given movies in terms
    of its popularity and well-knownness is Goodfellas.

<answer>(C) Goodfellas</answer>
==

Find a movie similar to Batman, The Usual Suspects, The Silence of the Lambs, Jurassic
    Park:
Options:
(A) The End of the Tour
(B) Forrest Gump
(C) Lust
(D) Caution
(E) Shelter
Show your final answer option bracketed between <answer> and </answer> at the end.

 The common theme among the given movies is that they are all critically acclaimed and
    popular films from the 1990s.

Out of the options, the movie that is most similar to the given movies in terms of its
    popularity and well-knownness is Forrest Gump.

<answer>(B) Forrest Gump</answer>
==
```

Listing 11: `movie_recommendation`: PromptAgent + Mutation. Test accuracy = 95.0%

```
Solve the following math problems by following the steps in the order of operations:

1. When multiplying or dividing two negative numbers, the result will be positive.
2. When multiplying or dividing a positive and a negative number, the result will be
    negative.
3. When adding or subtracting a negative number, it is the same as adding or subtracting
    its positive counterpart.
((6 - 0 * 5 + -3) * (6 - -7 + -2 - -7)) =
Show your final answer bracketed between <answer> and </answer>.
 ((6 - 0 * 5 + -3) * (6 - -7 + -2 - -7)) =
(6 - 0 + -3) * (6 - -7 + -2 - -7)
(6 + -3) * (6 - -7 + -2 - -7)
3 * (6 - -7 + -2 - -7)
3 * (6 + 7 + -2 + 7)
3 * (13 + -2 + 7)
3 * (11 + 7)
3 * 18
<answer>54</answer>
==

1. When multiplying or dividing two negative numbers, the result will be positive.
2. When multiplying or dividing a positive and a negative number, the result will be
    negative.
3. When adding or subtracting a negative number, it is the same as adding or subtracting
    its positive counterpart.
((8 * 9 * 0 - -1) - (-9 - -7 + -4 - 8)) =
Show your final answer bracketed between <answer> and </answer>.
 ((8 * 9 * 0 - -1) - (-9 - -7 + -4 - 8)) =
1. 8 * 9 * 0 - -1 = 0 + 1 = 1
2. -9 - -7 + -4 - 8 = -9 + 7 - 4 - 8 = -14
3. 1 - (-14) = 1 + 14 = 15
<answer>15</answer>
==

1. When multiplying or dividing two negative numbers, the result will be positive.
2. When multiplying or dividing a positive and a negative number, the result will be
    negative.
3. When adding or subtracting a negative number, it is the same as adding or subtracting
    its positive counterpart.
((1 * 0 - -5 - 3) + (9 * 6 + -4 - 0)) =
Show your final answer bracketed between <answer> and </answer>.
 ((1 * 0 - -5 - 3) + (9 * 6 + -4 - 0)) =
1. (1 * 0 - -5 - 3) = (0 + 5 - 3) = 2
2. (9 * 6 + -4 - 0) = (54 + -4 - 0) = 50
3. 2 + 50 = 52
<answer>52</answer>
==
```

Listing 12: `multistep_arithmetic_two`: ProTeGi + Mutation. Test accuracy = 82.5%

```
Given a set of instructions, determine whether following those instructions will take you
    back to the exact same spot you started from. Keep in mind any movements, including
    turns and the direction of any steps. For example, if you take 2 steps forward and
    then 2 steps backward, you will end up in the same spot. Determine your answer by
    saying "yes" or "no".

If you follow these instructions, do you return to the starting point? Take 2 steps. Take
    10 steps. Take 8 steps.
Options:
- Yes
- No
Show your final answer (Yes or No only) bracketed between <answer> and </answer>.
 <answer>No</answer>
==

If you follow these instructions, do you return to the starting point? Always face
    forward. Take 6 steps forward. Take 7 steps backward. Take 7 steps backward. Take 8
    steps forward.
Options:
- Yes
- No
Show your final answer (Yes or No only) bracketed between <answer> and </answer>.
 <answer>Yes</answer>
==

If you follow these instructions, do you return to the starting point? Take 6 steps. Turn
    around. Take 8 steps. Take 2 steps.
Options:
- Yes
- No
Show your final answer (Yes or No only) bracketed between <answer> and </answer>.
 <answer>No</answer>
==
```

Listing 13: `navigate`: ProTeGi + Random Search. Test accuracy = 68.5%

```
Q: I have a yam, a stalk of celery, a potato, a cabbage, two cauliflowers, an onion, a
    lettuce head, a head of broccoli, a carrot, and a garlic. How many vegetables do I
    have?
Show your final answer bracketed between <answer> and </answer>.
A: Let's break down the problem into smaller steps and solve each one. A: Let's break
    down the problem into smaller steps and solve each one.

1. Identify the vegetables:
   - Yam
   - Celery
   - Potato
   - Cabbage
   - Cauliflower (2)
   - Onion
   - Lettuce
   - Broccoli
   - Carrot
   - Garlic

2. Count the number of vegetables:
   - Yam: 1
   - Celery: 1
   - Potato: 1
   - Cabbage: 1
   - Cauliflower: 2
   - Onion: 1
   - Lettuce: 1
   - Broccoli: 1
   - Carrot: 1
   - Garlic: 1

3. Add up the counts:
   1 + 1 + 1 + 1 + 2 + 1 + 1 + 1 + 1 + 1 = 11

Therefore, the final answer is <answer>11</answer>.
==

Q: I have a cauliflower, a cabbage, a garlic, a lettuce head, an onion, a head of
    broccoli, a potato, and a carrot. How many vegetables do I have?
```

```
Show your final answer bracketed between <answer> and </answer>.
A: Let's break down the problem into smaller steps and solve each one. A: Let's break
    down the problem into smaller steps and solve each one.

1. Identify the vegetables:
    - Cauliflower
    - Cabbage
    - Garlic
    - Lettuce head
    - Onion
    - Broccoli
    - Potato
    - Carrot

2. Count the number of vegetables:
    - There are 8 items in the list.

Therefore, the answer is <answer>8</answer>.
==

Q: I have an accordion, a clarinet, and three drums. How many musical instruments do I
    have?
Show your final answer bracketed between <answer> and </answer>.
A: Let's break down the problem into smaller steps and solve each one. A: Let's break
    down the problem into smaller steps and solve each one.

1. We have one accordion.
2. We have one clarinet.
3. We have three drums.

Adding these up, we get:

1 + 1 + 3 = 5

Therefore, we have <answer>5</answer> musical instruments.
==
```

Listing 14: `object_counting`: OPRO + Random Search. Test accuracy = 99.0%

```
Q: Here is a table where the first line is a header and each subsequent line is a penguin
    :  name, age, height (cm), weight (kg) Louis, 7, 50, 11 Bernard, 5, 80, 13 Vincent,
    9, 60, 11 Gwen, 8, 70, 15  For example: the age of Louis is 7, the weight of Gwen is
    15 kg, the height of Bernard is 80 cm.   Which is the second heaviest penguin?
Options:
(A) Louis
(B) Bernard
(C) Vincent
(D) Gwen
(E) James
Show your final answer option bracketed between <answer> and </answer> at the end.
A: Let's proceed in a sequential manner. The heaviest penguin is Gwen, who weighs 15 kg.
    The second heaviest penguin must be either Louis, Bernard, or Vincent, as they all
    weigh less than Gwen.

Louis weighs 11 kg, Bernard weighs 13 kg, and Vincent weighs 11 kg. Therefore, Bernard is
    the second heaviest penguin.

<answer>(B)</answer>
==

Q: Here is a table where the first line is a header and each subsequent line is a penguin
    :  name, age, height (cm), weight (kg) Louis, 7, 50, 11 Bernard, 5, 80, 13 Vincent,
    9, 60, 11 Gwen, 8, 70, 15  For example: the age of Louis is 7, the weight of Gwen is
    15 kg, the height of Bernard is 80 cm.   We then delete the penguin named Bernard
    from the table.
How many penguins are more than 8 years old?
Options:
(A) 1
(B) 2
(C) 3
(D) 4
(E) 5
Show your final answer option bracketed between <answer> and </answer> at the end.
A: Let's proceed in a sequential manner. The question asks the number of penguins that
    are more than 8 years old.

The table has 4 penguins: Louis, Bernard, Vincent, and Gwen.

Bernard is deleted from the table, so we have 3 penguins left.
```

```
Louis is 7 years old, Vincent is 9 years old, and Gwen is 8 years old.

Therefore, there is only 1 penguin that is more than 8 years old.

<answer>(A)</answer>
==

Q: Here is a table where the first line is a header and each subsequent line is a penguin
    :  name, age, height (cm), weight (kg) Louis, 7, 50, 11 Bernard, 5, 80, 13 Vincent,
    9, 60, 11 Gwen, 8, 70, 15  For example: the age of Louis is 7, the weight of Gwen is
     15 kg, the height of Bernard is 80 cm.  How many penguins are less than 8 years old
    ?
Options:
(A) 1
(B) 2
(C) 3
(D) 4
(E) 5
Show your final answer option bracketed between <answer> and </answer> at the end.
A: Let's proceed in a sequential manner. Louis is 7 years old.
Bernard is 5 years old.
Vincent is 9 years old.
Gwen is 8 years old.

There are two penguins less than 8 years old.
<answer>(B)</answer>
==
```

Listing 15: `penguins_in_a_table`: APE + Random Search. Test accuracy = 89.7%

```
Q: On the table, I see three black stress balls, one orange stress ball, two purple
    keychains, one black dog leash, one orange keychain, three orange cat toys, one
    black keychain, and one purple stress ball. If I remove all the cat toys from the
    table, how many orange items remain on it?
Options:
(A) zero
(B) one
(C) two
(D) three
(E) four
(F) five
(G) six
(H) seven
(I) eight
(J) nine
(K) ten
(L) eleven
(M) twelve
(N) thirteen
(O) fourteen
(P) fifteen
(Q) sixteen
Show your final answer option bracketed between <answer> and </answer> at the end.
A: Let's take a methodical approach. A: Let's take a methodical approach.

There are three black stress balls, one orange stress ball, two purple keychains, one
    black dog leash, one orange keychain, three orange cat toys, one black keychain, and
     one purple stress ball.

If we remove all the cat toys from the table, we are left with:

Three black stress balls
One orange stress ball
Two purple keychains
One black dog leash
One orange keychain
One black keychain
One purple stress ball

Counting the orange items, we have:

One orange stress ball
One orange keychain

So, there are two orange items remaining on the table.

<answer>C</answer>
==
```

```
Q: On the floor, you see a bunch of things arranged in a row: a turquoise paperclip, an
    orange bracelet, a green keychain, and a silver pen. What is the color of the thing
    directly to the left of the silver thing?
Options:
(A) red
(B) orange
(C) yellow
(D) green
(E) blue
(F) brown
(G) magenta
(H) fuchsia
(I) mauve
(J) teal
(K) turquoise
(L) burgundy
(M) silver
(N) gold
(O) black
(P) grey
(Q) purple
(R) pink
Show your final answer option bracketed between <answer> and </answer> at the end.
A: Let's take a methodical approach. A: Let's take a methodical approach. The question
    asks for the color of the thing directly to the left of the silver thing. The given
    items are:
- turquoise paperclip
- orange bracelet
- green keychain
- silver pen

Reading from left to right, the silver pen is the last item. The item directly to its
    left is the green keychain. Therefore, the color of the thing directly to the left
    of the silver thing is green.

<answer>D</answer>
==

Q: On the floor, there is a green pencil, a yellow mug, a fuchsia booklet, and a pink pen
    . Is the booklet fuchsia?
Options:
(A) yes
(B) no
Show your final answer option bracketed between <answer> and </answer> at the end.
A: Let's take a methodical approach. <answer>A</answer>
==
```

Listing 16: `reasoning_about_colored_objects`: APE + Diversity. Test accuracy = 80.5%

```
Q: Which of the following is a humorous edit of this artist or movie name: 'spice girls'?
Options:
(A) spihe girls
(B) spic girls
(C) spice tgirls
(D) spite girls
Show your final answer option bracketed between <answer> and </answer> at the end.
A: Determine which answer choice represents the most plausible humorous edit. The
    humorous edit that stands out is:
<answer>(D) spite girls</answer>
This edit plays on the idea of spite, which is a feeling of resentment or ill will,
    adding a humorous twist to the original name "Spice Girls."
==

Q: Which of the following is a humorous edit of this artist or movie name: 'the men who
    stare at goats'?
Options:
(A) the men who stare at boats
(B) uhe men who stare at goats
(C) the men who stare vt goats
(D) the omen who stare at goats
Show your final answer option bracketed between <answer> and </answer> at the end.
A: Determine which answer choice represents the most plausible humorous edit. The
    humorous edit that stands out is:
(A) the men who stare at boats

This edit introduces a comical twist by replacing "goats" with "boats," creating an
    amusing mental image of men intently staring at boats instead of goats. The other
```

```
          options either contain typos or do not significantly alter the original name in a
          humorous way.

<answer>(A)</answer>
==

Q: Which of the following is a humorous edit of this artist or movie name: 'gone with the
       wind'?
Options:
(A) gowne with the wind
(B) gone with the wand
(C) gone with the wing
(D) gone witth the wind
Show your final answer option bracketed between <answer> and </answer> at the end.
A: Determine which answer choice represents the most plausible humorous edit. <answer>(B)
       gone with the wand</answer>
==
```

Listing 17: ruin_names: OPRO + Mutation. Test accuracy = 91.0%

```
Q: The following translations from German to English contain a particular error. That
      error will be one of the following types: Named Entities: An entity (names, places,
      locations, etc.) is changed to a different entity. Numerical Values: Numerical
      values (ordinals or cardinals), dates, and/or units are changed. Modifiers or
      Adjectives: The modifiers and adjectives pertaining to a noun are changed. Negation
      or Antonyms: Introduce or remove a negation or change comparatives to their antonyms
      . Facts: Trivial factual errors not pertaining to the above classes are introduced
      in the translations. Dropped Content: A significant clause in the translation is
      removed. Please identify that error.  Source: Boyd Kevin Rutherford ist ein US-
      amerikanischer Politiker.
Translation: Boyd is an American politician.
The translation contains an error pertaining to
Options:
(A) Modifiers or Adjectives
(B) Numerical Values
(C) Negation or Antonyms
(D) Named Entities
(E) Dropped Content
(F) Facts
Show your final answer option bracketed between <answer> and </answer> at the end.
A: Let's think step by step. The error in the translation is that the name "Boyd Kevin
      Rutherford" is reduced to just "Boyd".
The correct translation should be: Boyd Kevin Rutherford is an American politician.
Therefore, the error pertains to (D) Named Entities.

The final answer is: <answer>(D)</answer>
==

Q: The following translations from German to English contain a particular error. That
      error will be one of the following types: Named Entities: An entity (names, places,
      locations, etc.) is changed to a different entity. Numerical Values: Numerical
      values (ordinals or cardinals), dates, and/or units are changed. Modifiers or
      Adjectives: The modifiers and adjectives pertaining to a noun are changed. Negation
      or Antonyms: Introduce or remove a negation or change comparatives to their antonyms
      . Facts: Trivial factual errors not pertaining to the above classes are introduced
      in the translations. Dropped Content: A significant clause in the translation is
      removed. Please identify that error.  Source: Dunau, laut Ortstafel Dunau-Lathwehren
       und auf Landkarten auch als Lathwehren-Dunau bezeichnet, ist eine Ortslage des
      Stadtteils Lathwehren der Stadt Seelze in der Region Hannover in Niedersachsen.
Translation: Dunau, according to the local plaque Dunau-Lathwehren, is a locality of the
      district of Lathwehren in the city of Seelze in the Hanover region of Lower Saxony.
The translation contains an error pertaining to
Options:
(A) Modifiers or Adjectives
(B) Numerical Values
(C) Negation or Antonyms
(D) Named Entities
(E) Dropped Content
(F) Facts
Show your final answer option bracketed between <answer> and </answer> at the end.
A: Let's think step by step. The original German sentence mentions "laut Ortstafel Dunau-
      Lathwehren und auf Landkarten auch als Lathwehren-Dunau bezeichnet". This part is
      translated as "according to the local plaque Dunau-Lathwehren". The error here is
      that the translation does not mention "auf Landkarten auch als Lathwehren-Dunau
      bezeichnet", which means "also referred to as Lathwehren-Dunau on maps".
The correct translation should be:
Dunau, according to the local plaque Dunau-Lathwehren and also referred to as Lathwehren-
      Dunau on maps, is a locality of the district of Lathwehren in the city of Seelze in
      the Hanover region of Lower Saxony.
```

```
Therefore, the error pertains to (E) Dropped Content.

Final answer: <E>
==

Q: The following translations from German to English contain a particular error. That
    error will be one of the following types: Named Entities: An entity (names, places,
    locations, etc.) is changed to a different entity. Numerical Values: Numerical
    values (ordinals or cardinals), dates, and/or units are changed. Modifiers or
    Adjectives: The modifiers and adjectives pertaining to a noun are changed. Negation
    or Antonyms: Introduce or remove a negation or change comparatives to their antonyms
    . Facts: Trivial factual errors not pertaining to the above classes are introduced
    in the translations. Dropped Content: A significant clause in the translation is
    removed. Please identify that error.  Source: Karte mit allen Koordinaten: OSM |
    WikiMap
Translation: Map with no coordinates: OSM | WikiMap
The translation contains an error pertaining to
Options:
(A) Modifiers or Adjectives
(B) Numerical Values
(C) Negation or Antonyms
(D) Named Entities
(E) Dropped Content
(F) Facts
Show your final answer option bracketed between <answer> and </answer> at the end.
A: Let's think step by step. The original German sentence is "Karte mit allen Koordinaten
    : OSM | WikiMap". The English translation is "Map with no coordinates: OSM | WikiMap
    ". The error in the translation is that the word "allen" (all) has been translated
    as "no". This is an error of Negation or Antonyms.
<answer>(C)</answer>
==
```

Listing 18: `salient_translation_error_detection`: APE + Random Search. Test accuracy = 62.5%

```
Q: Which statement is sarcastic?
Options:
(A) You can still achieve the American Dream after working at Walmart
(B) You can still achieve the American Dream of working at Walmart
Show your final answer option bracketed between <answer> and </answer> at the end.
A: We should move forward in stages. <answer>(B)</answer>
==

Q: Which statement is sarcastic?
Options:
(A) Anyone can be successful without having inherited all their money first
(B) Nobody can be successful without having inherited all their money first
Show your final answer option bracketed between <answer> and </answer> at the end.
A: We should move forward in stages. <answer>(B)</answer>
==

Q: Which statement is sarcastic?
Options:
(A) We should start a petition to ban breathing
(B) We should start a petition to ban vaping
Show your final answer option bracketed between <answer> and </answer> at the end.
A: We should move forward in stages. <answer>(A)</answer>
==
```

Listing 19: `snarks`: APE + Random Search. Test accuracy = 87.4%

```
To address this question, we will divide it into smaller steps and use the provided
    knowledge about sports.

Provided knowledge:
 Dejounte Murray plays in the NBA.
 Goal tending is a basketball rule.
 Toni Kroos is a professional football player.
 FA Cup is a football competition.
 Travis Konecny is a hockey player.
 Bryce Harper is a baseball player.
 Tuukka Rask is a hockey player.
 Giorgio Chiellini is a football player.
 Maradona is a move in football.
 Back shoulder fade is a move in football.
 Powerplay is a term in hockey.
 Handball is an illegal move in football, where a player touches the ball with their hand
     during a match.
```

```
 Side-step three is a move in basketball.
 Studs up is a term in football.
 Blocking foul is a penalty in football.

Initially, we will define what it means for a sentence to be plausible in the context of
     sports. It is logical and consistent with the facts about sports when a sentence is
     plausible.

Now, we will look into the given four examples. For example 1, "Travis Konecny did a
     maradona on the defender," we can infer that this sentence is not plausible since
     maradona is a move in football, not hockey. Similarly, for example 2, "Bryce Harper
     hit the back shoulder fade," we can also infer that this sentence is not plausible
     because back shoulder fade is a move in football, not baseball. For example 3, "
     Tuukka Rask killed the powerplay," we can conclude that this sentence is plausible
     because powerplay is a term in hockey. Likewise, for example 4, "Giorgio Chiellini
     committed a handball in the FA Cup," we can also conclude that this sentence is
     plausible because handball is a penalty in football and the FA Cup is a football
     competition.

Based on the above analysis, the correct answers are:
Example 1: no
Example 2: no
Example 3: yes
Example 4: yes

Now, we will examine the two new examples. For example 1, "Dejounte Murray took a side-
     step three," we can conclude that this sentence is plausible since side-step three
     is a move in basketball. Similarly, for example 2, "Luka Modric went in studs up in
     the FA Cup," we can also conclude that this sentence is plausible because studs up
     is a term in football and the FA Cup is a football competition.

Therefore, the correct answers for the new examples are:
Example 1: yes
Example 2: yes
Is the following sentence plausible? "Neymar did a maradona on the defender in the
     Champions Leage Semifinal."
Show your final answer (yes or no only) bracketed between <answer> and </answer>.
 <answer>yes</answer>
==

Provided knowledge:
 Dejounte Murray plays in the NBA.
 Goal tending is a basketball rule.
 Toni Kroos is a professional football player.
 FA Cup is a football competition.
 Travis Konecny is a hockey player.
 Bryce Harper is a baseball player.
 Tuukka Rask is a hockey player.
 Giorgio Chiellini is a football player.
 Maradona is a move in football.
 Back shoulder fade is a move in football.
 Powerplay is a term in hockey.
 Handball is an illegal move in football, where a player touches the ball with their hand
      during a match.
 Side-step three is a move in basketball.
 Studs up is a term in football.
 Blocking foul is a penalty in football.

Initially, we will define what it means for a sentence to be plausible in the context of
     sports. It is logical and consistent with the facts about sports when a sentence is
     plausible.

Now, we will look into the given four examples. For example 1, "Travis Konecny did a
     maradona on the defender," we can infer that this sentence is not plausible since
     maradona is a move in football, not hockey. Similarly, for example 2, "Bryce Harper
     hit the back shoulder fade," we can also infer that this sentence is not plausible
     because back shoulder fade is a move in football, not baseball. For example 3, "
     Tuukka Rask killed the powerplay," we can conclude that this sentence is plausible
     because powerplay is a term in hockey. Likewise, for example 4, "Giorgio Chiellini
     committed a handball in the FA Cup," we can also conclude that this sentence is
     plausible because handball is a penalty in football and the FA Cup is a football
     competition.

Based on the above analysis, the correct answers are:
Example 1: no
Example 2: no
Example 3: yes
Example 4: yes
```

```
Now, we will examine the two new examples. For example 1, "Dejounte Murray took a side-
    step three," we can conclude that this sentence is plausible since side-step three
    is a move in basketball. Similarly, for example 2, "Luka Modric went in studs up in
    the FA Cup," we can also conclude that this sentence is plausible because studs up
    is a term in football and the FA Cup is a football competition.

Therefore, the correct answers for the new examples are:
Example 1: yes
Example 2: yes
Is the following sentence plausible? "Collin Sexton hit the buzzer beater."
Show your final answer (yes or no only) bracketed between <answer> and </answer>.
 <answer>yes</answer>
==

Provided knowledge:
 Dejounte Murray plays in the NBA.
 Goal tending is a basketball rule.
 Toni Kroos is a professional football player.
 FA Cup is a football competition.
 Travis Konecny is a hockey player.
 Bryce Harper is a baseball player.
 Tuukka Rask is a hockey player.
 Giorgio Chiellini is a football player.
 Maradona is a move in football.
 Back shoulder fade is a move in football.
 Powerplay is a term in hockey.
 Handball is an illegal move in football, where a player touches the ball with their hand
     during a match.
 Side-step three is a move in basketball.
 Studs up is a term in football.
 Blocking foul is a penalty in football.

Initially, we will define what it means for a sentence to be plausible in the context of
    sports. It is logical and consistent with the facts about sports when a sentence is
    plausible.

Now, we will look into the given four examples. For example 1, "Travis Konecny did a
    maradona on the defender," we can infer that this sentence is not plausible since
    maradona is a move in football, not hockey. Similarly, for example 2, "Bryce Harper
    hit the back shoulder fade," we can also infer that this sentence is not plausible
    because back shoulder fade is a move in football, not baseball. For example 3, "
    Tuukka Rask killed the powerplay," we can conclude that this sentence is plausible
    because powerplay is a term in hockey. Likewise, for example 4, "Giorgio Chiellini
    committed a handball in the FA Cup," we can also conclude that this sentence
    is plausible because handball is a penalty in football and the FA Cup is a football
    competition.

Based on the above analysis, the correct answers are:
Example 1: no
Example 2: no
Example 3: yes
Example 4: yes

Now, we will examine the two new examples. For example 1, "Dejounte Murray took a side-
    step three," we can conclude that this sentence is plausible since side-step three
    is a move in basketball. Similarly, for example 2, "Luka Modric went in studs up in
    the FA Cup," we can also conclude that this sentence is plausible because studs up
    is a term in football and the FA Cup is a football competition.

Therefore, the correct answers for the new examples are:
Example 1: yes
Example 2: yes
Is the following sentence plausible? "Stefon Diggs hit the slant pass."
Show your final answer (yes or no only) bracketed between <answer> and </answer>.
 <answer>yes</answer>
==
```

Listing 20: `sports_understanding`: ProTeGi + Mutation. Test accuracy = 90.0%

```
To solve this problem, we can break it down into smaller steps. The first step is to find
     the time when the person woke up. Once we have that information, we can then
    proceed to the next step, which is to identify the earliest time slot that has not
    been accounted for.

Today, James went to the swimming pool. Between what times could they have gone?
We know that:
James woke up at 5am.
Ashley saw James sitting on a rooftop from 7am to 9am.
Hannah saw James taking photos near the Eiffel Tower from 9am to 7pm.
William saw James buying cookies at a bakery from 7pm to 8pm.
```

```
Linda saw James waiting at the train station from 8pm to 10pm.
The swimming pool was closed after 10pm.
Between what times could James have gone to the swimming pool?
Options:
(A) 7pm to 8pm
(B) 8pm to 10pm
(C) 5am to 7am
(D) 7am to 9am
Show your final answer option bracketed between <answer> and </answer> at the end.
 The person woke up at 5am. Ashley saw James sitting on a rooftop from 7am to 9am. This
    means that James could not have gone to the swimming pool during this time. Hannah
    saw James taking photos near the Eiffel Tower from 9am to 7pm. This means that James
     could not have gone to the swimming pool during this time either. William saw James
     buying cookies at a bakery from 7pm to 8pm. This means that James could not have
    gone to the swimming pool during this time either. Linda saw James waiting at the
    train station from 8pm to 10pm. This means that James could not have gone to the
    swimming pool during this time either. Therefore, the only time slot that is left is
     from 5am to 7am.
<answer>(C)</answer>
==

Today, James went to the beach. Between what times could they have gone?
We know that:
James woke up at 5am.
Sean saw James walking towards the Statue of Liberty from 5am to 6am.
Michael saw James driving to the water park from 6am to 7am.
Anthony saw James reading at the library from 7am to 3pm.
William saw James getting a coffee at the cafe from 4pm to 9pm.
The beach was closed after 9pm.
Between what times could James have gone to the beach?
Options:
(A) 7am to 3pm
(B) 5am to 6am
(C) 4pm to 9pm
(D) 3pm to 4pm
Show your final answer option bracketed between <answer> and </answer> at the end.
 James woke up at 5am.
Sean saw James walking towards the Statue of Liberty from 5am to 6am.
Michael saw James driving to the water park from 6am to 7am.
Anthony saw James reading at the library from 7am to 3pm.
William saw James getting a coffee at the cafe from 4pm to 9pm.
The beach was closed after 9pm.
From the information given, we know that James was at the Statue of Liberty from 5am to 6
    am, at the water park from 6am to 7am, and at the library from 7am to 3pm. This
    means that he could not have gone to the beach during those times. He was also at
    the cafe from 4pm to 9pm, so he could not have gone to the beach then either.
Therefore, the only time slot that is not accounted for is from 3pm to 4pm.
<answer>(D)</answer>
==

Today, Elizabeth went to the park. Between what times could they have gone?
We know that:
Elizabeth woke up at 10am.
Emily saw Elizabeth buying a bike at the bike shop from 10am to 11am.
Leslie saw Elizabeth buying lunch at the deli from 11am to 2pm.
Sarah saw Elizabeth playing tennis at the tennis court from 4pm to 8pm.
The park was closed after 8pm.
Between what times could Elizabeth have gone to the park?
Options:
(A) 10am to 11am
(B) 11am to 2pm
(C) 4pm to 8pm
(D) 2pm to 4pm
Show your final answer option bracketed between <answer> and </answer> at the end.
 Elizabeth woke up at 10am. She was seen at the bike shop from 10am to 11am, at the deli
    from 11am to 2pm, and at the tennis court from 4pm to 8pm. The park was closed after
     8pm. Therefore, the earliest time slot that has not been accounted for is from 2pm
    to 4pm.

<answer>(D)</answer>
==
```

Listing 21: `temporal_sequences`: ProTeGi + Mutation. Test accuracy = 100.0%

```
Let's approach this task in a systematic manner. First, let's identify all the different
    objects that are being swapped in the context. Then, let's trace the swaps and keep
    track of the objects as they change hands. Finally, let's use our understanding of
    the swaps and the initial positions of the objects to answer the question. We also
    need to make sure that all the information necessary to answer the question is
```

```
      contained in the context and check that the number of players is the same as the
      number of objects.

Alice, Bob, Claire, Dave, and Eve are holding a white elephant gift exchange. At the
      start of the event, they are each holding a present of a different color: Alice has
      a purple present, Bob has a blue present, Claire has a brown present, Dave has a
      yellow present, and Eve has a black ball.
As the event progresses, pairs of people swap gifts. First, Alice and Bob swap their
      gifts. Then, Bob and Dave swap their gifts. Then, Claire and Bob swap their gifts.
      Then, Dave and Alice swap their gifts. Finally, Dave and Eve swap their gifts. At
      the end of the event, Dave has the
Options:
(A) purple present
(B) blue present
(C) brown present
(D) yellow present
(E) black ball
Show your final answer option bracketed between <answer> and </answer> at the end.
 Let's trace the swaps one by one:

1. Alice and Bob swap their gifts: Alice gives her purple present to Bob, and Bob gives
      his blue present to Alice. So now Alice has the blue present, and Bob has the purple
       present.

2. Bob and Dave swap their gifts: Bob gives his purple present to Dave, and Dave gives
      his yellow present to Bob. So now Bob has the yellow present, and Dave has the
      purple present.

3. Claire and Bob swap their gifts: Claire gives her brown present to Bob, and Bob gives
      his yellow present to Claire. So now Bob has the brown present, and Claire has the
      yellow present.

4. Dave and Alice swap their gifts: Dave gives his purple present to Alice, and Alice
      gives her blue present to Dave. So now Alice has the purple present again, and Dave
      has the blue present again.

5. Dave and Eve swap their gifts: Dave gives his blue present to Eve, and Eve gives her
      black ball to Dave. So now Dave has the black ball, and Eve has the blue present.

At the end of the swaps, Dave has the black ball.

Therefore, the answer is: <answer>(E)</answer>
==

Alice, Bob, Claire, Dave, and Eve are dancers at a square dance. At the start of a song,
      they each have a partner: Alice is dancing with Rodrigo, Bob is dancing with Sam,
      Claire is dancing with Izzi, Dave is dancing with Jamie, and Eve is dancing with
      Ophelia.
Throughout the song, the dancers often trade partners. First, Bob and Eve switch partners
      . Then, Alice and Bob switch partners. Then, Eve and Bob switch partners. Then, Dave
       and Alice switch partners. Finally, Eve and Claire switch partners. At the end of
      the dance, Alice is dancing with
Options:
(A) Rodrigo
(B) Sam
(C) Izzi
(D) Jamie
(E) Ophelia
Show your final answer option bracketed between <answer> and </answer> at the end.
 Let's identify the dancers and their initial partners:
- Alice is dancing with Rodrigo.
- Bob is dancing with Sam.
- Claire is dancing with Izzi.
- Dave is dancing with Jamie.
- Eve is dancing with Ophelia.

Now, let's trace the swaps:
- Bob and Eve switch partners: Bob is now dancing with Ophelia, and Eve is dancing with
      Sam.
- Alice and Bob switch partners: Alice is now dancing with Sam, and Bob is dancing with
      Rodrigo.
- Eve and Bob switch partners: Eve is now dancing with Rodrigo, and Bob is dancing with
      Sam again.
- Dave and Alice switch partners: Dave is now dancing with Alice, and Alice is dancing
      with Jamie.
- Eve and Claire switch partners: Eve is now dancing with Izzi, and Claire is dancing
      with Rodrigo.

At the end of the dance, Alice is dancing with Jamie.
```

```
The final answer is:
<answer>(D)</answer>
==

Alice, Bob, Claire, Dave, and Eve are dancers at a square dance. At the start of a song,
    they each have a partner: Alice is dancing with Lola, Bob is dancing with Melissa,
    Claire is dancing with Jamie, Dave is dancing with Rodrigo, and Eve is dancing with
    Helga.
Throughout the song, the dancers often trade partners. First, Bob and Eve switch partners
    . Then, Alice and Dave switch partners. Then, Eve and Bob switch partners. Then, Eve
     and Claire switch partners. Finally, Dave and Bob switch partners. At the end of
    the dance, Eve is dancing with
Options:
(A) Lola
(B) Melissa
(C) Jamie
(D) Rodrigo
(E) Helga
Show your final answer option bracketed between <answer> and </answer> at the end.
 Let's identify the dancers and their initial partners:
- Alice: Lola
- Bob: Melissa
- Claire: Jamie
- Dave: Rodrigo
- Eve: Helga

Now, let's trace the swaps:
1. Bob and Eve switch partners: Bob is now dancing with Helga, and Eve is dancing with
    Melissa.
2. Alice and Dave switch partners: Alice is now dancing with Rodrigo, and Dave is dancing
     with Lola.
3. Eve and Bob switch partners: Eve is now dancing with Bob, and Bob is dancing with
    Melissa again.
4. Eve and Claire switch partners: Eve is now dancing with Claire, and Claire is dancing
    with Bob.
5. Dave and Bob switch partners: Dave is now dancing with Melissa, and Bob is dancing
    with Rodrigo.

At the end of the dance, Eve is dancing with Claire.

<answer>(C)</answer>
==
```

Listing 22: `tracking_shuffled_objects_five_objects`: ProTeGi + Random Search. Test accuracy = 70.0%

```
Q: Alice, Bob, Claire, Dave, Eve, Fred, and Gertrude are dancers at a square dance. At
    the start of a song, they each have a partner: Alice is dancing with Lola, Bob is
    dancing with Ophelia, Claire is dancing with Sam, Dave is dancing with Helga, Eve is
     dancing with Patrick, Fred is dancing with Jamie, and Gertrude is dancing with
    Rodrigo.
Throughout the song, the dancers often trade partners. First, Fred and Bob switch
    partners. Then, Eve and Alice switch partners. Then, Fred and Eve switch partners.
    Then, Claire and Fred switch partners. Then, Gertrude and Alice switch partners.
    Then, Alice and Bob switch partners. Finally, Dave and Fred switch partners. At the
    end of the dance, Dave is dancing with
Options:
(A) Lola
(B) Ophelia
(C) Sam
(D) Helga
(E) Patrick
(F) Jamie
(G) Rodrigo
Show your final answer option bracketed between <answer> and </answer> at the end.
A: Let's follow a step-by-step process. Let's follow a step-by-step process.

At the start of the dance, the dancers are paired as follows:

Alice - Lola
Bob - Ophelia
Claire - Sam
Dave - Helga
Eve - Patrick
Fred - Jamie
Gertrude - Rodrigo

Step 1: Fred and Bob switch partners.

Alice - Lola
```

```
Bob - Jamie
Claire - Sam
Dave - Helga
Eve - Patrick
Fred - Ophelia
Gertrude - Rodrigo

Step 2: Eve and Alice switch partners.

Alice - Patrick
Bob - Jamie
Claire - Sam
Dave - Helga
Eve - Lola
Fred - Ophelia
Gertrude - Rodrigo

Step 3: Fred and Eve switch partners.

Alice - Patrick
Bob - Jamie
Claire - Sam
Dave - Helga
Eve - Ophelia
Fred - Lola
Gertrude - Rodrigo

Step 4: Claire and Fred switch partners.

Alice - Patrick
Bob - Jamie
Claire - Lola
Dave - Helga
Eve - Ophelia
Fred - Sam
Gertrude - Rodrigo

Step 5: Gertrude and Alice switch partners.

Alice - Rodrigo
Bob - Jamie
Claire - Lola
Dave - Helga
Eve - Ophelia
Fred - Sam
Gertrude - Patrick

Step 6: Alice and Bob switch partners.

Alice - Jamie
Bob - Rodrigo
Claire - Lola
Dave - Helga
Eve - Ophelia
Fred - Sam
Gertrude - Patrick

Step 7: Dave and Fred switch partners.

Alice - Jamie
Bob - Rodrigo
Claire - Lola
Dave - Sam
Eve - Ophelia
Fred - Helga
Gertrude - Patrick

At the end of the dance, Dave is dancing with Sam.

The final answer is <answer>(C)</answer>.
==

Q: Alice, Bob, Claire, Dave, Eve, Fred, and Gertrude are friends and avid readers who
    occasionally trade books. At the start of the semester, they each buy one new book:
    Alice gets Lolita, Bob gets Moby Dick, Claire gets Hound of the Baskervilles, Dave
    gets Catch-22, Eve gets The Fellowship of the Ring, Fred gets The Pearl, and
    Gertrude gets Frankenstein.
As the semester proceeds, they start trading around the new books. First, Dave and Bob
    swap books. Then, Alice and Eve swap books. Then, Bob and Fred swap books. Then,
    Gertrude and Claire swap books. Then, Fred and Dave swap books. Then, Alice and Fred
```

```
        swap books. Finally, Dave and Bob swap books. At the end of the semester, Claire
        has
Options:
(A) Lolita
(B) Moby Dick
(C) Hound of the Baskervilles
(D) Catch-22
(E) The Fellowship of the Ring
(F) The Pearl
(G) Frankenstein
Show your final answer option bracketed between <answer> and </answer> at the end.
A: Let's follow a step-by-step process. Let's follow a step-by-step process.

At the start of the semester, Alice has Lolita.

After the first swap, Dave has Lolita and Bob has Catch-22.

After the second swap, Alice has The Fellowship of the Ring and Eve has Lolita.

After the third swap, Bob has Lolita and Fred has Catch-22.

After the fourth swap, Gertrude has Lolita and Claire has Frankenstein.

After the fifth swap, Fred has Lolita and Dave has Catch-22.

After the sixth swap, Alice has Lolita and Fred has The Fellowship of the Ring.

After the seventh swap, Dave has Lolita and Bob has The Fellowship of the Ring.

At the end of the semester, Claire has Frankenstein.

<answer>(G)</answer>
==

Q: Alice, Bob, Claire, Dave, Eve, Fred, and Gertrude are dancers at a square dance. At
    the start of a song, they each have a partner: Alice is dancing with Ophelia, Bob is
     dancing with Melissa, Claire is dancing with Jamie, Dave is dancing with Sam, Eve
    is dancing with Patrick, Fred is dancing with Rodrigo, and Gertrude is dancing with
    Karl.
Throughout the song, the dancers often trade partners. First, Dave and Claire switch
    partners. Then, Alice and Eve switch partners. Then, Eve and Bob switch partners.
    Then, Claire and Bob switch partners. Then, Fred and Eve switch partners. Then,
    Gertrude and Dave switch partners. Finally, Dave and Alice switch partners. At the
    end of the dance, Fred is dancing with
Options:
(A) Ophelia
(B) Melissa
(C) Jamie
(D) Sam
(E) Patrick
(F) Rodrigo
(G) Karl
Show your final answer option bracketed between <answer> and </answer> at the end.
A: Let's follow a step-by-step process. Let's follow the steps one by one:

1. Dave and Claire switch partners.
Dave was dancing with Sam, and Claire was dancing with Jamie. So now, Dave is dancing
    with Jamie, and Claire is dancing with Sam.

2. Alice and Eve switch partners.
Alice was dancing with Ophelia, and Eve was dancing with Patrick. So now, Alice is
    dancing with Patrick, and Eve is dancing with Ophelia.

3. Eve and Bob switch partners.
Eve was dancing with Patrick, and Bob was dancing with Melissa. So now, Eve is dancing
    with Melissa, and Bob is dancing with Patrick.

4. Claire and Bob switch partners.
Claire was dancing with Sam, and Bob was dancing with Patrick. So now, Claire is dancing
    with Patrick, and Bob is dancing with Sam.

5. Fred and Eve switch partners.
Fred was dancing with Rodrigo, and Eve was dancing with Melissa. So now, Fred is dancing
    with Melissa, and Eve is dancing with Rodrigo.

6. Gertrude and Dave switch partners.
Gertrude was dancing with Karl, and Dave was dancing with Jamie. So now, Gertrude is
    dancing with Jamie, and Dave is dancing with Karl.

7. Dave and Alice switch partners.
```

```
Dave was dancing with Karl, and Alice was dancing with Patrick. So now, Dave is dancing
    with Patrick, and Alice is dancing with Karl.

At the end of the dance, Fred is dancing with Melissa.

<answer>(B)</answer>
==
```

Listing 23: `tracking_shuffled_objects_seven_objects`: APE + Mutation. Test accuracy = 86.0%

```
We will methodically address this by breaking it down into manageable portions. We will
    monitor the final status of every entity (e.g., Alice, Bob, and Claire) after the
    transactions noted in the problem. We will also monitor the flow of the transactions
    .

Alice, Bob, and Claire are dancers at a square dance. At the start of a song, they each
    have a partner: Alice is dancing with Helga, Bob is dancing with Patrick, and Claire
    is dancing with Sam.
Throughout the song, the dancers often trade partners. First, Claire and Alice switch
    partners. Then, Claire and Bob switch partners. Finally, Alice and Bob switch
    partners. At the end of the dance, Bob is dancing with
Options:
(A) Helga
(B) Patrick
(C) Sam
Show your final answer option bracketed between <answer> and </answer> at the end.
 Let's track the partners of Alice, Bob, and Claire throughout the dance:

- Initially, Alice is dancing with Helga, Bob is dancing with Patrick, and Claire is
    dancing with Sam.

- After the first switch, Claire and Alice switch partners, so Alice is now dancing with
    Sam, and Claire is dancing with Helga.

- After the second switch, Claire and Bob switch partners, so Claire is now dancing with
    Bob, and Bob is dancing with Helga.

- After the third switch, Alice and Bob switch partners, so Alice is now dancing with Bob
    , and Bob is dancing with Sam.

Therefore, at the end of the dance, Bob is dancing with Sam.

<answer>(C)</answer>
==

Alice, Bob, and Claire are dancers at a square dance. At the start of a song, they each
    have a partner: Alice is dancing with Lola, Bob is dancing with Patrick, and Claire
    is dancing with Melissa.
Throughout the song, the dancers often trade partners. First, Alice and Claire switch
    partners. Then, Bob and Claire switch partners. Finally, Claire and Alice switch
    partners. At the end of the dance, Alice is dancing with
Options:
(A) Lola
(B) Patrick
(C) Melissa
Show your final answer option bracketed between <answer> and </answer> at the end.
 Let's track the partners of Alice, Bob, and Claire throughout the dance:

- Initially, Alice is dancing with Lola, Bob is dancing with Patrick, and Claire is
    dancing with Melissa.

- After the first switch, Alice and Claire switch partners, so Alice is now dancing with
    Melissa, and Claire is dancing with Lola.

- After the second switch, Bob and Claire switch partners, so Bob is now dancing with
    Melissa, and Claire is dancing with Patrick.

- After the third switch, Claire and Alice switch partners again, so Claire is now
    dancing with Melissa, and Alice is dancing with Patrick.

At the end of the dance, Alice is dancing with Patrick.

Therefore, the answer is:
<answer>(B)</answer>
==
```

```
Alice, Bob, and Claire are friends and avid readers who occasionally trade books. At the
    start of the semester, they each buy one new book: Alice gets Catch-22, Bob gets
    Moby Dick, and Claire gets The Fellowship of the Ring.
As the semester proceeds, they start trading around the new books. First, Claire and
    Alice swap books. Then, Bob and Alice swap books. Finally, Claire and Bob swap books
    . At the end of the semester, Claire has
Options:
(A) Catch-22
(B) Moby Dick
(C) The Fellowship of the Ring
Show your final answer option bracketed between <answer> and </answer> at the end.
 **Initial Status:**

- Alice: Catch-22
- Bob: Moby Dick
- Claire: The Fellowship of the Ring

**First Trade (Claire and Alice swap books):**

- Alice: The Fellowship of the Ring
- Bob: Moby Dick
- Claire: Catch-22

**Second Trade (Bob and Alice swap books):**

- Alice: Moby Dick
- Bob: The Fellowship of the Ring
- Claire: Catch-22

**Third Trade (Claire and Bob swap books):**

- Alice: Moby Dick
- Bob: Catch-22
- Claire: The Fellowship of the Ring

**Final Status:**

- Alice: Moby Dick
- Bob: Catch-22
- Claire: The Fellowship of the Ring

Therefore, at the end of the semester, Claire has **The Fellowship of the Ring**.

<answer>(C)</answer>
==
```

Listing 24: `tracking_shuffled_objects_three_objects`: ProTeGi + Random Search. Test accuracy = 74.0%

```
Q: Question: Jim lies. Crista says Jim lies. Vernell says Crista tells the truth. Bernita
    says Vernell tells the truth. Millicent says Bernita tells the truth. Does
    Millicent tell the truth?
Show your final answer (Yes or No only) bracketed between <answer> and </answer>.
A: Let's follow a step-by-step approach. Let's follow a step-by-step approach.

1. Jim lies. So, Jim is a liar.

2. Crista says Jim lies. If Crista is telling the truth, then Jim is a liar. But we
    already know that Jim is a liar. So, Crista is telling the truth.

3. Vernell says Crista tells the truth. If Vernell is telling the truth, then Crista is
    telling the truth. We already know that Crista is telling the truth. So, Vernell is
    telling the truth.

4. Bernita says Vernell tells the truth. If Bernita is telling the truth, then Vernell is
    telling the truth. We already know that Vernell is telling the truth. So, Bernita
    is telling the truth.

5. Millicent says Bernita tells the truth. If Millicent is telling the truth, then
    Bernita is telling the truth. We already know that Bernita is telling the truth. So,
    Millicent is telling the truth.

Therefore, the answer is <answer>Yes</answer>.
==

Q: Question: Conception tells the truth. Elanor says Conception tells the truth. Alexis
    says Elanor tells the truth. Fidel says Alexis tells the truth. Delbert says Fidel
    lies. Does Delbert tell the truth?
Show your final answer (Yes or No only) bracketed between <answer> and </answer>.
A: Let's follow a step-by-step approach. Let's follow a step-by-step approach.
```

```
1. Conception tells the truth.
2. Elanor says Conception tells the truth. (Hence, Elanor also tells the truth.)
3. Alexis says Elanor tells the truth. (Hence, Alexis also tells the truth.)
4. Fidel says Alexis tells the truth. (Hence, Fidel also tells the truth.)
5. Delbert says Fidel lies.

If Delbert tells the truth, then Fidel lies. But if Fidel lies, then Delbert must be
     lying. Therefore, Delbert cannot be telling the truth.

<answer>No</answer>
==

Q: Question: Helene tells the truth. Leda says Helene lies. Antwan says Leda tells the
     truth. Andree says Antwan lies. Shaunda says Andree lies. Does Shaunda tell the
     truth?
Show your final answer (Yes or No only) bracketed between <answer> and </answer>.
A: Let's follow a step-by-step approach. Let's follow a step-by-step approach.

1. Helene tells the truth.
2. Leda says Helene lies. (This contradicts statement 1, so Leda is lying.)
3. Antwan says Leda tells the truth. (This contradicts statement 2, so Antwan is lying.)
4. Andree says Antwan lies. (This is true because Antwan is indeed lying.)
5. Shaunda says Andree lies. (This contradicts statement 4, so Shaunda is lying.)

Therefore, the final answer is <answer>No</answer>.
==
```

Listing 25: `web_of_lies`: APE + Random Search. Test accuracy = 99.5%

```
Q: Sort the following words alphabetically: List: roadster earl metaphoric hollandaise
     supranational psyche pedro gloss granola quarantine inauspicious penis baccarat
     arenaceous mackenzie blare bowman
Show your final answer bracketed between <answer> and </answer>.
A: Let's think step by step. The words sorted alphabetically are:
<answer>arenaceous, baccarat, blare, bowman, earl, gloss, granola, hollandaise,
     inauspicious, mackenzie, metaphoric, pedro, penis, psyche, quarantine, roadster,
     supranational</answer>
==

Q: Sort the following words alphabetically: List: almost delphic semitic unbeknownst
     astute deadlock libertarian antic levitate execution sanction scathe storehouse
     dandelion affable sweeney fortunate leverage horntail apache
Show your final answer bracketed between <answer> and </answer>.
A: Let's think step by step. **Step 1:** List the words alphabetically.

- affable
- almost
- antic
- apache
- astute
- dandelion
- deadlock
- delphic
- execution
- fortunate
- horntail
- leverage
- levitate
- libertarian
- sanction
- scathe
- semitic
- storehouse
- sweeney
- unbeknownst

**Step 2:** Place the words between <answer> and </answer>.

<answer>affable
almost
antic
apache
astute
dandelion
deadlock
delphic
execution
fortunate
```

```
horntail
leverage
levitate
libertarian
sanction
scathe
semitic
storehouse
sweeney
unbeknownst </answer>
==

Q: Sort the following words alphabetically: List: burley bela arapaho bacteria bock
Show your final answer bracketed between <answer> and </answer>.
A: Let's think step by step. **Step 1:** List the words in alphabetical order.

arapaho
bacteria
bela
bock
burley

**Step 2:** Place the words between <answer> and </answer>.

<answer>arapaho bacteria bela bock burley</answer>
==
```

Listing 26: `word_sorting`: No IO + Mutation. Test accuracy = 81.5%

