# OpenReview forum: "Teach Better or Show Smarter? On Instructions and Exemplars in Automatic Prompt Optimization"
_NeurIPS.cc/2024/Conference — NeurIPS 2024 poster_

### Official Review · Reviewer_dxC3 · 2024-07-09

**Soundness:** 2
**Presentation:** 3
**Contribution:** 2
**Rating:** 5
**Confidence:** 4

**Summary:**

This paper aims to study the performance impact of instruction optimization (IO) and exemplar selection (ES) both isolation and combination, as the automatic prompt optimization literature mostly only focuses on either IO or ES. Its empirical results on a wide range of tasks (e.g., BBH and MMLU) show that optimizing ES may bring more performance gain than IO, and highlight the importance of studying ES and its combination with IO.

**Strengths:**

- Compared with previous works on APO, this work provides relatively comprehensive experiments to show combining IO with ES can further boost the APO performance.
- This work examines quite a number of combinations of different IO and ES approaches.

**Weaknesses:**

1. The paper only considers PaLM 2 and Gemini Pro (which are from the same source) as the target models. It is meaningful to see if the conclusion (i.e., ES outweighs IO) still holds for different target models (e.g., ChatGPT/GPT-4).

2. Some important settings are not explained. For example, in line 199, why ES is only performed from the exemplars self-generated by the target model, instead of choosing from $\mathcal{D}_{val}$. For $\mathcal{D}_c(I^*)$, only selecting the correctly predicted samples may not bring the highest performance, as some works show that wrong exemplars in ICL may be more valuable. In many IO works, they typically use more than 100 query evaluation budgets, instead of choosing $m=32$ naively in this paper.

3. The authors have the following objective in their paper:
>We aim to meet this by evaluating ES and IO in the context of APO by answering the following ... 2) How do we make the **optimal use** of the limited data and computational budget under the current APO framework?

Because of the synergy between ES and IO, the optimal use of ES and IO should exist at the *joint optimization* of ES and IO. However, the authors have only shown the performances of combining independently optimized instruction and optimized exemplars conditioned on that instruction. The optimality should be found when the exemplars and instructions are optimized together and the corresponding combinatorial optimization problem should cater to the space $\mathcal{I}\times\mathcal{E}^k$ rigorously. Intuitively, the optimized instruction should also depend on the given exemplars at each iteration. So, if the authors want to completely study how the effect of combining IO and ES, they should consider this joint optimization problem. Unfortunately, to the best extent, this paper only empirically shows (through brute force) that there may exist an optimal allocation of budgets $m_{IO}$ and $m_{ES}$ used for running the 'mixing-and-matching' strategy of ES and IO, which is far from the so-called **optimal use**.

**Questions:**

1. How did you choose the seed instruction? Do different seed instructions have different outcomes?

2. In Table 3, why ProTeGi is worse than (NO IO) if its optimization starts from the same seed instruction?

3. If you revert the process, that means you start ES first and run IO conditioned on the best ES you found, will this make the $\Delta IO$ greater than  $\Delta ES$?

**Limitations:**

Yes

---

> ### Author Rebuttal · Authors · 2024-08-07
>
> We thank the reviewer for their constructive and insightful comments. We’d like to refer the reviewer to our point-by-point response below and also the common response to all reviewers. We hope that in light of our response, the reviewer could consider revising their rating if they feel their concerns have been adequately addressed. **To fully address the reviewer's concerns, we have to utilize an additional comment to complete our response. We would like to thank the reviewer in advance for their time and patience while reading our reply.**
>
> > Only considers PaLM 2 and Gemini Pro (which are from the same source)
>
> First, we’d like to emphasize that PaLM 2 and Gemini are created by the same company but do not share the “same source,” which we assume refers to the training data mixture, algorithms, and/or other protocols [1, 2]. Second, **we do acknowledge the reviewer’s concerns and add new results on GPT and Gemini 1.5 and we’d refer the reviewer to the common response for details** but we also present the key insights below. With the new experiments added, we believe that our results now cover a comprehensive set of LLMs with a continuum of model caliber, especially with Gemini 1.5 [performing at or very close to the state of the art](https://arena.lmsys.org/). The new results are in near unanimous agreement with the key insights in the paper, and we believe this presents clear evidence that the patterns in the paper are not due to artifacts of individual models.
>
> ### gpt-3.5-turbo-0125*
>
>
> |         |    No ES | Random | Search | Δ ES      |
> |:-------:|---------:|-------:|--------|-----------|
> |  No IO  |     59.0 |   68.6 | 76.8   | **+17.8** |
> |   APE   |     63.0 |   68.9 | 78.4   | **+15.4** |
> | ProTeGi |     68.9 |   72.2 | 80.2   | **+11.3** |
> | Δ IO    | **+9.9** | +3.6   | +3.4   |           |
>
> ### gemini-1.5-flash-001
>
> |         |    No ES | Random | Nearest | Diversity | All  | Search | Δ ES     |
> |:-------:|---------:|-------:|---------|-----------|------|-------:|----------|
> |  No IO  |     75.1 |   80.0 | 81.7    | 81.5      | 80.4 |   83.3 | **+8.2** |
> |   APE   |     77.5 |   81.2 | 83.7    | 81.6      | 81.2 |   85.0 | **+7.5** |
> | ProTeGi |     80.4 |   82.4 | 82.6    | 82.3      | 83.5 |   84.5 | +4.1     |
> | Δ IO    | **+5.3** | +2.2   | +2.0    | +0.8      | +3.1 | +1.8   |          |
>
> > ES is only performed from the exemplars self-generated by the target model instead of choosing from D_val.
>
> We’d like to note that the tasks considered, which are extensively used in automatic prompt optimization papers, require intermediate logical/numerical reasoning from the LLM to reach the correct answer. We, however, **only assume D_val to contain the final answer labels but not the intermediate steps**: Obtaining additional annotations incurs extra costs and could invalidate our claims if ES outperforms IO due to access to these additional data sources, which are not required or used by any IO methods we surveyed. This setup is consistent with previous and concurrent works [1,2]. We’d also like to clarify that this is a common setup – virtually all instruction optimization methods require validation sets without rationales, and it is used in popular ES frameworks like DSPy [4]. Quoting the DSPy authors in Page 6:
>
>     In practice, we typically assume labels only for (at most) the program’s final output, not the intermediate steps. This label efficiency is critical for modularity: building a new pipeline in DSPy requires simply recompiling the new pipeline’s code, not annotating data specific to the new pipeline.
>
> Given this, **simply sampling (input, final answer) from D_val to as exemplars are known to be sub-optimal as they contain no intermediate answers**, and they essentially encourage direct answering over chain-of-thought reasoning [3] – the suboptimality of direct answering is precisely why chain-of-thought prompting has been tremendously helpful. Without annotated intermediate steps, using model-generated rationale is common in previous works like DSPy [4] and concurrent works like [5] (referred to as “reinforced ICL”). Given that the first objective of our study is to derive generalizable insights for more optimal methods that jointly consider instruction and exemplars in a canonical framework, we believe it makes sense to follow previous works in this aspect. Also, as mentioned, the model-generated responses are free side-products whenever we evaluate an instruction on the validation set, and re-using them as exemplars comes at no additional cost.
>
> > only selecting the correctly predicted samples may not bring the highest performance, as some works show that wrong exemplars in ICL may be more valuable.
>
> While we agree the wrong answers are valuable (e.g., they are already used as gradient signals for self-reflection in ProTeGi) and we are aware of works using wrong exemplars in ICL for contrastive learning [6], to our knowledge, **using correct demonstrations is still the mainstream approach** and it’s known that modern LLMs are particularly sensitive to the correctness of exemplars [7]. Furthermore, as mentioned, **selecting from the correctly predicted samples as exemplars is common** in previous and concurrent works: In [4], Sec. 3.3, Page 6, the BootstrapFewShot teleprompter “collect(s) demonstrations of each module (i.e., examples of its input–output behavior) *that collectively lead to valid output*”). In [5], Sec. 3, Page 5, the authors “select rationales *that obtain the correct final answer*, …, and arrange them into in-context examples” for reinforced ICL. Therefore, while we agree that better use of negative feedback is a very promising future direction, not using it in the present work should not be a weakness per se.

---

> ### Author Response · Authors · 2024-08-07
> **Continuation of rebuttal to Reviewer dxC3 (1/2)**
>
> >  In many IO works, they typically use more than 100 query evaluation budgets, instead of choosing 32 naively in this paper.
>
> While we are aware of works that use >100 query budgets, other works, like DSPy (which used “10-20 trials” – see Page 9 of [4]) also set a query budget similar to us – in this sense, using smaller evaluation budget for the given small amount of labeled data is not more “naive” than a larger budget, since it is just a matter of hyperparameter choice. In fact, we’d argue that performance at a tighter budget is more practically impactful and relevant as a larger evaluation budget leads to worse latency and higher cost; in most real-life setups, we cannot assume that the user can afford a long wait time and/or large budget – a key purported advantage of APO is that it eliminates the need for expensive prompt engineering, and it is important that we do not substitute an expensive process with another. Second, **we’d like to refer the reviewer to Appendix B.10, Page 40 in our paper, where we extend the evaluation budget to 100** and compare the simple two-stage optimization against PromptBreeder, which takes hundreds of iterations to converge. It is evident that the simple algorithm performs on par or better than the much more complicated PromptBreeder in most cases and converges faster.
>
> > How did you choose the seed instruction? Do different seed instructions have different outcomes?
>
> We chose “Let’s think step by step”, a well-known prompt introduced by [8] as the seed instruction. This seed instruction is commonly used at initialization in numerous prior works [9,10, inter alia].
>
> To answer the second part of the reviewer’s question, we run experiments with Gemini 1.0 models using different seed instructions. We run both APE and the two-stage IO-ES on it – please see the results below for aggregated results across 11 tasks. Apart from “Let’s think step by step,” we also tested two other initial prompts presented in previous works “Let’s think logically” also from [8] and “Let’s work this out in a step by step way to be sure we have the right answer,” from [9] – we can see that while IO-only can be significantly affected by the seed (>10% difference between best and worst seeds), two-stage IO-ES effectively shrank the gap to <3%. This should demonstrate our general robustness to initialization. We will include these discussions in the final version of the paper.
>
> |                                                        Initial instruction (I_0) |  IO-only (APE) | Two-stage IO-ES |
> |---------------------------------------------------------------------------------:|-----:|-----------|
> |                                              Let's think step by step (original) [8] | 63.0 | 78.5      |
> |                                              Let’s think about this logically [8] | 56.0 | 75.8      |
> | Let's work this out in a step by step way to be sure we have the right answer [9] | 66.7 | 77.5      |
>
> > Because of the synergy between ES and IO, the optimal use of ES and IO should exist at the joint optimization of ES and IO. However, the authors have only shown the performances of combining independently optimized instruction and optimized exemplars conditioned on that instruction…  Intuitively, the optimized instruction should also depend on the given exemplars at each iteration. So, if the authors want to completely study the effect of combining IO and ES, they should consider this joint optimization problem.
>
> Firstly, we thank the reviewer for their suggestion, and **we’d like to bring their attention to Algorithm 1 in Appendix B.8, which is exactly an example of how one may approach the joint optimization problem by allowing optimized exemplars and instructions to condition on each other and co-evolve**. Since we empirically found Algorithm 1 to achieve a comparable performance to the two-stage algorithm **(please see the common response for details)**, we emphasized the simpler algorithm for the sake of parsimony and ease of implementation. We agree that discussions on this should be made clearer in the main text, and we will amend the paper accordingly.

---

> ### Author Response · Authors · 2024-08-07
> **Continuation of rebuttal to Reviewer dxC3 (2/2)**
>
> Secondly, while we fully agree that ways to better integrate the joint optimization are important (we acknowledged this towards the end of Page 8 — “*we note that the presented way to combine optimization-based IO and ES is a proof of concept and room for future improvement can be vast: e.g., instead of two-stage optimization, it is also possible to optimize them jointly, interleavedly, or otherwise better exploit the inter-dependence between instructions and exemplars*”), **it is still an open research question**. Furthermore, **recognizing the importance of this joint optimization *problem* is the premise and the prerequisite to how to derive better *solutions***, the latter of which is where the reviewer’s concerns focused on. However, as argued in the paper and concurred by the other reviewers, we believe that the importance of the problem itself is under-appreciated, and exposing its benefit is a core contribution of ours. As discussed, most existing works only focus on either aspect but not both, and **our two-stage approach**, while simple and straightforward, has already led to significant improvement and is certainly **more optimal than instruction OR exemplar optimization only**. Thus, while we agree that there is always room for algorithmic improvement, and we will make this point even clearer in the paper, we also argue that our contributions should be assessed relative to the status quo in the literature.
>
> > In Table 3, why ProTeGi is worse than (NO IO) if its optimization starts from the same seed instruction?
>
> We followed the standard validation-test split and the test set is only evaluated once after selecting the instruction that led to the best validation accuracy. However, better validation performance does not always generalize to the test set (overfitting), and in this case the optimized instruction actually led to worse test performance.
>
> > If you revert the process, that means you start ES first and run IO conditioned on the best ES you found, will this make $\Delta IO$ larger than $\Delta ES$
>
> Firstly, we’d like to point out that the relative importance of ES and IO ($\Delta ES$ vs $\Delta IO$) has been studied both *in isolation* and *when combined* -- the ordering is only relevant when we combine them, but **the relative importance was first studied when each is performed separately**. With reference to Table 1-3, the results on the first row (“No IO”) are all obtained **without any instruction optimization**, and as we highlighted in bold on Page 6, optimized exemplars on top of seed instructions still outperform optimized instructions but with unoptimized exemplars, relative to the common reference point (i.e., seed instruction without exemplars).
>
> Second, as we detail in the common response, we also performed experiments with the opposite order where in Stage 1, we optimize the exemplars based on the seed instruction, and in Stage 2, we freeze the exemplars and optimize the instruction. We performed experiments on a subset of all tasks using both Gemini 1.0 and 1.5 models, using APE to optimize instructions and random search to optimize exemplars with a total budget of m=32 (the setup presented in Fig 6 of the paper). We find that using the swapped order (ES-IO) led to comparable aggregated results (on Gemini 1.0: IO first: 78.5%; ES first: 79.7%; on Gemini 1.5: IO first: 87.8%, ES first: 87.0%). Relative to ES only (Gemini 1.0: 78.2%; Gemini 1.5: 85.8%), it should be clear that **the bulk of the improvement is still attributable to ES, regardless of the ordering**.
>
> As mentioned above, another alternative is to use multi-stage interleaved optimization, as detailed in our response to the previous question. As we mentioned earlier, we highlighted the simple two-stage framework because, empirically, we found the interleaved algorithm to perform roughly similarly in practice.
>
>
> [1] Anil et al. (2023). Palm 2 technical report. arXiv preprint arXiv:2305.10403.
>
> [2] Google (2023). Gemini: a family of highly capable multimodal models. arXiv preprint arXiv:2312.11805.
>
> [3] Suzgun et al. (2022). Challenging big-bench tasks and whether chain-of-thought can solve them. arXiv preprint arXiv:2210.09261.
>
> [4] Khattab et al. (2024). DSPy: Compiling declarative language model calls into self-improving pipelines. ICLR.
>
> [5] Agarwal et al. (2024). Many-shot in-context learning. arXiv preprint arXiv:2404.11018.
>
> [6] Gao & Das (2024). Customizing Language Model Responses with Contrastive In-Context Learning. AAAI.
>
> [7] Wei et al. (2023). Larger language models do in-context learning differently. arXiv preprint arXiv:2303.03846.
>
> [8] Kojima et al. (2022). Large language models are zero-shot reasoners. NeurIPS.
>
> [9] Yang et al (2024). Large language models as optimizers. ICLR.
>
> [10] Wang et al. (2024). PromptAgent: Strategic planning with language models enables expert-level prompt optimization. ICLR.

---

> ### Author Response · Authors · 2024-08-12
> **We'd love to hear your opinion on the rebuttal**
>
> Dear reviewer dxC3,
>
> We thank you again for your time and discussion. Our impression with your original review is that you were concerned about 1) experiments on additional LLMs 2) some clarification about experimental setup (why exemplars are generated and the choice of evaluation budget) and 3) better joint optimization between IO and ES.
>
> To address your concerns, on 1), we conducted experiments on two new LLMs; on 2), we clarified that the setup we adopt is consistent to that of several existing works, and we have already included experiments of our algorithm under a higher budget in App. B.10 and 3) we pointed out that we have, in fact, already considered the joint optimization in App B.8. We hope our response has alleviated your concerns — as we are approaching the end of the discussion period, we’d be grateful if you could read our response and let us know if you have additional questions. Your insights and replies are highly appreciated. We’d also be grateful if you could reconsider the rejection rating if you feel the concerns have been sufficiently addressed.
>
> Thank you.
>
> Best,
>
> Authors

---

> > ### Comment · Reviewer_dxC3 · 2024-08-14
> >
> > Thanks for the response and additional results. Many of my concerns have been addressed. But for joint optimization, I think the author at least should propose a naive method to consider jointly optimizing prompts and exemplars (e.g., use a Bayesian optimization approach to model the prompt and exemplar together in a finite set and search for the promising combinations). As the evaluation and clarity of this paper remain an issue, I will increase my score to 5.

---

### Official Review · Reviewer_1T7N · 2024-07-11

**Soundness:** 2
**Presentation:** 3
**Contribution:** 3
**Rating:** 6
**Confidence:** 5

**Summary:**

The paper studies the instruction optimization and exemplar selection in auto prompt optimization. It has the following contributions:
1) Showing that optimizing ES improves significantly over IO and the importance of ES
2) Showing the synergy effect between ES and IO
3) Extensive experiments are conducted to validate the conclusions

**Strengths:**

1. This paper is well-written with good presentation
2. The formulation of ES and IO as optimization is clear
3. The experiments on ES and IO and a mix-matching of these two are clear and extensive
4. The idea of comparing ES and IO is novel to me

**Weaknesses:**

1. The authors have made a significant effort in summarizing the existing works, however, there are still some of the SOTA IO methods not mentioned:
- Connecting Large Language Models with Evolutionary Algorithms Yields Powerful Prompt Optimizers: https://arxiv.org/abs/2309.08532
- Use Your INSTINCT: INSTruction optimization for LLMs usIng Neural bandits Coupled with Transformers: https://arxiv.org/abs/2310.02905
2. Some of the baselines are mentioned in related work, however, the author did not compare with them (e.g., InstructZero). This can be important since these methods are SOTA methods of IO and may potentially produce much better results than the methods compared in this paper. It is true that the authors use simple methods for ES, however, do note that exemplars are natural data points (good data) while instructions are usually generated differently by different methods and hence a more careful design is needed.
3. The no IO is based on a very good instruction “let’s think step by step” while no ES is just no examplar at all. Therefore for fair comparison /drawing insights from the experiments, the improvement on ES needs to be compared to random (or at least nearest) to say which optimization is more important.
4. As the author mentioned in mix-matching section, the improvement of the R.S., Mutation in Table 1, 2 can be because of the extra computation/queires to LLM used in the search phase. How much does the delta in these tables are from this factor? And how would this affect the conclusion of comparing the importance of ES and IO? These are unclear from the current experiments.

**Questions:**

1. The experiments are all conducted on PaLM and Gemini which are from the same company and hence some similar underlying designs. Can the results generalize to other major LLMs, e.g., GPT-4, Claude, Llama.
2. How does the author implement Nearest? Is it finding the nearest for each test input? or for a whole validation dataset?
3. ES after IO is used to run the result in Table 1 and 2, have the authors tried IO after ES, will the result change?

**Limitations:**

The limitations are stated in the weakness. It mainly lies in the mismatch between experimental settings and the insights.

---

> ### Author Rebuttal · Authors · 2024-08-07
>
> We thank the reviewer for their constructive and positive comments about our work. Please see below for our point-by-point response, which we believe has addressed the reviewer’s concerns thoroughly. We hope that the reviewer could consider revising their score if they feel the same way. **To fully address the reviewer's concerns, we have to utilize an additional comment to complete our response. We would like to thank the reviewer in advance for their time and effort in reading through our reply.**
>
> > Additional related works, ..., Some of the baselines are mentioned in related work, however, the author did not compare with them.
>
> We thank the reviewer for bringing related works to our attention. We actually mentioned the first work (reference 14) in Sec 2, and we will discuss the second work in the camera-ready version. Due to the pace of advancements in this field, we hope the reviewer will sympathize with us, saying that it is not feasible to experimentally compare all methods proposed, and thus we opted to include the methods that are representative enough across the different genres. We believe that both works mentioned can be deemed to belong to the family of methods with evolutionary-like design spearheaded by APE and subsequently improved by better search algorithms and/or novel operators. The first work explicitly used an evolutionary framework, most notably with the new crossover operators. In contrast, the second work used an enhanced surrogate for Bayesian optimization, which is more sample-efficient than evolutionary algorithms and more effective than Gaussian processes. Still, on a high level, it has similar optimization goals. Both works, however, only optimize instructions and fix exemplars, e.g., on the footnote of Page 3 of the second work, the authors explicitly said they “*omitted the dependence on these factors (the set of exemplars) here as they are kept fixed for a task*”. This is exactly the kind of argument we mentioned on Line 56 about "focusing on one objective at a time." As discussed throughout the paper, we argue that while reasonable, this separate treatment risks missing out on (significant) benefits that we demonstrate from treating exemplars and instructions holistically -- thus, we believe that even if these works outperform IO baselines, they should still benefit from better exemplar optimization as their search space is essentially a subspace of the joint instruction-exemplar search space.
>
> > The no IO is based on a very good instruction “let’s think step by step” while no ES is just no examplar at all. The improvement on ES needs to be compared to random (or at least nearest) to say which optimization is more important.
>
> We agree that the reviewer’s suggestion should be the preferred practice in an ideal world where the importance of jointly optimizing exemplars and instructions is well-known. However, as we noted in Lines 133-135, many prominent instruction optimization methods still do not use exemplars (not even random sampling), even though this seems to be a “free lunch” in a way, based on our finding – this also puzzled us, and is one of the initial motivations for writing the paper, and we chose the respective reference point (“Let’s think step by step” and no exemplars) also by following previous works – for example, OPRO and PromptAgent made the exact same design decision.
>
> Secondly, based on Table 1-3, it is clear that **even if we set the reference point to Random, the difference between the *best ES method* and *random* still outweighs the performance difference across virtually any IO strategies**, which suggests that choosing *which* ES strategy is still the dominant decision; we can also observe that the gap between different IO methods shrinks further with random exemplars. Thus, we believe that the choice of the reference point will not significantly affect the paper’s conclusion. We nevertheless believe that the reviewer’s suggestion is valuable, and we will incorporate the above discussion into the final version of the paper.
>
> > How much does the delta in these tables are from this factor? And how would this affect the conclusion of comparing the importance of ES and IO? These are unclear from the current experiments.
>
> We’d like to refer the reviewer to Fig 6 and the paragraph starting on Line 304 – we emphasize that in Fig 6  (and Table 16 on Page 35), the total evaluation budget is the same as the ES/IO only (blue) cells in Table 1-2 to ensure fairness and to precisely remove the confounding factor of difference in compute mentioned by the reviewer. It is clear the benefit from mix-and-match persists, and the difference attributable to additional compute (i.e., the difference with the rightmost cell in the table of Fig 6) is minimal. In general, we used color coding to denote the optimization cost – RS and Mutation in Tables 1-2 the reviewer referred to are colored orange because they used double the amount of computation. The blue cells in the table in Fig 6 feature similar mix-and-match results but with halved budgets so that they consume the same budget as the other blue cells in Table 1-2.

---

> ### Author Response · Authors · 2024-08-07
> **Continuation of rebuttal to Reviewer 1T7N**
>
> > Additional LLMs
>
> We thank the reviewer for their suggestions. We have included additional results on GPT and Gemini 1.5 below. With the new experiments added, we believe that our results now cover a comprehensive set of LLMs with a continuum of model caliber, especially with the Gemini 1.5 model family [performing at or very close to the state of the art](https://arena.lmsys.org/) – made possible by the long-context ability, we also included an “All” column in Gemini 1.5 experiments, which includes all valid demonstrations without any selection. The new results are in near unanimous agreement with the key insights in the paper, and we believe this presents clear evidence that the patterns in the paper are not due to artifacts of individual models. We’d like to refer the reviewer to the common response for more details about the new experiments conducted.
>
> ### gpt-3.5-turbo-0125*
>
>
> |         |    No ES | Random | Search | Δ ES      |
> |:-------:|---------:|-------:|--------|-----------|
> |  No IO  |     59.0 |   68.6 | 76.8   | **+17.8** |
> |   APE   |     63.0 |   68.9 | 78.4   | **+15.4** |
> | ProTeGi |     68.9 |   72.2 | 80.2   | **+11.3** |
> | Δ IO    | **+9.9** | +3.6   | +3.4   |           |
>
> ### gemini-1.5-flash-001
>
> |         |    No ES | Random | Nearest | Diversity | All  | Search | Δ ES     |
> |:-------:|---------:|-------:|---------|-----------|------|-------:|----------|
> |  No IO  |     75.1 |   80.0 | 81.7    | 81.5      | 80.4 |   83.3 | **+8.2** |
> |   APE   |     77.5 |   81.2 | 83.7    | 81.6      | 81.2 |   85.0 | **+7.5** |
> | ProTeGi |     80.4 |   82.4 | 82.6    | 82.3      | 83.5 |   84.5 | +4.1     |
> | Δ IO    | **+5.3** | +2.2   | +2.0    | +0.8      | +3.1 | +1.8   |          |
>
> > Nearest implementation
>
> To implement nearest, we embed each test input using the [Gecko embedding model](https://arxiv.org/pdf/2403.20327) and retrieve the nearest exemplar in D_c based on input similarity, so each test input is likely to receive different exemplars as a result of this.
>
> > ES after IO is used to run the result in Table 1 and 2, have the authors tried IO after ES, will the result change?
>
> We thank the reviewer's suggestion. To answer this question, we also performed experiments with the opposite order where, in Stage 1, we optimize the exemplars based on the seed instruction, and in Stage 2, we freeze the exemplars and optimize the instruction. We performed experiments on a subset of all tasks using both Gemini 1.0 and 1.5 models, using APE to optimize instructions and random search to optimize exemplars with a total budget of m=32 (the setup presented in Fig 6 of the paper). We find that using the swapped order (ES-IO) led to comparable aggregated results (on Gemini 1.0: IO first: 78.5%; ES first: 79.7%; on Gemini 1.5: IO first: 87.8%, ES first: 87.0% – please refer to the common response for task-specific breakdown). We believe these new experiments provide sufficient evidence that the conclusion drawn in the paper is not due to any bias created by a specific ordering — we will incorporate a more comprehensive evaluation of this in the camera-ready copy.

---

> ### Author Response · Authors · 2024-08-12
> **We'd love to hear your opinion on the rebuttal**
>
> Dear reviewer 1T7N,
>
> We’d like to thank you once again for your review. From your original review, we believe that your concerns were about 1) additional IO techniques 2) concerns on baseline selection and 3) additional LLMs.
>
> To address your concerns, 1) we clarified the rationale for our IO selection. Furthermore, we observe that on challenging tasks requiring chain of thoughts, none of the additional related works (EvoPrompt*, INSTINCT and InstructZero) compared against ProTeGi, which from our experiments is a very strong baseline because it additionally utilizes self-reflection on top of better evolution-like paraphrasing that these works focused on; we showed that the benefit of ES persists even with ProTeGi – after all, we do not aim to propose a new instruction optimizer but rather to derive generalizable insights from representative methods. 2) We clarified that we are following prior works, and also that changing the ES reference to “random exemplars” does not materially affect the conclusion. On 3) we added experiments on two additional LLMs.
>
> As the discussion period ends, we would greatly appreciate it if you could review our response and share any further questions. Your insights are highly valued, and we look forward to your feedback. We’d also be grateful if you could consider increasing the score if you feel the concerns have been sufficiently addressed.
>
> Thank you!
>
> Best,
>
> Authors
>
> *EvoPrompt compared against ProTeGi on simple text classification tasks like SST-2 and AGNews only, which arguably aren’t commensurate with the caliber of modern LLMs (e.g., [1] achieved similar performance using the much older 355M RoBERTa with prompting, with differences of at most a few percentage points compared to EvoPrompt results with a 7B Alpaca [1]). On more challenging tasks, EvoPrompt, however, only compared against manual instruction (“Let’s think step by step”; referred to as “No IO” in our paper) and APE (App C.3, with ~3% improvement, which is smaller than the margin we showed with ES across all models (PaLM 2, Gemini 1.0 + 1.5, GPT-3.5) considered). The other works did not compare against ProTeGi in all experiments.
>
> [1] TEMPERA: Test-Time Prompting via Reinforcement Learning, ICLR 2023

---

> > ### Comment · Reviewer_1T7N · 2024-08-13
> > **Thanks for your response**
> >
> > Thanks for your additional results. Most of my concerns are addressed by your response. Therefore, I will increase my score. Please add these additional results to your revision.

---

> > > ### Author Response · Authors · 2024-08-13
> > >
> > > We'd like to thank the reviewer for engaging in the author feedback process and their valuable suggestions. As promised, we'll make sure to include additional results and discussions into the final version of the paper.

---

### Official Review · Reviewer_uqie · 2024-07-13

**Soundness:** 2
**Presentation:** 3
**Contribution:** 3
**Rating:** 5
**Confidence:** 3

**Summary:**

This paper provides a comprehensive investigation of instruction optimization and exemplar selection.

**Strengths:**

- The paper is very well motivated, and studies a very important problem, which is the relative importance of exemplar selection and instruction optimization. Therefore, the paper is likely to inspire ideas for future research in these ares.
- The paper is well written and well organized. I particularly like the way some results are presented, such as Table 1 and Figure 6. These results are very nicely organized and clearly presented, effectively conveying the intended insights.
- A particularly nice feature of the proposed algorithm is that it only uses the single validation set for both IO and ES.

**Weaknesses:**

- The performances achieved by the different combinations of algorithms are indeed good, however, in terms of the relative comparison of the importance of IO and ES, I think it is possible that an unfair advantage is given to ES. First of all, in all methods studied in this paper, IO is performed first followed by ES, this allows ES to adapt to an already optimized IO, while IO is not able to adapt to exemplars since it is performed first before any ES. In addition, only those exemplars on which the optimized instruction predicts correctly are used for ES, this also gives an advantage to ES since it uses information from an already optimized instruction, whereas IO is not able to exploit information from already optimized exemplars since IO is performed first. Therefore, it puts into question the reliability of a major conclusion drawn from the paper, i.e., ES is much more important than IO.
- As mentioned in the last paragraph of Section 2, to the best of my knowledge, there is indeed a general lack of works on joint IO and ES. I'd like to point the authors to two concurrent works along this line: https://arxiv.org/abs/2405.18369, https://arxiv.org/abs/2405.16122. The second work may be of more interest, because it is exactly using something similar to the combinatorial BO approach which is alluded to on lines 295-296.

**Questions:**

- The total number of prompt evaluations is selected as $m=32$. In my impression, this is in general a small number compared to the other related works on IO or ES. For example, OPRO uses more than 100 evaluations in most of its experiments. I wonder will the conclusions change if a larger $m$ is used?
- It has been widely reported that the ordering of the exemplars matters a lot. I wonder how do you determine the ordering?

**Limitations:**

A limitation is discussed in the Conclusion section.

---

> ### Author Rebuttal · Authors · 2024-08-07
>
> We thank the reviewer for their insightful and positive comments! Please see below for our response. We hope that in light of our response, the reviewer will consider revising their rating if they feel their concerns have been sufficiently addressed. **To fully address the reviewer's concerns, we have to utilize an additional comment to complete our response. We would like to thank the reviewer in advance for their time and effort in reading through our reply**.
>
> > Unfair advantage to ES
>
> Firstly, we’d like to point out that the relative importance of ES and IO has been studied both *in isolation* and *when combined*. While the reviewer is correct that we performed ES after IO *when they are combined* (we will respond to this in the next reply), **the relative importance was first studied when each is performed separately**. With reference to Table 1-3, the results on the first row (“No IO”) are all obtained **without any instruction optimization**, and as we highlighted in bold on Page 6, optimized exemplars on top of seed instructions still outperform optimized instructions but with unoptimized exemplars, relative to the common reference point (i.e., seed instruction without exemplars). This is our first finding that supports our Insight 2, and since the two modes of optimization are performed separately under the exact same setup (i.e., the same call budgets and the same number of labeled data) over the common starting point, there is no unfair advantage of one over another in this experiment when we derive their relative importance.
>
> > IO is performed first followed by ES, this allows ES to adapt to an already optimized IO, while IO is not able to adapt to exemplars since it is performed first before any ES.
>
> The main reason for us to place ES after IO in the paper was because we only assume labels, not annotated rationales, in the validation set to ensure fairness: obtaining additional annotations incurs extra costs and could invalidate our claims if ES outperforms IO due to access to these additional data sources, which are not required or used by the IO methods. To generate rationales, we reuse model outputs from instruction evaluations on the validation set — since the model outputs depend on individual instructions, it makes sense to optimize instructions first, followed by optimization on the exemplars conditioned on that instruction.
>
> **Following the reviewer’s suggestions, we also performed experiments with the opposite order** where in Stage 1, we optimize the exemplars based on the seed instruction, and in Stage 2, we freeze the exemplars and optimize the instruction. We performed experiments on a subset of all tasks using both Gemini 1.0 and 1.5 models, using APE to optimize instructions and random search to optimize exemplars with a total budget of m=32 (the setup presented in Fig 6 of the paper). We find that using the swapped order (ES-IO) led to comparable aggregated results (on Gemini 1.0: IO first: 78.5%; ES first: 79.7%; on Gemini 1.5: IO first: 87.8%, ES first: 87.0% – please refer to the common response for task-specific breakdown). We believe these new experiments provide sufficient evidence that the conclusion drawn in the paper is not due to any bias created by a specific ordering — we will incorporate a more comprehensive evaluation of this in the camera-ready copy.
>
> Beyond a simple two-stage design, we agree that there can be alternative ways to allow instructions to adapt to exemplars. In fact, we have already explored this in Appendix B.8, Page 36, where in Algorithm 1, we interleaved instruction and exemplar optimization, and in this case, **optimized instructions and optimized exemplars can adapt to each other**. As mentioned in Line 1087, this algorithm achieved a comparable performance compared to the two-stage approach, so we presented the latter in the main text because we believe that simplicity should be favored when the performance is similar. Still, we agree that better integration of exemplar and instruction optimization can be an important open research question, as we also mentioned towards the end of Page 8. Appendix B.8. We will clarify this point better in the final version of the paper.
>
> > only those exemplars on which the optimized instruction predicts correctly are used for ES.
>
> As discussed, we’d like to emphasize that 1) the exemplars are generated from the validation set, *whose labels are required anyway* for performance estimation, and selecting the correct predictions is a simple filtering step that requires neither additional data nor computation. Instruction optimizers utilize the validation set in the same way by using the validation accuracy as the optimization objective – to compute this metric; we also have to count the number of correctly predicted samples. ES is essentially doing the same, except that rather than simply counting and aggregating the instances, it also traces back to the individual samples that make up the metric – in this sense, using the correctly predicted outputs as exemplars does not create unfairness per se. 2) Bootstrapping correct predictions is also commonly used in previous and contemporary works [1, 2]: In DSPy [1], Section 3.3, Page 6, the BootstrapFewShot teleprompter “collect(s) demonstrations of each module (i.e., examples of its input-output behavior) *that collectively lead to valid output*”). In [2], Section 3, Page 5, the authors “select rationales *that obtain the correct final answer*, …, and arrange them into in-context examples” for reinforced ICL. Thus, while we agree that better use of negative feedback can be a very promising future direction, not using it in the present work should not be a weakness of our analysis.

---

> ### Author Response · Authors · 2024-08-07
> **Continuation of rebuttal to Reviewer uqie**
>
> > Additional related works
>
> We thank the reviewer for bringing related works to our attention. We agree the second paper is particularly relevant. As the reviewer aptly pointed out, the method is exactly a “combinatorial Bayesian optimization” approach mentioned by us. The authors also motivated their work with the importance of exemplar selection. They proposed a joint ES-IO algorithm, and we are glad that contemporary research on prompt optimization has put more effort into improving both exemplars and instructions holistically, which is exactly the practice championed in our paper. The key difference is that the key contribution of the mentioned paper is to propose a specific method (EASE). In contrast, we aim to obtain general insights on the best practices for prompt design by drawing upon the literature. We believe the contributions are complementary to each other (for example, we believe it is possible that the improvement brought by EASE over the pure instruction optimization methods shown in the paper could be partly explained by the exemplar optimization it performed). We will incorporate more detailed discussions of these related works in the final version of the paper.
>
> > The choice of m, and whether a larger m will affect the conclusion.
>
> We chose a relatively small m because a tighter budget is more practically impactful and relevant as a larger evaluation budget leads to worse latency and higher cost; in most real-life setups, we cannot assume that the user can afford a long wait time and/or large budget – a key purported advantage of APO is that it eliminates the need for expensive prompt engineering, and it is important that we do not substitute an expensive process with another. Second, **we’d like to refer the reviewer to Appendix B.10, Page 40 in our paper, where we perform some preliminary experiments where we increased the evaluation budget to 100** to compare against PromptBreeder, which takes hundreds of iterations to converge. It is evident that the simple algorithm performs on par or better than the much more complicated PromptBreeder in most cases and converges faster.
>
> > Ordering of exemplars
>
> We thank the reviewer for the suggestion – it is indeed well-known that the ordering of the exemplars can affect downstream performance. However, our high-level impression is that for the frontier models, the effect of ordering seems to be secondary to the choice of exemplars. Secondly, for a data-driven exemplar optimization algorithm, the order is implicitly optimized at the same time – for example, in random search, each iteration draws an *ordered* tuple of demonstrations, and an argmax over all draws is taken for testing – a data-driven optimization method only considers performance metric but is agnostic to what contributed to it, and it is possible that the optimality of the chosen demo set is partly attributable to order.
>
> [1] Khattab et al. (2024). DSPy: Compiling declarative language model calls into self-improving pipelines. ICLR.
>
> [2] Agarwal et al. (2024). Many-shot in-context learning. arXiv preprint arXiv:2404.11018.

---

> ### Author Response · Authors · 2024-08-12
> **We'd love to hear your opinion on the rebuttal**
>
> Dear reviewer uqie,
>
> We thank you once again for your time. From your original review, your main concern was about the fairness of comparison between IO and ES.
>
> To address your concerns, we 1) clarified that we first conducted IO and ES separately over the same reference and 2) conducted additional experiments in a swapped order (i.e., ES before IO). As we are approaching the end of the discussion period, we’d be grateful if you could read our response and let us know if you have additional questions? Your comments have been tremendously helpful in improving our paper, and we’d like to hear your opinion on the rebuttal. We’d also be grateful if you could consider increasing the score if you feel the concerns have been sufficiently addressed.
>
> Thank you!
>
> Best,
>
> Authors

---

### Official Review · Reviewer_Hw9e · 2024-07-18

**Soundness:** 3
**Presentation:** 4
**Contribution:** 2
**Rating:** 5
**Confidence:** 3

**Summary:**

This paper profoundly discusses how the deisgn of Automatic Prompt Optimization (APO) methods influence the performance of current instruction-following pre-trained large language models (LLMs). Concretely, the authors first systematically summarize and compartmentalize current APO methods into two lines: instruction optimziation (IO) and exempler sleection (ES), with their mathematical definitions. Then a series thorough and incremental experiments are carried out, of which primary discoveries includes: 1) reusing model-generated data (obtained from validation phase) as exempler shows performance improvement against standalone IO methods, with better efficiency. 2) combining IO and ES together would bring "1+1>2" performance improvement. Under these experimental results, the authors claim that studying ES methods should be put as top priority of APO studies.

**Strengths:**

1. This paper is very well-writen. The related works are appropriately summarized. The mathematical definition of APO, IO and ES are crystal clear and hence helps identify the difference between IO and ES. The experimental results are carefully arranged and presented.
2. The findings, though obtained from limited tasks and models, are still well supported by the current experimental results. It is very meaningful for the APO community, and deserves more attention and further investigation and development.

**Weaknesses:**

After reading this paper from top to the bottom, I found my concerns revolve around limited evaluation: I would appreciate the authors if they could at least conduct an additional experiment on GPT-3.5 Turbo to reinforce the convincingness of all findings, since GPT models hold quite some users. To summarize, although I fully understand that ideally comprehensive evaluation experiment is never easy, the above aspects deserve to be investigated, at least. Besides, the contribution of this paper mainly involves benchmarking and concluding meaningful findings, which in my view, should be more suitable for Benchmark Track of NeurIPS.

**Questions:**

see Weaknesses.

**Limitations:**

The authors have summarized the limitations of this work in Conclusion of the paper, which are objective. I strongly suggest the authors include more datasets (e.g., multimodal tasks) and models (e.g., GPT series) in their future works as they promised.

---

> ### Author Rebuttal · Authors · 2024-08-07
>
> We thank the reviewer for their constructive feedback, which also included clear, actionable items (e.g., GPT evaluations). We’d be grateful if the reviewer could review our response below. We also hope they could consider revising their rating if they feel their concerns have been adequately addressed.
>
> > Limited evaluations.
>
> We thank the reviewer for their suggestions. We have included additional results on GPT, as the reviewer suggested, and Gemini 1.5 below. With the new experiments added, we believe that our results now cover a comprehensive set of LLMs with a continuum of model caliber, especially with the Gemini 1.5 model family  [performing at or very close to the state of the art](https://arena.lmsys.org/) – made possible by the long-context ability, we also included an “All” column in Gemini 1.5 experiments, which includes all valid demonstrations without any selection. The new results are in near unanimous agreement with the key insights in the paper, and we believe this presents clear evidence that the patterns in the paper are not due to artifacts of individual models. We’d like to refer the reviewer to the common response for more details about the new experiments conducted.
>
> ### gpt-3.5-turbo-0125*
>
> |         |    No ES | Random | Search | Δ ES      |
> |:-------:|---------:|-------:|--------|-----------|
> |  No IO  |     59.0 |   68.6 | 76.8   | **+17.8** |
> |   APE   |     63.0 |   68.9 | 78.4   | **+15.4** |
> | ProTeGi |     68.9 |   72.2 | 80.2   | **+11.3** |
> | Δ IO    | **+9.9** | +3.6   | +3.4   |           |
>
> ### gemini-1.5-flash-001
>
> |         |    No ES | Random | Nearest | Diversity | All  | Search | Δ ES     |
> |:-------:|---------:|-------:|---------|-----------|------|-------:|----------|
> |  No IO  |     75.1 |   80.0 | 81.7    | 81.5      | 80.4 |   83.3 | **+8.2** |
> |   APE   |     77.5 |   81.2 | 83.7    | 81.6      | 81.2 |   85.0 | **+7.5** |
> | ProTeGi |     80.4 |   82.4 | 82.6    | 82.3      | 83.5 |   84.5 | +4.1     |
> | Δ IO    | **+5.3** | +2.2   | +2.0    | +0.8      | +3.1 | +1.8   |          |
>
>
> > “the contribution of this paper mainly involves benchmarking and concluding meaningful findings, which in my view, should be more suitable for Benchmark Track of NeurIPS.”
>
> We thank the reviewer for appreciating our extensive benchmarking and meaningful findings. We believe that the main track is more suitable, as we have not proposed a new benchmark (we reused existing datasets) in this study, but our goal is to shed light on the best practices for prompt design, considering the optimization holistically. We also proposed a simple and effective framework by optimizing both exemplars and instructions that show significant improvements in downstream task performance.
>
> > Additional datasets
>
> We thank the reviewer's suggestions. As mentioned in the conclusion, we chose BIG-Bench and MMLU because, as noted on line 165, they are widely used in contemporary automatic prompt optimization (APO) papers. These datasets cover numerous challenging tasks, including reasoning and information-seeking types, and their performance improvements correlate with human-in-the-loop settings like LMSys. Our first objective, as stated on line 62, is to compare performance gains using IO and ES methods on a fairground. Thus, these datasets are ideal since the original authors also use them to showcase their methods. Having said that, we agree and reiterate that incorporating more open-ended and realistic tasks is crucial for our study and the broader APO field.

---

> > ### Comment · Reviewer_Hw9e · 2024-08-09
> >
> > Thank you for the response. All my concerns are addressed more or less. I still insist that this paper serves more as a benchmark track paper but also agree with the authors that the contributions are also suitable for main track. I raise my score to 5.

---

> > > ### Author Response · Authors · 2024-08-13
> > >
> > > We'd like to thank the reviewer again for engaging in the rebuttal and appreciating our contributions -- we'll make sure to incorporate the reviewer's suggestions into the final version of the paper. As we are approaching the end of author discussion period, we are more than happy to answer any further questions or discuss any additional improvements that would potentially make the reviewer evaluate our work even more positively.

---

### Official Review · Reviewer_1DvC · 2024-07-22

**Soundness:** 4
**Presentation:** 4
**Contribution:** 3
**Rating:** 7
**Confidence:** 3

**Summary:**

This work focus on evalution and comparison study of automatic prompt optimization (APO) methods. These methods are broadly categorized into instruction optimization (IO) and exemplar selection (ES). This paper seeks to bridge the gap between these two methods by comprehensively comparing the performance of representative IO and ES techniques both isolation and combination. This work found that how to select examplars can outweigh how to optimize instructions.

**Strengths:**

1. The presentation of this work is good. From the textual part, the authors clearly present the definition and representative methods of IO and ES. From those graphical presentation, they clearly present the comparison results of different methods.
2. The setting of experimental study is convincing. The sequence from 'initial instruction + No exemplars' -> 'Optimized instruction + No examplars' -> Optimized instruction + Random exemplars' -> 'Initial instruction + Optimized exemplars' -> 'Optimized instruction + Optimized exemplars' make the comparison results clear and convicing.
3. The finding of this work is insightful, this work found that ES is more important than IO which a bit do not fit in with the current direction of the study. So, this work may help to provide a reference for future research.

**Weaknesses:**

1. In the experimental study part, this work only use two kinds of LLMs, PaLM 2 and Gemini, which is not quite adequate.
2. The evaluation criteria are a bit simple, only considering the acc rate. For engneering requirements, how many efforts (e.g. number of tokens; expense of the APO process; how much level of ) will be required during the prompt optimization is also important. Other reasonable metrics maybe also of help.

**Questions:**

1. What's the difference between methods from IO and ES in terms of the cost. Are there any differences in statistical characteristics between these two categories?

**Limitations:**

See weakness and question part.

---

> ### Author Rebuttal · Authors · 2024-08-07
>
> We thank the reviewer for their positive feedback! Please see below for our response to their questions and concerns.
>
> > More LLM choices
>
> We thank the reviewer for their suggestions. We have included additional results on GPT and Gemini 1.5 below. With the new experiments added, we believe that our results now cover a comprehensive set of LLMs with a continuum of model caliber, especially with Gemini 1.5 model family [performing at or very close to the state of the art](https://arena.lmsys.org/) – made possible by the long-context ability, we also included an “All” column in Gemini 1.5 experiments, which includes all valid demonstrations without any selection. The new results are in near unanimous agreement with the key insights in the paper, and we believe this presents clear evidence that the patterns in the paper are not due to artifacts of individual models. We’d like to refer the reviewer to the common response for more details about the new experiments conducted.
>
> ### gpt-3.5-turbo-0125*
>
> |         |    No ES | Random | Search | Δ ES      |
> |:-------:|---------:|-------:|--------|-----------|
> |  No IO  |     59.0 |   68.6 | 76.8   | **+17.8** |
> |   APE   |     63.0 |   68.9 | 78.4   | **+15.4** |
> | ProTeGi |     68.9 |   72.2 | 80.2   | **+11.3** |
> | Δ IO    | **+9.9** | +3.6   | +3.4   |           |
>
> ### gemini-1.5-flash-001
>
> |         |    No ES | Random | Nearest | Diversity | All  | Search | Δ ES     |
> |:-------:|---------:|-------:|---------|-----------|------|-------:|----------|
> |  No IO  |     75.1 |   80.0 | 81.7    | 81.5      | 80.4 |   83.3 | **+8.2** |
> |   APE   |     77.5 |   81.2 | 83.7    | 81.6      | 81.2 |   85.0 | **+7.5** |
> | ProTeGi |     80.4 |   82.4 | 82.6    | 82.3      | 83.5 |   84.5 | +4.1     |
> | Δ IO    | **+5.3** | +2.2   | +2.0    | +0.8      | +3.1 | +1.8   |          |
>
> > Additional metrics and evaluation criteria, ..., What's the difference between methods from IO and ES in terms of the cost. Are there any differences in statistical characteristics between these two categories?
>
> We mainly used accuracy because **we controlled the cost and ensured the different IO and ES methods have approximate fairness in terms of the number of iterations** (each iteration involves the evaluation of a prompt on the validation set) to ensure the validity of any conclusion drawn from performance comparison (see caption of Table 1; note that instruction optimization methods also require calls to an optimizer model that are typically more capable and expensive; exemplar optimization requires no such models) – we will clarify this further in the table captions when we get a chance in revising the paper. For example, in all result tables like Tables 1-3, we have used color-coding to highlight the approximate computational costs in terms of the number of LLM calls (in Fig 6, we further show that the improvement from orange cells which use twice as many evaluations does not simply come from the additional compute). We used the number of iterations (each iteration corresponds to a full evaluation of a prompt on the validation set) as the main cost metric – most previous works we cited in Sec 2 such as OPRO and PromptAgent used the same metric.

---

> > ### Comment · Reviewer_1DvC · 2024-08-09
> > **Response to Authors**
> >
> > Thanks for your reponse and all of my questions have been addressed. Good work and I will maintain my score.

---

> ### Author Response · Authors · 2024-08-13
>
> We thank the reviewer again for their time and effort engaging in the rebuttal, and we will make sure to incorporate the reviewer's suggestions into the final version of our paper.

---

### Official Review · Reviewer_JLax · 2024-07-30

**Soundness:** 3
**Presentation:** 4
**Contribution:** 3
**Rating:** 7
**Confidence:** 4

**Summary:**

This work studies automatic prompt optimization, seeking to compare and to connect the literatures on instruction optimization and on example selection. They find that existing approaches for using LMs to self-generate examples and selecting between them can outperform the best existing instruction optimizers, but that combining the two is effective. They use this to recommend larger emphasis on few-shot examples in future prompt optimization work.

**Strengths:**

1. The work asks a timely question that is formulated well, i.e. about the "relative importance and performance impact of ES and IO, both in isolation and when combined together". This challenges overemphasis on instruction optimization (IO) in plenty of recent work.

2. In the experimental evaluation, the authors consider a wide and representative set of four Instruction Optimization methods from APE to OPRO, and take substantial steps to equalize their budgets for comparison.

3. The authors do an excellent job at communicating their findings from a large number of comparisons in the form of several concrete and overall well-argued/supported Insights.

**Weaknesses:**

1. The authors evaluate on BIG-Bench and MMLU tasks, which to my knowledge are extremely limited in scope. Most are narrow reasoning puzzles or manipulation tasks, e.g. multiple choice, and may not reflect the type or complexity of typical prompts people use in the increasingly elaborate open-ended LM systems out there. This has an effect on the extent of "self-generated" examples, which in this work are reduced to just chain of thought reasoning if I understand correctly, where all "inputs" and "labels" (in this case final answers) are known in advance, rather than trajectories of actual input/output labels for prompts per example. How would such open-endedness interact with the findings?

2. The authors already find very compelling results with ES, but the scope of existing example-based optimizers in the cited work like DSPy seem wider in terms of the search/optimization method and in the generation approach. For the latter, the authors employ a large LM to optimize the instructions for smaller LMs, but related work permits similarly a large LM (or the same LM but in already example-optimized form) to generate the examples for small LMs. As a result of this asymmetry, the effect of the source of instruction or example optimization is not explored or isolated. This seems unlikely to affect the results direction, but it may affect their magnitude substantially, perhaps making IO+ES less attractive than just ES.

**Questions:**

1. This is not a weakness of the present paper, but in the manner described here APE and OPRO are presented rather similarly in Sec 3.1. Overall, this raises the question of how the authors ensured their implementations are faithful to the original algorithms while being fair to all methods within a new test environment.

2. I appreciate the note in lines 199-201 about self-generation. The way this is presented suggests the authors thing it's important, but it's rather buried now and hard to concretely understand without an example or further elaboration, especially because the tasks considered in this work are rather "simple", e.g. not clearly multi-step. In addition, the term Example Selection (ES), which is not particularly universal in the literature, is perhaps setting this up for confusion, since selection directly assumes the presence of a set from which to select.

---

> ### Author Rebuttal · Authors · 2024-08-07
>
> We thank the reviewer for their insightful and positive feedback! Please see below for our point-by-point response to the reviewer’s specific comments.
>
> > Limitations of BIG-Bench and MMLU and more open-ended/realistic tasks.
>
> We thank the reviewer for their suggestions! The primary reason we picked BIG-Bench and MMLU is that, as mentioned on line 165, these benchmarks (or similar tasks) are still heavily featured in many, if not most, contemporary automatic prompt optimization (APO) papers. These datasets encompass a large number of different challenging tasks, including reasoning-heavy or information-seeking types, and performance improvement in them still seems to be highly correlated with human-in-the-loop arenas like LMSys. These tasks are also still used to demonstrate LLM’s general problem-solving caliber, and both are featured in Gemini and GPT technical reports. As our first objective, as mentioned on line 62, is to “*compare the performance gain by … IO and ES methods on the fairground*”, we felt that using these datasets, which are also the ones the original authors often rely on to show the effectiveness of their proposed methods, is fittest for this purpose. We also argue that our evaluation is already more comprehensive than most existing APO works, which often choose small subsets of these tasks that often differ from paper to paper, which further confounds fair comparison. Nevertheless, we agree with the reviewer that using more open-ended tasks with more complicated and realistic tasks is important both for our specific study and for the broader APO field, which, as mentioned earlier, still relies on the aforementioned tasks to a considerable extent. We will endeavor to expand our analysis on more open-ended and/or multi-hop problems and will add a comment on the reviewer’s suggestion in Conclusion.
>
> > Scope of example-based optimizers in existing works like DSPy
>
> We thank the reviewer for pointing this out – existing works like DSPy indeed introduced exemplar optimization options that are more advanced than random search, such as TPE-based Bayesian optimization. We opted for the simpler methods in the present work for two reasons: 1) both our preliminary results and the results reported by DSPy authors in concurrent works like Table 2 of [1] show random search is still a remarkably strong baseline; in many cases, the more advanced optimizers perform similarly to it, and 2) while valuable, the comparison between different exemplar optimization methods is not a key focus of the present study – indeed, a key observation is that exemplar optimization as simple as random search can outperform and synergize with much more advanced instruction optimization methods, and using more advanced exemplar optimization could only strengthen rather than undermine this finding. We will further clarify the above discussions in the final draft of the paper.
>
> [1] Opsahl-Ong et al. (2024). Optimizing Instructions and Demonstrations for Multi-Stage Language Model Programs.
> > Asymmetry due to the use of large LMs to optimize instructions and the effect on the relative performance.
>
> We agree that this is an important and plausible explanation for the stronger performance of methods like ProTeGi, which rely on reasoning-based self-critique, an ability that can be strongly dependent on the general caliber of the optimizer model. Nevertheless, the reported synergy should not entirely come from this asymmetry. Concretely, Figs 5 and 6 show the benefit is also present when we combine APE with exemplar optimization. Unlike ProTeGi, APE generates new instructions only by paraphrasing old instructions without altering their semantic meaning (e.g., “Let’s think step by step” -> “Let’s follow a step-by-step process”). This operation arguably does not require a particularly strong model. Yet, we can still observe improvement – this, in our opinion, alludes to the presence of certain inherent synergy. We will nevertheless incorporate the reviewer’s remarks in our discussion under “Insight 3”; additionally, we also believe that the reviewer’s suggestion can be of independent interest as an effective, low-cost distillation from a powerful model to a smaller model through prompts, which are fixed at inference.
>
> > Descriptions of APE vs OPRO and faithfulness to original algorithms
>
> We hope the reviewer can sympathize with our rather condensed descriptions of the algorithms due to the space constraint – indeed, the key difference between APE and OPRO is that whereas APE explicitly requests paraphrasing of the better-performing instructions, OPRO treats the LLM as a black box and let the LLM to determine its actions for the next generation, and we will highlight this difference better in the final version of the paper. Regarding the remark on the faithfulness of the algorithms, we note that where possible, we adapt the official implementation. Some changes are inevitable, including but not limited to parity of the number of generations, eval budgets, and initialization across the methods. These are to ensure broad fairness across the methods, and we believe should not lead to a significant discrepancy in findings.
>
> > Note about self-generation (Line 199-201)
>
> As discussed in the common response, we do not assume annotated rationales in the validation set, which has made self-generated exemplars important, and we note that such a design decision has also been taken in previous works like DSPy and many-shot ICL. In the final version of the paper, we will provide a concrete example and incorporate the discussions in the common response to make this point clearer. We will also clarify further what we meant by “exemplar selection” to strengthen the point that in tasks where reasoning is required, exemplars are at least partially bootstrapped from the model and do not entail a simple selection from a fixed set. We thank the reviewer for the suggestions, which will likely improve the clarity of the paper further.

---

> > ### Comment · Reviewer_JLax · 2024-08-07
> >
> > Thank you for the detailed response. It helps me maintain my high score of this work.

---

> ### Author Response · Authors · 2024-08-13
> **Thank you**
>
> Thank you for your constructive feedback and positive assessment of our work! We will make sure to incorporate the reviewer's suggestions into the final version of the paper.

---

### Author Rebuttal · Authors · 2024-08-07

We thank all reviewers for their time and feedback! In the common response below, we’d like to provide in-depth answers to some of the questions asked by multiple reviewers.

## Additional LLMs (Reviewers 1DvC, Hw9e, dxC3)

Multiple reviewers asked for evaluations on additional LLMs. We acknowledge this feedback and have validated the key findings with GPT (gpt-3.5-turbo-0125) and Gemini 1.5 (gemini-1.5-flash-001). Please see the table below for aggregated results; for task-specific breakdown similar to Fig 4, we refer the reviewer to Fig. S1 (GPT) and S2 (Gemini 1.5) in the uploaded pdf.

### gpt-3.5-turbo-0125*

|         |    No ES | Random | Search | Δ ES      |
|:-------:|---------:|-------:|--------|-----------|
|  No IO  |     59.0 |   68.6 | 76.8   | **+17.8** |
|   APE   |     63.0 |   68.9 | 78.4   | **+15.4** |
| ProTeGi |     68.9 |   72.2 | 80.2   | **+11.3** |
| Δ IO    | **+9.9** | +3.6   | +3.4   |           |

### gemini-1.5-flash-001

|         |    No ES | Random | Nearest | Diversity | All  | Search | Δ ES     |
|:-------:|---------:|-------:|---------|-----------|------|-------:|----------|
|  No IO  |     75.1 |   80.0 | 81.7    | 81.5      | 80.4 |   83.3 | **+8.2** |
|   APE   |     77.5 |   81.2 | 83.7    | 81.6      | 81.2 |   85.0 | **+7.5** |
| ProTeGi |     80.4 |   82.4 | 82.6    | 82.3      | 83.5 |   84.5 | +4.1     |
| Δ IO    | **+5.3** | +2.2   | +2.0    | +0.8      | +3.1 | +1.8   |          |

*Due to rate limits and time constraints, for these sets of newly added experiments, we tested a subset of 11 tasks, but we can already observe patterns consistent with our claims in the paper. We will endeavor to include a full set of results in the camera-ready version. The Gemini 1.5 results are already full sets and are comparable to Tables 1-2 in the paper.

With the new experiments added, we believe that our results now cover a comprehensive set of LLMs with a continuum of model caliber, especially with the Gemini 1.5 model family [performing at or very close to the state of the art](https://arena.lmsys.org/) – made possible by the long-context ability, we also included an “All” column in Gemini 1.5 experiments, which includes all valid demonstrations without any selection. The new results are in near unanimous agreement with the key insights in the paper, and we believe this presents clear evidence that the patterns in the paper are not due to artifacts of individual models.

## Ordering of ES and IO or better joint optimization techniques (Reviewers Uqie, 1T7N, dxC3)

Several reviewers asked why we had exemplar selection after instruction optimization and whether alternative ordering would be possible. The main reason for us to place ES after IO in the paper was that we only assume labels, not annotated rationales, in the validation set to ensure fairness: obtaining additional annotations incurs extra costs and could invalidate our claims if ES outperforms IO due to access to these additional data sources, which are not required or used by the IO methods. To generate rationales, we reuse model outputs from instruction evaluations on the validation set — since the model outputs depend on individual instructions, it makes sense to optimize instructions first, followed by optimization on the exemplars conditioned on that instruction.

Following the reviewer’s suggestions, we also performed experiments in the opposite order where, in Stage 1, we optimized the exemplars based on the seed instruction, and in Stage 2, we froze the exemplars and optimized the instruction. We performed experiments on a subset of all tasks using both Gemini 1.0 and 1.5 models, using APE to optimize instructions and random search to optimize exemplars with a total budget of m=32 (the setup presented in Fig 6 of the paper). We find that using the swapped order (ES-IO) led to comparable aggregated results. We refer the reviewers to detailed [Tables S1 and S2](https://openreview.net/forum?id=IdtoJVWVnX&noteId=cw2sfTOORc) in the additional comment below, which we believe provides sufficient evidence that the conclusion drawn in the paper is not due to any bias created by a specific ordering — we will incorporate a more comprehensive evaluation on this in the camera-ready copy.

On a related note, beyond a simple two-stage approach, as Reviewer dxC3 aptly suggested, a better strategy in principle is to **jointly optimize instructions and exemplars and allow them to co-evolve by alternating between instruction and exemplar optimization successively in a multi-stage manner. We’d like to note that we already considered a possible implementation of it in Appendix B.8**. As mentioned in Line 1087, we found the algorithm (Algorithm 1) achieved a comparable performance (74.6%) compared to the two-stage approach (74.5%, which corresponds to the m_io=16, m_es=16 result in Fig. 6 – see [Table S3](https://openreview.net/forum?id=IdtoJVWVnX&noteId=cw2sfTOORc) for task-specific breakdown) on PaLM 2, so we presented the latter in the main text based on the parsimony principle of Occam’s razor that simplicity should be favored when the performance is similar.  Nonetheless, we agree that better-integrating exemplar and instruction optimization is an important, open research question, as mentioned towards the end of page 8.

---

### Author Response · Authors · 2024-08-07
**Detailed tables of the overall response**

**Table S1**: Effect of different IO/ES order on gemini-1.0-pro-002
|        gemini-1.0-pro-002       | IO only | ES only | Combined, *IO first* | Combined, *ES First* |
|:-------------------------------:|---------|---------|:------------------:|-------------------:|
|             boolean_expressions |  79.50% |  91.00% |             90.50% |             90.50% |
|                causal_judgement |  54.67% |  59.33% |             66.00% |             66.00% |
|              date_understanding |  73.50% |  78.50% |             79.50% |             74.50% |
|                geometric_shapes |  33.00% |  71.00% |             67.50% |             84.50% |
| logical_deduction_seven_objects |  31.00% |  50.00% |             47.50% |             51.00% |
| logical_deduction_three_objects |  49.00% |  76.50% |             78.00% |             79.00% |
|        multistep_arithmetic_two |  77.50% |  84.00% |             79.50% |             81.00% |
|                        navigate |  58.00% |  85.50% |             89.50% |             89.00% |
|                 object_counting |  78.50% |  95.50% |             94.50% |             94.00% |
|             penguins_in_a_table |  78.63% |  84.62% |             93.16% |             84.62% |
|              temporal_sequences |  79.50% |  84.00% |             77.50% |             82.50% |
| (Average)                       |  62.98% |  78.18% |             78.47% |             79.69% |

**Table S2**: Effect of different IO/ES order on gemini-1.5-flash-001
| gemini-1.5-flash-001            | IO only | ES only | Combined, *IO first* | Combined, *ES First* |
|---------------------------------|---------|---------|:------------------:|-------------------:|
|             boolean_expressions |  90.50% |  99.50% |             96.00% |             99.00% |
|                causal_judgement |  68.00% |  67.33% |             67.33% |             63.33% |
|              date_understanding |  80.00% |  84.50% |             84.00% |             83.00% |
|                geometric_shapes |  54.50% |  82.00% |             90.50% |             81.00% |
| logical_deduction_seven_objects |  50.00% |  70.00% |             80.50% |             72.00% |
| logical_deduction_three_objects |  91.00% |  89.50% |             92.50% |             93.50% |
|        multistep_arithmetic_two |  91.50% |  91.50% |             92.50% |             91.50% |
|                        navigate |  66.00% |  69.00% |             74.00% |             85.00% |
|                 object_counting |  86.50% |  95.50% |             93.50% |             93.50% |
|             penguins_in_a_table |  95.73% |  94.87% |             95.73% |             94.87% |
|              temporal_sequences |  97.50% |  99.50% |             99.50% |            100.00% |
| (Average)                       |  79.20% |  85.75% |             87.82% |             86.97% |

**Table S3**: detailed per-task breakdown of interleaved optimization presented in Appendix B.8 with PaLM 2
|                                    Task | Test accuracy |
|----------------------------------------:|:-------------:|
|                     boolean_expressions |        88.50% |
|                        causal_judgement |        70.00% |
|                      date_understanding |        75.50% |
|                       disambiguation_qa |        72.00% |
|                        formal_fallacies |        58.50% |
|                        geometric_shapes |        69.50% |
|                              hyperbaton |        93.50% |
|          logical_deduction_five_objects |        41.50% |
|         logical_deduction_seven_objects |        55.50% |
|         logical_deduction_three_objects |        91.00% |
|                    movie_recommendation |        79.50% |
|                multistep_arithmetic_two |        75.50% |
|                                navigate |        65.00% |
|                         object_counting |        85.50% |
|                     penguins_in_a_table |        89.74% |
|         reasoning_about_colored_objects |        74.00% |
|                              ruin_names |        86.00% |
|     salient_translation_error_detection |        58.00% |
|                                  snarks |        81.12% |
|                    sports_understanding |        82.00% |
|                      temporal_sequences |       100.00% |
|  tracking_shuffled_objects_five_objects |        52.00% |
| tracking_shuffled_objects_seven_objects |        63.00% |
| tracking_shuffled_objects_three_objects |        64.50% |
|                             web_of_lies |        89.00% |
|                            word_sorting |        80.00% |
|                               (Average) |        74.63% |

---

### Decision · Program_Chairs · 2024-09-25

**Decision:**

Accept (poster)

**Comment:**

Following the author rebuttal, all six reviewers have expressed positive ratings for this paper. I appreciate the efforts made by the authors; their work demonstrates novelty and potential impact, making it suitable for NeurIPS. However, the authors should consider the necessary revisions in the final version, particularly those highlighted by the reviewers.